# LMGame-Bench: How Good are LLMs at Playing Games?

**Lanxiang Hu**[*1] **Mingjia Huo**[*1] **Yuxuan Zhang**[†1] **Haoyang Yu**[†1] **Eric P. Xing**[2]
**Ion Stoica**[3] **Tajana Rosing**[1] **Haojian Jin**[1] **Hao Zhang**[1]
[1]UC San Diego  [2]MBZUAI  [3]UC Berkeley

## Abstract

Playing video games requires perception, reasoning, memory, and long-horizon planning—exactly the faculties expected of modern large language and vision–language models (LLMs/VLMs). We introduce LMGame-Bench, a benchmark built on six popular games spanning platformer, puzzle, and narrative games through a unified Gym-style API. Unlike prior game benchmarks that entangle multiple skills, LMGame-Bench employs a modular harness—including perception, memory, and reasoning modules—that can be toggled to selectively probe distinct capabilities. The benchmark further improves robustness through prompt standardization and contamination mitigation. Evaluation of 13 state-of-the-art models demonstrates that LMGame-Bench remains challenging yet effectively discriminates among models. Correlation analysis reveals that individual games align with core LLM capabilities, providing a quantitative framework for interpreting performance. Finally, LMGame-Bench exposes models' limitations in visual state extraction, reflection, spatiotemporal reasoning, and long-context reasoning, pointing to concrete directions for model improvement.

## 1 Introduction

Interactive games are emerging as powerful benchmarks for large language and vision–language models (LLMs/VLMs) (Shi et al., 2025; Ruoss et al., 2025; Wang et al., 2025; Paglieri et al., 2024; Costarelli et al., 2024; Wu et al., 2023). With explicit rules and measurable outcomes, games provide well-defined metrics for success. Their complexity challenges models to demonstrate diverse abilities, including reasoning, long-horizon planning, and adaptation to dynamic opponents. In addition, games contain built-in difficulty scaling from simple to complex, which makes them well-suited for distinguishing models without being easily saturated. These qualities together make games a uniquely effective testbed for evaluating frontier models.

Prior game benchmarks often entangle multiple skills at once—vision perception, reasoning, and memory—making it difficult to diagnose why models succeed or fail or to attribute errors to specific capabilities (Hu et al., 2024; Paglieri et al., 2024; Zhang et al., 2024; Ruoss et al., 2025). Even seemingly "simple" games can test LLMs/VLMs across several axes: parsing the board state from pixels, reasoning about legal and high-value moves, and maintaining coherent multi-turn strategies. When all these factors are evaluated simultaneously, results become hard to interpret and distinct failure modes remain hidden. Besides, the benchmark should be neither too hard for progress, as in Zhang et al. (2024), nor too easy and already solved by current models, as in parts of Ruoss et al. (2025). This calls for a benchmark with modular design, one that maintains challenging without saturation while isolating individual capabilities for fine-grained diagnosis.

We introduce LMGame-Bench, a benchmark built on six video games spanning platformers, puzzle solvers, and a narrative-driven detective game through a unified Gym-style API (Towers et al., 2024), evaluating core model capabilities such as vision perception, reasoning, long-horizon planning, and narrative understanding. When evaluated with a direct "screenshot→action" design, we observed that models achieve low performance, often close to random action-taking baselines. This is because

---

*Equal contributions. Correspondence to haozhang@ucsd.edu, lah003@ucsd.edu.
†Significant contributions.

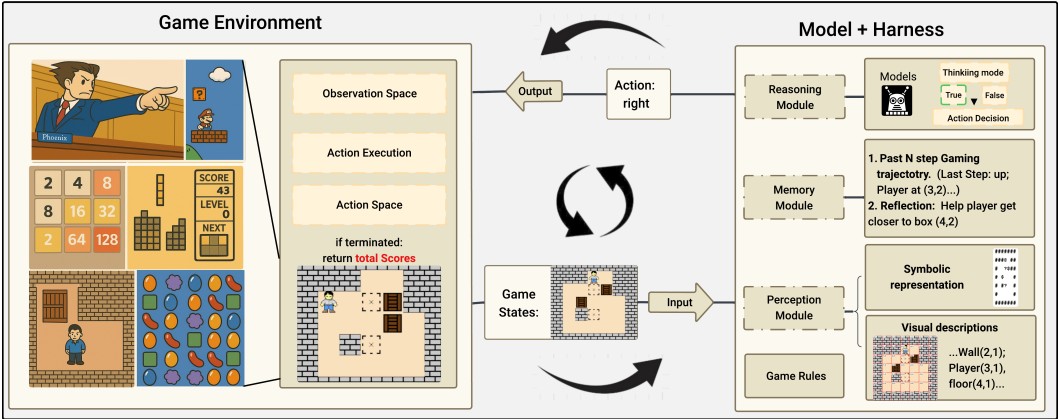

Figure 1: LMGame-Bench uses modular harnesses—such as perception, memory, and reasoning modules—to systematically extend a model's game-playing capabilities, allowing the model to engage with a simulated game environment through iterative interaction loops.

even advanced reasoning models fall short in visual perception and limited long-horizon decision making (Yang et al., 2024; Waytowich et al., 2024; Mosquera et al., 2024). To address this issue and also enable controlled evaluation, we enrich our evaluation settings by developing gaming *harness* (i.e. additional modules integrated into the agent workflow to support LLM–game interaction), including perception, memory, and reasoning modules to amortize vision perception limitations and facilitate long-horizon planning as shown in Fig. 1. By enabling individual or combined modules, we ablate perception aids, memory and reflection to isolate each skill while holding the underlying game constant, making it possible to disentangle specific strengths and weaknesses (e.g., perception vs. planning) that would otherwise remain hidden. Furthermore, LMGame-Bench improves robustness by two additional components: (i) mitigating data contamination once it is detected to emphasize reasoning over memorization, and (ii) employing a standardized prompt optimization technique to reduce variance. Together, these techniques serve as principled scaffolds that enable more reliable evaluation of LLM agents.

We evaluate leading models in LMGame-Bench. Results from 13 models across 6 games demonstrate that LMGame-Bench presents a challenging benchmark far from being saturated. The benchmark effectively differentiates models: o3 and o1 achieve top-2 best performance across all games, followed by Gemini-2.5-pro-preview and Claude-3.7-Sonnet. Among non-reasoning models, GPT-4.1 leads the pack. To move beyond raw scores, we introduce quantitative analysis based on correlation and low rank matrix factorization, linking LMGame-Bench to domain-specific benchmarks. This reveals latent relationships between individual games and isolated LLM capabilities, providing a quantitative framework for interpreting gaming performance.

LMGame-Bench exposes critical weaknesses of current models: First, current VLMs struggle to extract the states of board games (e.g. Sokoban, Tetris) directly from images (Anthropic, 2025), which is a task trivial for humans. By converting images into textual representation, our perception module significantly boosts model performance, with the largest gains observed in reasoning models that excel at textual reasoning. Meanwhile, non-reasoning models frequently fail to detect invalid moves (e.g., repeatedly making the same invalid move in 2048). Our memory module, which tracks history and enables self-reflection, helps these models self-correct. Other failures include misaligning spatial choices with temporal dynamics and losing information over long contexts. By revealing diverse failure modes, LMGame-Bench points to concrete directions for advancing both model development and agentic design.

In summary, our paper makes the following contributions:

- We introduce LMGame-Bench , the first benchmark that evaluates video games with or without scaffolds, enabling controlled comparisons across models.

- We demonstrate that, under different settings, LMGame-Bench can clearly discriminate among state-of-the-art models.

- We present quantitative analyses—including correlation and latent factor modeling—that reveal how different games align with core model capabilities.
- We show that current models exhibit persistent limitations, pointing to concrete directions for improving model capabilities.

## 2   RELATED WORK

**Games as AI Testbeds.**   Games have long served as foundational benchmarks in AI research, particularly in reinforcement learning. From TD-Gammon (Tesauro et al., 1995) to AlphaGo (Silver et al., 2017), they have offered controlled environments for studying planning and sequential decision-making. OpenAI Gym (Brockman et al., 2016) further standardized this paradigm by providing a unified interface for interacting with diverse environments. More recently, games have been adopted to evaluate LLM agents on specific domains, such as grid-based games (Nasir et al., 2024), open-ended strategy games (Hopkins et al., 2025), or murder-mystery games (Xie et al., 2024a). Evaluations on individual tasks such as Pokémon Red/Blue (Comanici et al., 2025; Anthropic, 2025) and the Kaggle AI Chess Exhibition Tournament (Risdal, 2025; Olszewska & Risdal, 2025) also highlight the potential of games as testbeds for LLMs/VLMs. Others evaluate natural language reasoning through text-based or conversational games (Costarelli et al., 2024; Hudi et al., 2025; Qiao et al., 2023; Hu et al., 2024), lacking visual understanding. Multimodal gaming benchmarks such as BALROG (Paglieri et al., 2024) evaluate grid-based navigation tasks and textual reasoning, while LMAct (Ruoss et al., 2025) examines the relationship between model performance and the number of expert demonstrations, though half of tasks are nearly solved by existing models. Meanwhile, both work primarily emphasize qualitative observations of game-related LLM abilities. VideoGameBench (Zhang et al., 2024) features complex 3D environments, but tasks are often too difficult for current models to show progress. Our work is distinct in game choice (inherent difficulty scaling), harness design (to probe diverse abilities), data contamination mitigation, and quantitative evaluation, which together provide a more systematic and durable benchmark for frontier models.

**LLM Agentic Benchmarks.**   Current agentic benchmarks tend to focus on domain-specific tasks—such as code editing (Jimenez et al., 2023), web browsing (Zhou et al., 2023; He et al., 2024), API control flows (Trivedi et al., 2024), GUI control (Agashe et al., 2024) or system operations (Xie et al., 2024b). Recent efforts also explored broader evaluation across multiple aspects (Liu et al., 2024; Mialon et al., 2023). While these benchmarks provide valuable insights into specialized domains, games offer a complementary setting that is both scalable and skill-diverse, enabling evaluation of a wider range of general agentic behaviors.

## 3   LMGAME-BENCH

We build the backbone of LMGame-Bench on six well-known games (§ 3.1) to evaluate leading models' performance without scaffolds. However, directly evaluating models on games in their original forms poses challenges, and we introduce mitigation techniques in § 3.2.

### 3.1   BENCHMARK DESIGN

In designing LMGame-Bench, we intentionally recycle well-known games not only for their familiarity and popularity but also because they encapsulate a broad spectrum of reasoning and interaction skills, and contain inherent difficulty scaling to better distinguish models. Our goal is to preserve the original game settings that are carefully designed to challenge human cognition. In this section, we highlight the broad range of perception and generation abilities evaluated by the games.

### 3.1.1   GAMES

**Super Mario Bros.** Super Mario Bros is a side-scrolling platformer game where the player controls Mario to navigate through obstacles, defeat enemies, and reach the end of each level by navigating through the environment. Success requires precise timing and strategic movement, making it a classic benchmark for evaluating (1) visual perception, (2) spatiotemporal reasoning in 2D for character control, and (3) goal-directed planning with partial observability (Rintanen, 2004).

**Tetris.** Tetris is a tile-matching puzzle game where players must strategically rotate and place falling Tetris tiles of 7 different geometric shapes to complete and clear horizontal lines. The game emphasizes (1) visual perception for pattern recognitions, (2) spatial reasoning for correct tile matching and geometric rotations (Lau-Zhu et al., 2017), and (3) long-horizon planning with partial observability for decision-making on where and how to drop a tile (Demaine et al., 2003).

**Sokoban.** Sokoban is a grid-based puzzle game where the player pushes boxes to designated target locations within confined spaces. It emphasizes (1) visual perception, (2) spatial reasoning to navigate both the character and the box, and (3) long-horizon planning to avoid deadlocks (Culberson, 1997). The game's low fault tolerance is especially pronounced. Many actions are irreversible, and a single wrong move can fail the puzzle.

**Candy Crush.** Candy Crush is a match-three puzzle game where players swap adjacent candies to form aligned sequences and trigger cascading effects to eliminate matched sequences. It requires (1) visual perception to identify different candies, (2) spatial reasoning to anticipate chain reactions at different locations, and (3) long-horizon planning to conserve moves to maximize total points. The gameplay features limited moves, making it crucial to plan moves carefully.

**2048.** 2048 is a sliding-tile puzzle game where players combine numbered tiles on a grid to reach the 2048 tile. It evaluates (1) visual perception for tracking tile values and movements, (2) spatial reasoning to manage merging paths, and (3) goal-directed planning to maximize merge potential (Zaky, 2014). Errors compound quickly due to the game's limited space and could lead to irreversible failure states.

**Ace Attorney.** Ace Attorney is an interactive courtroom-drama visual novel in which the player, acting as defense attorney Phoenix Wright, must investigate crime scenes, interview witnesses, and present evidence in court to reveal contradictions and secure a "Not Guilty" verdict. The game stresses (1) long-context language understanding, tracking hundreds of dialogue turns, testimonies, and evidentiary facts, (2) causal & deductive reasoning under partial observability—linking dispersed clues, inferring hidden motives, and spotting logical gaps, and (3) long-horizon, low-fault-tolerance decision making, to decide when to press, object, or present evidence over multi-stage trials.

### 3.1.2 GAME SETTINGS

LMGame-Bench maintains the integrity of the original design choices, which ensure scalability. In this section, we focus on standardizing game settings including inputs and outputs of the gaming environments as part of our benchmark design as shown in Fig. 1 (right). We can formalize a gaming process as a partially or fully observable Markov Decision Process (MDP) with the following definitions as a generalizable formalism applicable to all games.

**Observation Space Representations.** Many existing games are graphical, requiring players to process multimodal information, including visual, textual, and spatial cues, from the user interface to interpret game states and make decisions (Liu et al.; Zhang et al., 2025). We denote symbolic and graphical representations of the game states as the set of all possible observations $S$. We don't make a distinction between game state space and observation space, which is not the focus of this work.

**Action Space.** Let the set of all actions in action space be $\mathcal{A}$. To interface with the game, LMGame-Bench considers multi-turn interactions. LMGame-Bench streams game states $s_i \in S$ to model $M$, each time it generates action $a_i \in A$ in response to the current state to maximize rewards, which are meticulously crafted scores in classical games, with details specified in Metrics below.

**Difficulty.** Games in LMGame-Bench are designed with varying levels of difficulty, structured along two key dimensions: (1) fault tolerance and (2) state-action space complexity. We define three levels of fault tolerance -low (one wrong move fails), medium (errors accumulate but can be recovered), and high (many mistakes can be tolerated without significantly affecting future game states). We employ a memory module to curb search-space explosion (§ 3.2.1).

### 3.1.3 METRICS

In line with Gymnasium (Towers et al., 2024), we treat a reward as a function $\mathcal{R}: \mathcal{S} \times \mathcal{A} \times \mathcal{S} \rightarrow \mathbb{R}$ which returns the payoff obtained when the agent executes action $a \in \mathcal{A}$ in state $s \in \mathcal{S}$ and transitions to state $s'$. LMGame-Bench adopts this definition to reports either progression rewards or long-

horizon rewards. *Progression rewards* provide dense, stepwise feedback as the agent makes forward advances. They apply to games with a primarily linear structure or no fixed endpoint, offering incremental scores that increase with each step—e.g., Mario's horizontal distance, the running score in Tetris, 2048's cumulative merge total, or Candy Crush's cumulative eliminated candies. *Long-horizon rewards* offer sparse credit awarded only upon completing a multi-step objective. These are common in games built around multi-step puzzles or multi-stage narratives, where rewards are granted after achieving the full goal, such as solving all boxes in Sokoban or a courtroom sequence being correctly completed in Ace Attorney. To ensure comparability across games, we represent these rewards into a continuous raw score that sensitively captures performance. These raw scores are the primary evaluation signal as shown in Table 1. See Appx. A for detailed metrics for each game.

## 3.2 BENCHMARK EFFECTIVENESS ENHANCEMENT

While using games as evaluation presents challenging environments and breadth, we find that directly evaluating models on games exposes several challenges: low discriminability, contamination risk, and prompt variance. In this section, we address these issues by introducing gaming scaffolds for LLMs, contamination detection, and prompt standardization, enabling LMGame-Bench to function as a more robust benchmark that reliably differentiates LLMs.

### 3.2.1 GAMING HARNESS

Excluding text-only models, 40% of game runs without the harness fail to outperform a random-play baseline. To raise the cap and bring higher contrast, LMGame-Bench provides a suite of harness modules that can be toggled on or off for any experiment (workflow in Fig. 1). Activating the harness boosts scores far beyond both random play and the unharnessed setting, creating clearer performance gaps between models. With harnessing, 86.7% of game runs beat the random baseline, and paired-sample t-tests confirm that harnessed runs score significantly higher than their unharnessed counterparts on Candy Crush, 2048, Tetris, Ace Attorney, and Sokoban (details in Appx. B and C.3).

**Perception Modules.** Since the video games are inherently multimodal, we build perception modules that convert UI inputs into symbolic representations or textual descriptions of game states to facilitate understanding. For grid-based games (Sokoban, Candy Crush, 2048, Tetris), the module converts the visual layout into a text-based table from game backends, listing object coordinates and their properties, e.g. "Box at (2,3)". This allows models to directly understand spatial relationships in replacement of raw image inputs to minimize perception errors. For text-based games (e.g. Ace Attorney), the module extracts dialogues and describes visual elements in text format to provide narrative context and critical visual cues. Likewise, we use perception module to extract visual elements in Super Mario Bros to facilitate decision making.

**Memory Modules.** Some games, like Sokoban and Tetris, exhibit a rapidly growing decision space as gameplay advances and interactive elements scale (e.g., boxes, grid size, tetromino types). As a result, they come with higher difficulty levels than the other games. To better distinguish models, we integrate additional memory modules into LMGame-Bench. This setup allows selective activation of two components: (1) a transient memory module, which records the past $N$ game states and actions, and (2) a reflection module, which encodes explicit lessons learned to avoid failure, inducing actions in specific game states, thereby helping to narrow the action space.

**Reasoning Modules.** Reasoning models (Guo et al., 2025; OpenAI, 2025) have emerged as a new inference paradigm, where models explore multiple reasoning paths and synthesize a more accurate answer at the end. We support such reasoning traces by allowing models to be evaluated with or without long chain-of-thought (long-CoT) reasoning.

### 3.2.2 DATA CONTAMINATION

Because LMGame-Bench reuses publicly available game assets, many images and scripts may already appear in model pre-training data. To ensure the model isn't merely recalling artifacts, we test vision-level data contamination in *Super Mario Bros* and text-level data contamination in *Ace Attorney*, whose sprite and dialogue are widely distributed online (Appx. D). The other games, Tetris, 2048, Candy Crush, and Sokoban, feature combinatorial state spaces (Dor & Zwick, 1999; Demaine et al., 2003; Gualà et al., 2014), making overlap with training data negligible.

**Vision-level.** We assess whether models recall the visual structure of *Super Mario Bros* level 1-1 by prompting them to reorder shuffled RGB frames. Only a few models exhibit a moderate positive alignment, yet these alignment scores do not significantly track with their performance rankings. This suggests that they rely on local perception rather than memorized sequences. Since our metric evaluates how far models can play Super Mario Bros, we focus on vision-level contamination that may expose future frames, and disregard contamination within the current frame-such as prior knowledge that a "?" brick may contain a mushroom-as it does not affect sequence prediction.

**Text-level.** In *Ace Attorney*, we test whether the models reproduce public fan transcripts. Using Sentence-BERT similarity, we find a strong correlation between output similarity and performance, especially in a 6-model subset. However, after applying structured prompt-based mitigation—including entity masking, paraphrasing, and enforced reasoning (Dong et al., 2024; Carlini et al., 2018) —the correlation disappears, and model rankings instead align with judged reasoning quality (Fig. 2).

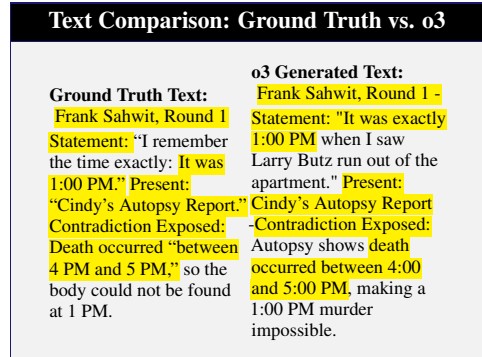
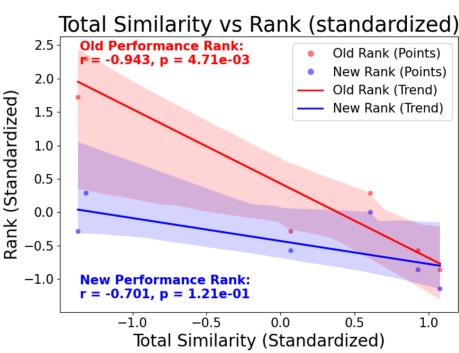

Figure 2: (Left) Example from Ace Attorney showing contradictions in O3-generated text vs. ground truth. (Right) Effect of mitigation on similarity-performance correlation; red and blue lines show correlations with old and new leaderboard ranks, respectively.

### 3.2.3 PROMPT STANDARDIZATION

Prompt engineering improves LLM performance across games (Paglieri et al., 2024; Wang et al.), yet even empirically tuned prompts can vary by more than $\pm 1\sigma$ (Table 14). To stabilize results, LMGame-Bench applies a two-stage strategy: first, we adopt a canonical agentic format $[\{\mathcal{J}_{[\min(0,i-N):i-1]}\}, R_{i-1}, s_i] \mapsto a_i$, where $\mathcal{J}$ is the trajectory of the latest $N$ turns $(s_j, a_j, r_j)$ and $R_{i-1}$ is the memory reflection; second, we use DSPy's SIMBA optimizer (Khattab et al., 2024) to iteratively refine prompts via introspective mini-batch ascent guided by game rewards. Across three runs, this standardization reduces prompt variance by 33.8%–63.5% in games such as 2048 (Appx. E), yielding more consistent performance across models.

## 4 EXPERIMENTS

In this section, we present the rankings of 13 state-of-the-art models, both with and without the gaming harness, evaluated on LMGame-Bench. We also analyze the effectiveness of each harness module (§ 4.1), as well as the modules' combined effectiveness. We then conduct a qualitative analysis on our harness design and models' failure cases (§ 4.2), and investigate how gaming environments reveal core LLM capabilities through correlation analysis, low-rank factorization, linear modeling (§ 4.3). We quantify the issue of gaming data contamination and propose mitigation techniques in Appx. D.

### 4.1 MODEL PERFORMANCE

When putting LLMs and VLMs in gaming environments, we first study if they can play the games well without gaming harness. Table 1 shows most models perform poorly. Specifically, over three fourths of models often score no points on Sokoban and Ace Attorney without harness support. On Tetris and Candy Crush, their scores are close to random play, which suggests they succeed by chance

Table 1: Model performance raw scores, evaluated in both with and without harness. For games marked with $^\dagger$, evaluation for text-only models is not supported, as vision understanding is essential for decision-making. The reported results represent averages over three runs, except for models or games marked with *, which are based on a single run due to the high costs as of May 2025. "N/A" indicates a non-applicable evaluation setting, as the specific model does not support image input.

| Model | Harness | Sokoban | Super Mario Bros$^\dagger$ | Tetris | 2048 | Candy Crush | Ace Attorney* |
|---|---|---|---|---|---|---|---|
| claude-3-5-sonnet-20241022 | No | 0.0±0.0 | 1540.0±21.7 | 12.3±2.5 | 57.8±16.4 | 17.0±18.1 | 1.0±0.0 |
| | Yes | 0.0±0.0 | 1267.7±484.1 | 14.7±1.2 | 108.2±5.8 | 106.0±53.4 | 2.0±0.0 |
| claude-3-7-sonnet-20250219 (thinking) | No | 0.0±0.0 | 1430.0±162.2 | 13.0±0.0 | 114.2±7.2 | 126.3±69.1 | 3.0±0.0 |
| | Yes | 2.3±1.5 | 1418.7±660.3 | 16.3±2.3 | 113.3±3.1 | 484.0±53.7 | 7.0±0.0 |
| deepseek-r1 | No | N/A | N/A | N/A | N/A | N/A | N/A |
| | Yes | 1.3±1.2 | N/A | 14.3±0.6 | 105.2±12.2 | 447.3±45.1 | 0.0±0.0 |
| gemini-2.5-flash-preview-04-17 (thinking) | No | 0.0±0.0 | 1540.7±262.4 | 19.0±4.6 | 107.4±3.4 | 97.7±36.1 | 1.0±0.0 |
| | Yes | 1.7±1.5 | 1395.0±240.1 | 16.3±3.2 | 106.6±5.3 | 334.7±65.5 | 4.7±1.2 |
| gemini-2.5-pro-preview-05-06 (thinking) | No | 1.0±0.0 | 1025.3±443.2 | 12.3±3.1 | 120.5±3.9 | 177.3±64.9 | 8.0±0.0 |
| | Yes | 4.3±0.6 | 1498.3±203.4 | 23.3±0.6 | 117.3±5.9 | 416.3±6.8 | 7.7±0.7 |
| grok-3-mini-beta (thinking) | No | N/A | N/A | N/A | N/A | N/A | N/A |
| | Yes | 5.7±0.6 | N/A | 21.3±7.1 | 118.6±7.1 | 254.0±107.8 | 0.0±0.0 |
| llama-4-maverick-17b-128e-instruct-fp8 | No | 0.0±0.0 | 786.0±462.6 | 11.7±1.2 | 44.6±11.8 | 32.3±41.4 | 0.0±0.0 |
| | Yes | 0.0±0.0 | 1468.7±555.7 | 10.3±1.5 | 106.0±3.8 | 128.7±57.2 | 0.0±0.0 |
| gpt-4.1-2025-04-14 | No | 0.0±0.0 | 1991.3±1018.5 | 13.0±1.7 | 94.5±17.0 | 101.0±120.2 | 0.0±0.0 |
| | Yes | 0.0±0.0 | 2126.3±1778.4 | 13.7±0.6 | 105.7±7.0 | 182.0±28.7 | 3.3±1.2 |
| gpt-4o-2024-11-20 | No | 0.0±0.0 | 1028.3±656.0 | 14.7±2.1 | 70.4±15.2 | 59.0±54.6 | 0.0±0.0 |
| | Yes | 0.0±0.0 | 2047.3±528.2 | 14.0±3.6 | 106.7±3.5 | 147.3±53.4 | 0.0±0.0 |
| o1-2024-12-17 * | No | 0.0±0.0 | 1434.0±0.0 | 13.0±0.0 | 128.1±0.0 | 90.0±0.0 | 3.0±0.0 |
| | Yes | 2.3±0.6 | 855.0±0.0 | 35.0±0.0 | **128.9±0.0** | 159.0±0.0 | **16.0±0.0** |
| o1-mini-2024-09-12 | No | N/A | N/A | N/A | N/A | N/A | N/A |
| | Yes | 1.3±0.6 | N/A | 11.7±1.2 | 114.0±3.7 | 48.0±33.9 | 0.0±0.0 |
| o3-2025-04-16 * | No | 2.0±0.0 | 1955.0±0.0 | 31.0±0.0 | 128.2±0.0 | 106.0±0.0 | 8.0±0.0 |
| | Yes | **8.0±2.8** | **2266.7±1219.9** | **42.0±0.0** | 128.6±5.6 | **647.0±0.0** | **11.3±4.1** |
| o4-mini-2025-04-16 | No | 1.3±0.6 | 1348.3±178.1 | 15.0±3.6 | 97.6±29.2 | 110.7±49.7 | 2.0±0.0 |
| | Yes | 5.3±1.2 | 1448.0±161.0 | 25.3±8.5 | 120.6±4.9 | 487.3±198.0 | 2.7±1.2 |
| Random | – | 0.0±0.0 | 987.0±414.5 | 10.2±1.8 | 100.4±7.8 | 116.5±51.5 | 0.0±0.0 |
| Human (avg) | – | 9.7±3.9 | 4333.3±2718.3 | 353.3±139.6 | 115.5±10.9 | 283.3±10.7 | 17.3±3.4 |

rather than understanding. As a result, it's numerically hard to distinguish models given poor model performances and randomness inherent to games.

To address this issue, we design different levels of harness, as described in § 3.1.2, to better differentiate model capabilities. Compared with our three human players (graduate students given only a brief explanation of the game rules), the gaming harness shifts models from mostly underperforming humans to performing at a comparable level or even surpassing them. Results in Table 2 also show that the harness leads to consistent and sometimes substantial gains for both reasoning and non-reasoning models, with a complete model performance listed in Table 4. We provide detailed analysis for each module below.

**Perception Modules.** In grid-based games like Sokoban, vision harness helps models perform better by providing them with textual representation of game states read from the game backend. Reasoning models show substantial improvements, revealing that structured spatial inputs can unlock planning capabilities not expressed under image inputs. In games with more complex graphical interface like Super Mario Bros, we uses o3 to generate textual descriptions. However, this module plays a less significant role, as a substantial gap remains between textual descriptions and the spatiotemporal information required for accurate decision-making.

**Memory Modules.** Memory proves particularly impactful in temporally extended games. In 2048, non-reasoning models improve substantially with memory support, not only raises average scores but also reduces variance (Table 4), highlighting its importance in long-horizon planning. The effect is even more dramatic in Candy Crush, a game with complex temporal dependencies and delayed rewards. This reinforces memory's role in preserving context and strategy over time.

**Combined Support.** Table 2 shows that enabling both modules or one essential module often leads to stronger performance. Detailed statistical analysis (Appx. B) shows that harness can also pull scores far away from random play, statistically improve performance, and reduce variance for more stable benchmark results.

Table 2: Model performance on *Sokoban*, *2048*, *Tetris*, and *Candy Crush* under various conditions. Scores are averaged across reasoning models (o4-mini, Gemini-2.5-Pro, Claude-3.7-Sonnet) and non-reasoning models (Llama-4-Maverick, Claude-3.5-Sonnet, GPT-4o). ZS indicates zero-shot without any module support or memory prompt. See Table 4 for complete per-model results.

| Model Group | Game | ZS | +Memory | +Perception | +Both |
|---|---|---|---|---|---|
| Reasoning Models | Sokoban | 0.9 | 0.9 | **4.0** | **4.0** |
| Non-reasoning Models | | 0.0 | 0.0 | 0.0 | 0.0 |
| Reasoning Models | 2048 | 111.0 | 113.4 | 116.6 | **117.1** |
| Non-reasoning Models | | 57.6 | 102.5 | 71.1 | **107.0** |
| Reasoning Models | Tetris | 13.4 | 15.1 | **26.4** | 21.6 |
| Non-reasoning Models | | 12.9 | 12.4 | 13.7 | **13.9** |
| Reasoning Models | Candy Crush | 138.1 | 161.1 | 229.7 | **462.5** |
| Non-reasoning Models | | 36.1 | 97.6 | 66.3 | **127.3** |

## 4.2 QUALITATIVE ANALYSIS

This section presents examples and qualitative insights into our harness design and discusses representative failure cases across the evaluated games, highlighting clear directions for model improvement. All examples are presented in Appx. C.

**Image Perception.** Current LLMs struggle with visual inputs. For instance, when given a screenshot of a game board from Tetris or Sokoban, they fail to convert it into a textual grid or a coordinate–object list (Fig. 5), revealing a surprising limitation. In contrast, for 2048 the $4 \times 4$ board is small enough to fall within current models' visual perception capabilities. This pattern aligns with Table 4, where the perception module benefits Tetris, Sokoban, and Candy Crush, but offers little advantage for 2048 for Gemini-2.5-Pro and Claude-3.7-Sonnet. These foundational perceptual skills are preliminary steps toward image reasoning, yet remain underdeveloped in state-of-the-art models.

**Reflection on Failures.** Without memory-harness support, models frequently fall into loops of repeating the same ineffective or invalid actions, such as endlessly proposing impossible merges in 2048 or pushing boxes into walls in Sokoban. This is most evident in non-reasoning models (Table 2), which lack reasoning abilities to identify valid moves. In contrast, with appended history and a self-reflection mechanism in our memory module, models can recognize when a previous action is invalid (Fig. 7). These results demonstrate that harness-level memory provides an effective way to mitigate repeated failures and improve task efficiency.

We also compare reflection quality between GPT-4o and o3 (Appx. C.3). GPT-4o focuses on surface effects ("up increased empty spaces but no merges"), while o3 provides fine-grained analysis ("two 2's became a 4, two 16's became 32's, opening an extra cell") and ties it to long-term corner-dominance strategy. Unlike one-off Q&A, games require multi-turn reasoning where each move shapes the next. o3 excels by linking immediate outcomes to future plans, showing how local decisions drive long-term success.

**Spatiotemporal Reasoning.** In Super Mario Bros, models often fail to coordinate actions with the temporal dynamics of the environment. A correct policy must not only decide which action to take but also how long to sustain it. Two typical failure cases (Fig. 8) illustrate this limitation: when encountering a tall pipe, models frequently execute a jump with too few frames, failing to gain the necessary height; and when approaching a gap, they often initiate the jump too early, losing forward momentum before reaching the far edge. These mistakes reveal weak spatiotemporal reasoning and highlight the difficulty of synchronizing spatial planning with temporal duration.

**Long-Context.** In Ace Attorney, which requires reasoning across long stretches of dialogue, we observe frequent failures in maintaining consistency with earlier evidence (Fig. 9). For instance, even when a direct contradiction is available in the evidence list, models sometimes fail to surface it and instead continue the cross-examination without objection. These errors highlight the difficulty of long-horizon retrieval and attribution: information may be present in the context window, but models do not reliably connect it to new claims.

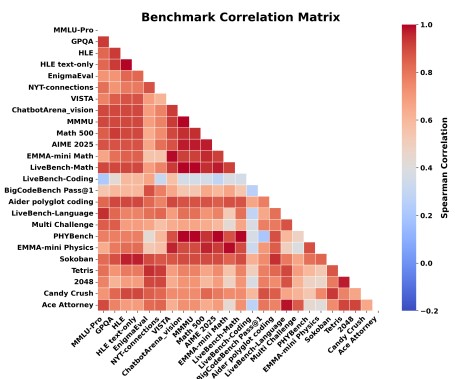 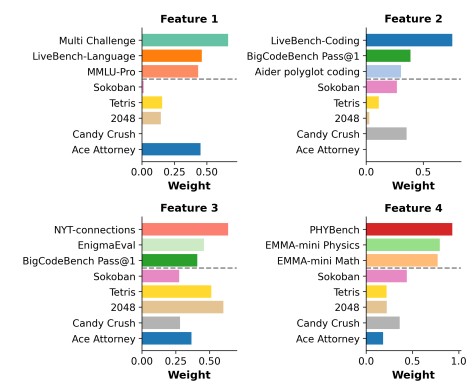

(a) Spearman Correlation among LMGame-Bench and other benchmarks.

(b) Top-weight benchmarks under each feature after low-rank decomposition.

Figure 3: Correlation analysis and latent feature decomposition among benchmarks.

## 4.3 CORRELATION ANALYSIS

Games are designed to challenge human recognition skills, requiring a combination of core LLM capabilities for strong performance. To study these connections, we collect results from 8 models on 20 established benchmarks spanning factual knowledge, physics, mathematics, coding, visual reasoning, language understanding, and puzzle solving, and analyze their correlations with gaming performance below. See Appx. F.1 for the complete model and benchmark list, and Table 16 for an additional analysis on how each game decomposes into core LLM skills through linear modeling.

**Correlation Analysis**. We calculate Spearman's rank correlation coefficient to assess alignment between model performance on games from LMGame-Bench and widely-used benchmarks. Results from Fig. 3 (left) reveal positive correlations between several games and commonly used benchmarks. Sokoban correlates strongly with math and coding benchmarks, while Tetris and 2048 align with pattern recognition tasks such as EnigmaEval and NYT-Connections. Candy Crush shows links to coding, hinting at algorithmic reasoning, and Ace Attorney correlates with LiveBench-Language, reflecting narrative understanding. Super Mario Bros is excluded, as the spatiotemporal reasoning it requires is unique compared with the listed benchmarks, with further details in Appx. F.2.

**Latent Ability Decomposition.** To uncover relationships between the benchmarks and the capabilities of the model, we apply a low-rank matrix factorization to the model–benchmark performance matrix. This decomposes each LLM as a vector in latent ability space and each benchmark, including our games, as a sparse, weighted combination of these abilities. As shown in Fig. 3b, the four features align with language and multi-task knowledge, coding, symbolic and puzzle-solving skills, and physical reasoning, respectively. The results show our games require different subsets of these latent abilities. *Sokoban* emphasizes symbolic and physical reasoning (Features 3 and 4), while *Ace Attorney* strongly engages long-context language reasoning (Feature 1). *Tetris* and *2048* mainly represent spatial reasoning (Features 3), and *Candy Crush* reflects visual pattern recognition with moderate ties to coding (Features 2 and 3). This suggests that games cover compositional capabilities rather than isolated skills.

## 4.4 OTHER LIMITATIONS AND DISCUSSION

Overall, our design is effective in distinguishing models, identifying key failure modes, and evaluating core LLM capabilities. However, two limitations still remain. (1) Performance variance continues to be high in partially observable games like Super Mario Bros. Notably, human performance also exhibits higher variance in this game, suggesting that the variability arises from the game's stochastic dynamics. (2) The computational cost remains substantial, as generating actions could result in long reasoning chains that are highly repetitive across multiple turns (Appx. B.4), highlighting the need for improved model capabilities and more efficient inference to reduce operational costs.

## 5 CONCLUSION

We introduce LMGame-Bench, the first agentic benchmark for evaluating LLMs on games with and without gaming harness support. LMGame-Bench leverages a gaming harness composed of agentic modules to better distinguish state-of-the-art models. Our benchmark identifies and addresses data contamination through a series of mitigation, and it reduces prompt variance by integrating a two-stage prompt optimization using DSPy. Additionally, we show that LMGame-Bench can be regarded as a composition of core LLM capabilities, supported by a comprehensive quantitative analysis. By identifying the key causes of model failures, our qualitative analysis provides clear insights for advancing model development.

## ETHICS STATEMENT

All authors have read and adhere to the ICLR Code of Ethics. This work does not involve human subjects, sensitive personal data, or experiments with potential to cause harm. No confidential or proprietary data were used. The methods and experiments are designed in compliance with principles of research integrity, fairness, and transparency. We acknowledge that any potential societal impacts, including limitations or biases of large language models, are explicitly discussed in the paper, and all conclusions are the sole responsibility of the authors.

## REPRODUCIBILITY STATEMENT

We have made significant efforts to ensure the reproducibility of our results. Detailed descriptions of the models, training procedures, and hyperparameters are included in the main text and Appendix. All datasets used are publicly available, and the preprocessing steps are fully documented. Ablation studies are provided to validate robustness of results. These resources collectively allow independent researchers to verify and build upon our work.

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

# A    DESIGN DETAILS

## A.1    METRICS

Because games are well-designed, we utilize their built-in metrics to quantify models' proficiency. For each game, we choose the single score that most faithfully reflects a model's capability, then transform and normalize it onto a continuous, linear scale. This curation ensures that our evaluation can sensitively capture performance differences and supports consistent statistical analysis, and we call it *raw scores* (Table 1).

In addition to the raw scores, we introduce *procedural progress score* to capture the critical information in games, such as obstacle-passed count, box placement, and tile milestones, and we report the results in Table 3. Because procedural progress scores emphasize only key game states, they can be coarse and sometimes insufficiently discriminative across models. Hence, the main paper reports raw scores, while procedural metrics are provided here for completeness. Here are the detailed design of metrics.

- **Sokoban:**
  - Raw score: Total number of boxes pushed onto targets, summed over all levels, until the first deadlock.
  - Procedural progress score: The highest level reached and the number of boxes successfully placed in the last level.

- **Super Mario Bros.:**
  - Raw score: Cumulative horizontal distance traveled by Mario (in game units) across all levels, until all three lives are lost or the final level is completed.
  - Procedural progress score: The total number of enemies, pipes, and gaps successfully passed, which captures a model's ability to navigate structure rather than just distance traveled. Note that none models have entered the wrap zone in level 1-2 (only one human evaluator entered).

- **Tetris:**
  - Raw score: Total reward equals pieces placed plus ten times the number of lines cleared, measured until game over. Each placed piece yields +1; each cleared line yields +10.
  - Procedural progress score: The number of lines that have been cleared.

- **2048:**
  - Raw score: Sum of all merged tile values (e.g. merging two 2's yields +4), recorded until the board stagnates (no merges or moves that change the board for ten consecutive turns). We then report

$$\text{Score}_{2048} \;=\; 10 \times \log_2\!\big(\text{total merged sum}\big).$$

  - Procedural progress score: The max-tile milestones (e.g. 64/128/256/512), reporting the majority score.

- **Candy Crush:**
  - Raw score: Total number of candies eliminated over a fixed 50–move session.
  - Procedural progress score: The same as the Linear progress score.

- **Ace Attorney:**
  - Raw score: Total count of correct actions (evidence submissions, dialogue choices, etc.) across all case levels, measured until five incorrect decisions (lives) have been used.
  - Procedural progress score: The highest level reached and the number of courtroom actions completed within that level.

Table 3: Model performance measured by average finished game progress.

| Model | Harness | Sokoban Level / Box | Super Mario Bros† Obstacles Passed | Ckpt Passed | Tetris Cleared Lines | Additional Pieces | 2048 Max Tile | Candy Crush Cleared Boxes | Ace Attorney* Level / Step |
|---|---|---|---|---|---|---|---|---|---|
| claude-3-5-sonnet-20241022 | No | L1 - 0.0 Box | 9.0 | 0.77 | 0 | 12 | 16 | 17.0 | L1 - 1.0 Step |
| | Yes | L1 - 0.0 Box | 7.3 | 0.63 | 0 | 15 | 128 | 106.0 | L1 - 2.0 Step |
| claude-3-7-sonnet-20250219 (thinking) | No | L1 - 0.0 Box | 8.3 | 0.71 | 0 | 13 | 256 | 126.3 | L1 - 3.0 Step |
| | Yes | L2 - 0.3 Box | 7.7 | 0.67 | 0 | 16 | 256 | 484.0 | L2 - 2.0 Step |
| deepseek-r1 | No | N/A | N/A | N/A | N/A | N/A | N/A | N/A | N/A |
| | Yes | L1 - 1.3 Box | N/A | N/A | 0 | 14 | 128 | 447.3 | L1 - 0.0 Step |
| gemini-2.5-flash-preview-04-17 (thinking) | No | L1 - 0.0 Box | 8.7 | 0.74 | 0 | 16 | 128 | 97.7 | L1 - 1.0 Step |
| | Yes | L1 - 1.7 Box | 8.0 | 0.69 | 0 | 19 | 128 | 334.7 | L1 - 4.0 Step |
| gemini-2.5-pro-preview-05-06 (thinking) | No | L1 - 1.0 Box | 6.0 | 0.51 | 0 | 16 | 256 | 177.3 | L2 - 3.0 Step |
| | Yes | L3 - 1.3 Box | 8.7 | 0.75 | 1 | 13 | 256 | 416.3 | L2 - 2.0 Step |
| grok-3-mini-beta (thinking) | No | N/A | N/A | N/A | N/A | N/A | N/A | N/A | N/A |
| | Yes | L4 - 0.7 Box | N/A | N/A | 0.33 | 11 | 256 | 254.0 | L1 - 0.0 Step |
| llama-4-maverick-17b-128e-instruct-fp8 | No | L1 - 0.0 Box | 4.7 | 0.39 | 0 | 10 | 16 | 32.3 | L1 - 0.0 Step |
| | Yes | L1 - 0.0 Box | 8.3 | 0.73 | 0 | 12 | 128 | 128.7 | L1 - 0.0 Step |
| gpt-4.1-2025-04-14 | No | L1 - 0.0 Box | 10.3 | 1.06 | 0 | 13 | 64 | 101.0 | L1 - 0.0 Step |
| | Yes | L1 - 0.0 Box | 9.3 | 1.00 | 0 | 14 | 128 | 182.0 | L1 - 2.0 Step |
| gpt-4o-2024-11-20 | No | L1 - 0.0 Box | 5.7 | 0.51 | 0 | 14 | 32 | 59.0 | L1 - 0.0 Step |
| | Yes | L1 - 0.0 Box | 10.7 | 1.08 | 0 | 15 | 128 | 147.3 | L1 - 0.0 Step |
| o1-2024-12-17 * | No | L1 - 0.0 Box | 8.0 | 0.68 | 0 | 13 | **512** | 90.0 | L1 - 3.0 Step |
| | Yes | L2 - 0.3 Box | 6.0 | 0.53 | 1 | 18 | **512** | 159.0 | **L3 - 2.0 Step** |
| o1-mini-2024-09-12 | No | N/A | N/A | N/A | N/A | N/A | N/A | N/A | N/A |
| | Yes | L1 - 1.3 Box | N/A | N/A | 0 | 12 | 256 | 48.0 | L1 - 0.0 Step |
| o3-2025-04-16 * | No | L2 - 0.0 Box | 10.0 | 1.04 | 1 | 11 | **512** | 106.0 | L2 - 3.0 Step |
| | Yes | **L5 - 0.0 Box** | **15.0** | 1.72 | **2** | 12 | 512 | **647.0** | **L3 - 2.0 Step** |
| o4-mini-2025-04-16 | No | L1 - 1.3 Box | 7.7 | 0.68 | 0 | 15 | 256 | 110.7 | L1 - 2.0 Step |
| | Yes | L4 - 0.3 Box | 8.3 | 0.71 | 0.33 | 15 | **512** | 487.3 | L1 - 4.0 Step |
| Random | – | L1 - 0.0 Box | 6.5 | 0.63 | 0 | 17 | 64 | 116.5 | L1 - 0.0 Step |
| Human (avg) | – | L5 - 1.7 Box | 22.7 | 2.35 | 33.0 | 15 | 256 | 283.3 | L3 - 3.3 Step |

## A.2 Prompts of Harness Modules

### A.2.1 Memory Module

We design the memory module to consist of both game history and self-reflection. Below we provide the prompt for self-reflection using 2048 as an example. Prompts for other games follow the same overall structure, with game-specific instructions adapted to their respective rules.

---

**Memory Module (Game: 2048)**

**system_prompt:**
You are an analytical assistant for a 2048 AI agent.
Your task is to generate a brief, insightful reflection on the game state changes and the effectiveness of recent actions.Focus on strategic insights and patterns that would help the agent make better decisions.
Keep your reflections short, precise, and actionable.

**user_prompt:**
Please analyze the following 2048 game states and actions to generate a brief reflection:
Previous Game States and Actions:
{PREVIOUS GAME HISTORY}
Focus your reflection on:
1. How the game state changed after the last action
2. Whether the action was effective for the situation
3. Patterns or issues to be aware of
4. Any strategic insights for future actions
Keep your reflection under 100 words and focus only on the most important insights.

---

### A.2.2 Perception Module

Our benchmark allows toggling the perception module on or off, enabling controlled evaluation of a model's image perception capability. As described in Section 3.2.1, for board games, if we want to bypass perception entirely and only evaluate reasoning, the perception module **directly reads the game backend** to obtain precise textual states. However, for Ace Attorney and Super Mario Bros., designing reliable rule-based extractors is non-trivial due to their rich, nuanced visual scenes. Therefore, we adopt o3, a **state-of-the-art VLM** at the time of paper writing, to generate structured textual descriptions of the game state, with prompts shown below. As illustrated in the prompts, the VLM is used solely for parsing the game state, including identifying objects, dialogue, UI components, and available options.

**Perception Module (Game: Ace Attorney)**

You are now playing courtroom games. Carefully analyze the current scene and provide the following information.

1. Game State Detection Rules:
 - Cross-Examination mode is indicated by ANY of these:
  * Green dialog text
  * Options in screen
  * An evidence window visible in the middle of the screen
 - Conversation mode is indicated by:
  * Dialog text can be any color (most commonly white, but also blue, red, etc.)
  * Or none of the Cross-Examination indicators are present

2. Dialog Text Analysis:
 - Look at the bottom area where dialog appears
 - Note the color of the dialog text (green/white/blue/red)
 - Determine if the current dialog is a full sentence
 - Extract the speaker's name and their dialog
 - Format must be exactly: Dialog: NAME: dialog text

3. Scene Analysis:
 - Describe any visible characters and their expressions/poses
 - Describe any other important visual elements or interactive UI components
 - You MUST explicitly mention:
  * If the current dialogue is incomplete and there is a down arrow at the bottom of text box (NOT left arrow or right arrow), then set Dialogue Continuation to 'Yes'; otherwise, set it to 'False'.
  * Whether there is exclamation mark icons in the upper right corner
  * The exact UI elements present at the upper corner 'L Press' and 'Present R' if in cross examination mode
  * Whether there is an evidence window visible
  * If options appears, you need to mention:
   - The text of each option in order from top to bottom
   - Which one is currently selected (Use the pointing hand icon at the beginning of the line to determine the selected option. Do NOT assume the bottom option is selected by default — selection depends entirely on the hand icon.)
  * If evidence window is visible, you need to mention:
   - Name of the currently selected evidence
   - Whether this is the evidence you intend to present

Format your response EXACTLY as:
Game State: <'Cross-Examination' or 'Conversation'>
Dialog: NAME: dialog text
Dialogue Continuation: <'Yes' or 'No'>
Options: option1, selected; option2, not selected; option3, not selected
Evidence: NAME: description
Scene: <detailed description including dialog color, options text (if exist), blue bar presence, UI elements, evidence window status and contents, and other visual elements>

---

**Perception Module (Game: Super Mario Bros.)**

Analyze this frame from Super Mario Bros. with the 5x5 grid overlay and identify game elements in each grid cell. Your task is to identify and locate game elements in a 5x5 grid overlay on the screen.

Identify the following elements and their approximate positions in (x,y) grid coordinates (0,0 top-left to 4,4 bottom-right):
- Mario (player character)
- Pipes (green obstacles)
- Goombas (brown mushroom enemies)
- Koopas (turtle enemies)
- Gaps/pits (areas where Mario can fall)
- Question blocks (blocks with ? that can be hit)
- Brick blocks (breakable blocks)
- Coins
- Power-ups (if visible)
- Flag pole (end of level)

Your response must be in valid JSON format with the following structure:

```
{
 "mario": {"x": int, "y": int},
 "environment": {
   "pipes": [{"x": int, "y": int, "height": "small | medium | large"}],
   "goombas": [{"x": int, "y": int, "distance": "very_close | close | medium | far"}],
   {OMIT THE STATE FORMAT FOR OTHER OBJECTS}
 },
 "game_state": {
   "scroll_direction": "right | left | stationary",
   "mario_state": "small | big | fire | invincible",
   "immediate_threats": ["goomba" | "koopa" | "gap" | "pipe"],
   "obstacles_ahead": ["goomba" | "koopa" | "gap" | "pipe"]
 }
}
```
Ensure all coordinates are integers within the 0-4 range for the 5x5 grid. If an element is not present, include it as an empty array or null as appropriate. For immediate_threats, only include elements that pose an immediate danger to Mario. Your output should be ONLY the JSON object, without any surrounding text or markdown.

## B  HARNESS EFFECTIVENESS: QUANTITATIVE ANALYSIS

In this section, we employ statistical methods to explore the effectiveness of applying all harness combined in bringing improvements to model performance. Given the tiny sample due to the cost of running latest models, results should be considered preliminary. We also present the complete result of gaming harness ablation in Table 4 as a complementary of Table 2.

Given that gameplay inherently involves random noise, we aim for our harnessed model performance to be both noise-resistant and consistent, enabling clearer assessment of the model's true ability. Accordingly, we make two key claims: (1) harnessed evaluations better isolate model ability from game randomness; and (2) harnessed performance is more consistent and robust to random variation.

### B.1  SEPARATION FROM RANDOM BASELINE: GLASS'S $\delta$ EFFECT SIZES

To quantify how far harnessed and unharnessed model evaluations depart from random play, we simulated 30 random runs per game to estimate the baseline mean $\bar{X}_{\mathrm{rand}}$ and standard deviation $s_{\mathrm{rand}}$. Glass's $\delta$ for each model–game–condition is then (Glass, 1976):

$$\delta = \frac{\bar{X}_{\mathrm{model}} - \bar{X}_{\mathrm{rand}}}{s_{\mathrm{rand}}} \qquad (1)$$

Because Sokoban and Ace Attorney exhibit zero variance under random play, we exclude them, focusing on the four remaining games. Importantly, harnessed runs yield positive $\delta$ in 38 out of

Table 4: Game scores of different models in *Sokoban*, *2048*, *Tetris*, and *Candy Crush* under various conditions. ZS indicates zero-shot without any module support or memory prompt.

| Model | Game | ZS | +Memory Only | +Perception Only | +Both |
|---|---|---|---|---|---|
| o4-mini-2025-04-16 | Sokoban | $1.3\pm0.6$ | $1.3\pm0.6$ | $\mathbf{5.3\pm2.1}$ | $5.3\pm1.2$ |
| gemini-2.5-pro-03-25 | | $1.0\pm0.0$ | $1.0\pm0.0$ | $\mathbf{6.0\pm2.0}$ | $4.3\pm0.6$ |
| claude-3-7-sonnet | | $0.0\pm0.0$ | $0.3\pm0.6$ | $0.7\pm0.6$ | $\mathbf{2.3\pm1.5}$ |
| llama-4-maverick | | $0.0\pm0.0$ | $0.0\pm0.0$ | $0.0\pm0.0$ | $0.0\pm0.0$ |
| claude-3-5-sonnet | | $0.0\pm0.0$ | $0.0\pm0.0$ | $0.0\pm0.0$ | $0.0\pm0.0$ |
| gpt-4o-2024-11-20 | | $0.0\pm0.0$ | $0.0\pm0.0$ | $0.0\pm0.0$ | $0.0\pm0.0$ |
| o4-mini-2025-04-16 | 2048 | $97.6\pm29.2$ | $115.1\pm9.7$ | $117.0\pm6.4$ | $\mathbf{120.6\pm4.9}$ |
| gemini-2.5-pro-03-25 | | $\mathbf{120.5\pm3.9}$ | $118.0\pm8.5$ | $117.4\pm5.8$ | $117.3\pm5.9$ |
| claude-3-7-sonnet | | $114.2\pm7.2$ | $107.1\pm5.1$ | $\mathbf{115.3\pm2.3}$ | $113.3\pm3.1$ |
| llama-4-maverick | | $44.6\pm11.8$ | $98.1\pm3.8$ | $73.7\pm15.6$ | $\mathbf{106.0\pm3.8}$ |
| claude-3-5-sonnet | | $57.8\pm16.4$ | $102.5\pm1.6$ | $66.3\pm9.6$ | $\mathbf{108.2\pm5.8}$ |
| gpt-4o-2024-11-20 | | $70.4\pm15.2$ | $\mathbf{107.0\pm6.3}$ | $73.3\pm5.4$ | $106.7\pm3.5$ |
| o4-mini-2025-04-16 | Tetris | $15.0\pm3.6$ | $14.7\pm3.2$ | $\mathbf{38.0\pm11.3}$ | $25.3\pm8.5$ |
| gemini-2.5-pro-03-25 | | $12.3\pm3.1$ | $15.0\pm1.0$ | $21.3\pm3.5$ | $\mathbf{23.3\pm0.6}$ |
| claude-3-7-sonnet | | $13.0\pm0.0$ | $15.7\pm3.1$ | $\mathbf{20.0\pm3.6}$ | $16.3\pm2.3$ |
| llama-4-maverick | | $\mathbf{11.7\pm1.2}$ | $10.3\pm1.5$ | $8.7\pm1.5$ | $10.3\pm1.5$ |
| claude-3-5-sonnet | | $12.3\pm2.5$ | $13.3\pm4.7$ | $14.3\pm3.1$ | $\mathbf{14.7\pm1.2}$ |
| gpt-4o-2024-11-20 | | $14.7\pm2.1$ | $14.0\pm2.0$ | $\mathbf{18.0\pm6.6}$ | $16.7\pm3.5$ |
| o4-mini-2025-04-16 | Candy Crush | $110.7\pm49.7$ | $202.3\pm88.0$ | $320.0\pm3.5$ | $\mathbf{487.3\pm198.0}$ |
| gemini-2.5-pro-03-25 | | $177.3\pm64.9$ | $93.7\pm58.4$ | $386.7\pm138.5$ | $\mathbf{416.3\pm6.8}$ |
| claude-3-7-sonnet | | $126.3\pm69.1$ | $187.3\pm151.6$ | $270.3\pm240.0$ | $\mathbf{484.0\pm53.7}$ |
| llama-4-maverick | | $32.3\pm41.4$ | $123.3\pm83.9$ | $110.0\pm23.4$ | $\mathbf{128.7\pm57.2}$ |
| claude-3-5-sonnet | | $17.0\pm18.1$ | $\mathbf{120.3\pm41.5}$ | $10.7\pm8.6$ | $106.0\pm53.4$ |
| gpt-4o-2024-11-20 | | $59.0\pm54.6$ | $49.3\pm38.4$ | $78.3\pm24.7$ | $\mathbf{147.3\pm53.4}$ |

40 model–game pairs, compared to only 26 out of 40 for unharnessed runs—demonstrating that harnessed evaluations are far more consistently pulled away from the random baseline. Across those 40 pairs, harnessed runs outperform unharnessed in 29 cases (72.5%), with overall averages

$$\bar{\delta}_{\text{harness}} = 3.334, \quad \bar{\delta}_{\text{no}} = 0.750, \quad \Delta^* = \bar{\delta}_{\text{harness}} - \bar{\delta}_{\text{no}} = 2.585 \tag{2}$$

This demonstrates that the harness pulls model scores substantially farther from randomness than unharnessed evaluations.

## B.2 DIRECT COMPARISON OF HARNESSED VS. UNHARNESSED: PAIRED-SAMPLE T-TEST

Beyond Glass's $\delta$, we directly compare harnessed and unharnessed mean scores via paired-sample t-tests (Gosset, 1908) across our ten models for each game. All six games exhibit positive mean improvements under harnessing; for five of them—Candy Crush (+217.50 points, $t(9) = 4.22$, $p = 0.0022$), Sokoban (+1.97 points, $t(9) = 3.02$, $p = 0.0144$), 2048 (+17.81 points, $t(9) = 2.36$, $p = 0.0424$), Ace Attorney (+3.20 points, $t(9) = 2.36$, $p = 0.0427$), and Tetris (+5.60 points, $t(9) = 2.27$, $p = 0.0490$)—the increase is statistically significant at $p < 0.05$. Super Mario Bros. shows a smaller, non-significant gain (+289.10 points, $t(9) = 1.45$, $p = 0.1806$).

Figure 4 displays the full distribution of per-model score differences (Harness – No Harness) for each game, with boxes indicating the interquartile range and whiskers covering 1.5× IQR. Candy Crush and Sokoban show the largest median gains, while Super Mario Bros. exhibits the greatest spread, underscoring its high inherent stochasticity.

## B.3 CONSISTENCY OF PERFORMANCE: COEFFICIENT OF VARIATION ACROSS CONDITIONS

Because our per-model samples are small ($n \approx 3$), raw variance comparisons can be misleading. We therefore compute the coefficient of variation (expressed as a percentage)

Table 5: Glass's $\delta$ per Model, Condition, and Game (rounded to 3 decimals)

| Model | Cond. | 2048 | Candy Crush | SMB | Tetris |
|---|---|---|---|---|---|
| claude-3-5-sonnet-20241022 | With | 0.992 | –0.204 | 1.763 | 2.524 |
| | Without | –5.446 | –1.933 | 2.593 | 1.215 |
| claude-3-7-sonnet-20250219 (thinking) | With | 1.648 | 7.140 | 2.223 | 3.459 |
| | Without | 1.752 | 0.191 | 2.258 | 1.589 |
| gemini-2.5-flash-preview-04-17 (thinking) | With | 0.787 | 4.238 | 2.151 | 3.459 |
| | Without | 0.883 | –0.366 | 2.595 | 4.955 |
| gemini-2.5-pro-preview-05-06 (thinking) | With | 2.148 | 5.825 | 2.466 | 7.386 |
| | Without | 2.558 | 1.182 | 1.024 | 1.215 |
| gpt-4.1-2025-04-14 | With | 0.675 | 1.273 | 4.382 | 1.963 |
| | Without | –0.762 | –0.301 | 3.970 | 1.589 |
| gpt-4o-2024-11-20 | With | 0.793 | 0.599 | 4.141 | 2.150 |
| | Without | –3.833 | –1.117 | 1.033 | 2.524 |
| llama-4-maverick-17b-128e-instruct-fp8 | With | 0.707 | 0.236 | 2.376 | 0.093 |
| | Without | –7.124 | –1.635 | 0.293 | 0.841 |
| o1-2024-12-17 | With | 3.631 | 0.826 | 0.504 | 13.930 |
| | Without | 3.530 | –0.515 | 2.270 | 1.589 |
| o3-2025-04-16 | With | 3.516 | 10.306 | 8.404 | 17.856 |
| | Without | 3.541 | –0.204 | 3.859 | 11.686 |
| o4-mini-2025-04-16 | With | 2.577 | 7.204 | 2.313 | 8.508 |
| | Without | –0.368 | –0.113 | 2.009 | 2.711 |

Table 6: Paired-Sample t-Test Results for Harnessed vs. Unharnessed Mean Scores

| Game | $\Delta$ Mean | $\%\Delta$ | $t\,(df = 9)$ | $p$ |
|---|---|---|---|---|
| Candy Crush | +217.50 | +224.8% | 4.22 | 0.0022 ** |
| Sokoban | +1.97 | +537.5% | 3.02 | 0.0144 * |
| 2048 | +17.81 | +22.4% | 2.36 | 0.0424 * |
| Ace Attorney | +3.20 | +123.1% | 2.36 | 0.0427 * |
| Tetris | +5.60 | +27.1% | 2.27 | 0.0490 * |
| Super Mario Bros. | +289.10 | +19.3% | 1.45 | 0.1806 |

$* \, p < 0.05$, $** \, p < 0.01$

$$\text{CV} = \frac{s}{\overline{X}} \times 100\% \qquad (3)$$

for each model–game–condition (Random, With Harness, Without Harness) to measure relative dispersion around the mean. Sokoban and Ace Attorney are excluded (zero random variance), and models with only a single run (o1, o3) are omitted. Table 7 lists the rounded CV values (in %) for the remaining ten models across four games.

Across the four games, harnessed runs yield lower CV than random play in 8/8 cases for 2048 (100.0%), 6/8 for Candy Crush (75.0%), 6/8 for Super Mario Bros. (75.0%), and 5/8 for Tetris (62.5%). Comparing harnessed to unharnessed, CV is lower under harness in 6/8 for 2048 (75.0%), 8/8 for Candy Crush (100.0%), 4/8 for Super Mario Bros. (50.0%), and 4/8 for Tetris (50.0%). Overall, out of 32 valid model–game pairs, 25 (78.1%) have smaller CV under harness versus random, and 22 (68.8%) have smaller CV under harness versus unharnessed. These results indicate that the harness not only elevates mean performance but also lowers the coefficient of variation—i.e. reduces relative score dispersion—which yields more stable, reliable assessments of model ability.

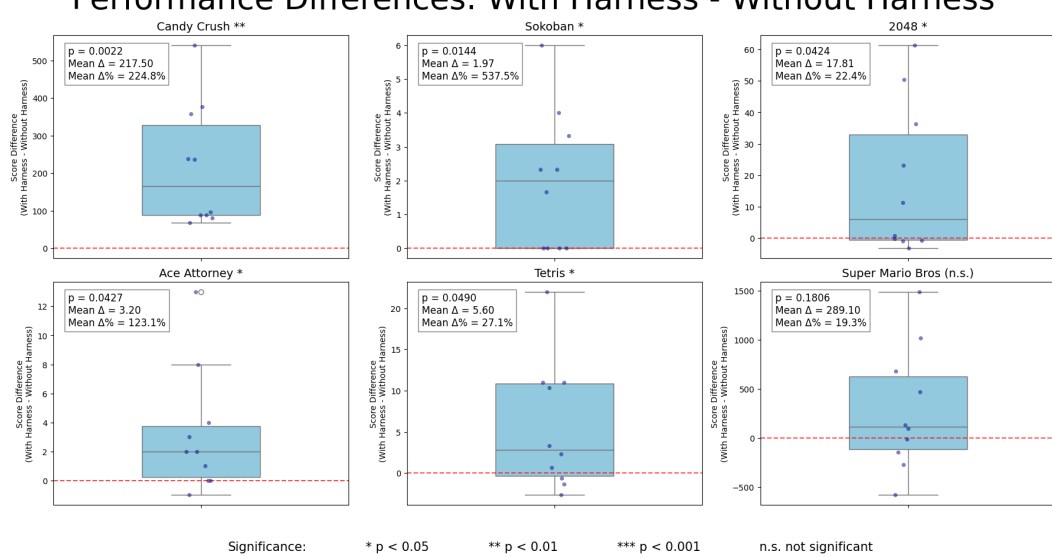

Figure 4: Distribution of paired score improvements (Harness – No Harness) across ten models for each game.

Table 7: Coefficient of Variation (CV %) by Model, Condition, and Game

| Model | Condition | 2048 | Candy Crush | Super Mario Bros | Tetris |
|---|---|---|---|---|---|
| Random | Random | 7.798 | 44.183 | 41.995 | 17.535 |
| claude-3-5-sonnet-20241022 | With Harness | 5.351 | 50.372 | 38.184 | 7.873 |
| | Without Harness | 28.377 | 106.371 | 1.409 | 20.405 |
| claude-3-7-sonnet-20250219 (thinking) | With Harness | 2.766 | 11.090 | 46.546 | 14.139 |
| | Without Harness | 6.285 | 54.694 | 11.341 | 0.000 |
| gemini-2.5-flash-preview-04-17 (thinking) | With Harness | 4.962 | 19.581 | 17.210 | 19.681 |
| | Without Harness | 3.172 | 36.978 | 17.034 | 24.119 |
| gemini-2.5-pro-preview-05-06 (thinking) | With Harness | 4.993 | 1.635 | 13.573 | 2.474 |
| | Without Harness | 3.245 | 36.599 | 43.228 | 24.771 |
| gpt-4.1-2025-04-14 | With Harness | 6.605 | 15.763 | 83.636 | 4.225 |
| | Without Harness | 17.945 | 119.047 | 51.144 | 13.323 |
| gpt-4o-2024-11-20 | With Harness | 3.266 | 36.249 | 25.797 | 25.754 |
| | Without Harness | 21.545 | 92.509 | 63.796 | 14.193 |
| llama-4-maverick-17b-128e-instruct-fp8 | With Harness | 3.556 | 44.459 | 37.835 | 14.783 |
| | Without Harness | 26.470 | 128.055 | 58.857 | 9.897 |
| o4-mini-2025-04-16 | With Harness | 4.030 | 40.634 | 11.116 | 33.572 |
| | Without Harness | 29.880 | 44.866 | 13.210 | 24.037 |

## B.4 COST ANALYSIS AND SCALABILITY

Agentic evaluations are token-billed; major providers charge per input/output token, so longer contexts and extra tool calls increase spend.[1] Running LMGame-Bench remains more controllable than full-scale software-agent benchmarks (e.g., SWE-bench) under comparable model sets and query counts. Tables 8–9 summarize order-of-magnitude totals and per-step estimates from cached logs. Overall, *with-harness* costs exceed *no-harness* due to (i) longer contexts from appended trajectories and (ii) an additional reflection call in the memory module; optional perception may add a vision-to-text call.

*Scalability levers.* Strategies include bounding or summarizing trajectories, triggering reflection every $k$ steps or after invalid moves, caching and de-duplicating perception frames, routing perception

---

[1] https://openai.com/api/pricing; https://ai.google.dev/gemini-api/docs/billing; https://www.anthropic.com/news/1m-context

Table 8: Order-of-magnitude benchmark costs (illustrative), from estimated queries and average tokens per query.

| Benchmark | Est. q | Model | Avg. in | Avg. out | Est. $ |
|---|---|---|---|---|---|
| SWE-bench | 2,294 | o3 | 100k | 20k | $830 |
| SWE-bench | 2,294 | claude-sonnet-4-0 | 100k | 20k | $1,380 |
| SWE-bench | 2,294 | gemini-2.5-pro | 100k | 20k | $750 |
| LMGame-Bench | 2,000 | o3 | 20k | 10k | $80 |
| LMGame-Bench | 2,000 | claude-sonnet-4-0 | 20k | 10k | $130 |
| LMGame-Bench | 2,000 | gemini-2.5-pro | 20k | 10k | $60 |

Table 9: Estimated *per-step* costs (USD) from cached logs for two games under four harness settings.

| Model | Game | No harness | Harness (perc) | Harness (mem) | Harness (both) |
|---|---|---|---|---|---|
| claude-3-7-sonnet-latest | candy_crush | 0.011 | 0.021 | 0.047 | 0.048 |
| claude-3-7-sonnet-latest | twenty_forty_eight | 0.015 | 0.026 | 0.029 | 0.031 |
| gemini-2.5-pro | candy_crush | 0.005 | 0.009 | 0.019 | 0.020 |
| gemini-2.5-pro | twenty_forty_eight | 0.016 | 0.019 | 0.020 | 0.021 |
| gpt-4.1 | candy_crush | 0.008 | 0.014 | 0.029 | 0.025 |
| gpt-4.1 | twenty_forty_eight | 0.008 | 0.014 | 0.016 | 0.016 |
| o1 | candy_crush | 0.055 | 0.105 | 0.212 | 0.218 |
| o1 | twenty_forty_eight | 0.054 | 0.102 | 0.114 | 0.121 |

and memory tasks to smaller models while reserving frontier models for action decisions, batching evaluations and reusing prompts across seeds, and early-stopping low-performing runs using interim thresholds.

## C   QUALITATIVE ANALYSIS

Here we give examples on models' performances to support the qualitative analysis in Section 4.2, providing insights into our harness design and discussing representative failures.

### C.1   IMAGE PERCEPTION

#### C.1.1   BOARD GAMES

Figure 5 provides examples of image perception failures on Gemini 2.5 Pro with thinking enabled. Note that similar failures have been observed from other leading LLMs, including GPT-4o and Claude 3.7 Sonnet. These examples show that even advanced multimodal LLMs struggle with basic image perception of game boards. Such perceptual skills are foundational for higher-level reasoning on image-based grid games, yet they remain unreliable in practice.

- Example 1 (Tetris). The task is straightforward: given a screenshot of a Tetris board, the model is asked to convert the grid into a 2D text representation. Despite the simplicity of the task, the model produces an incorrect transcription of the board layout. When further prompted to count the number of occupied blocks in the bottom row, the model repeatedly miscalculates the answer, even though the correct count can be visually verified from the image.

- Example 2 (Sokoban). The model is asked to produce a 1D text representation of a Sokoban board, mapping coordinates to object categories such as wall, player, goal, box, or empty. While the mapping rules are explicitly provided, the model incorrectly assigns most non-wall positions.

- Example 3 (2048). The model is asked to transcribe a 2048 board from an image into a textual grid. Unlike in Tetris and Sokoban, the model succeeds in reproducing the board layout. This relative success is likely due to the board's small size ($4\times4$) and the visually

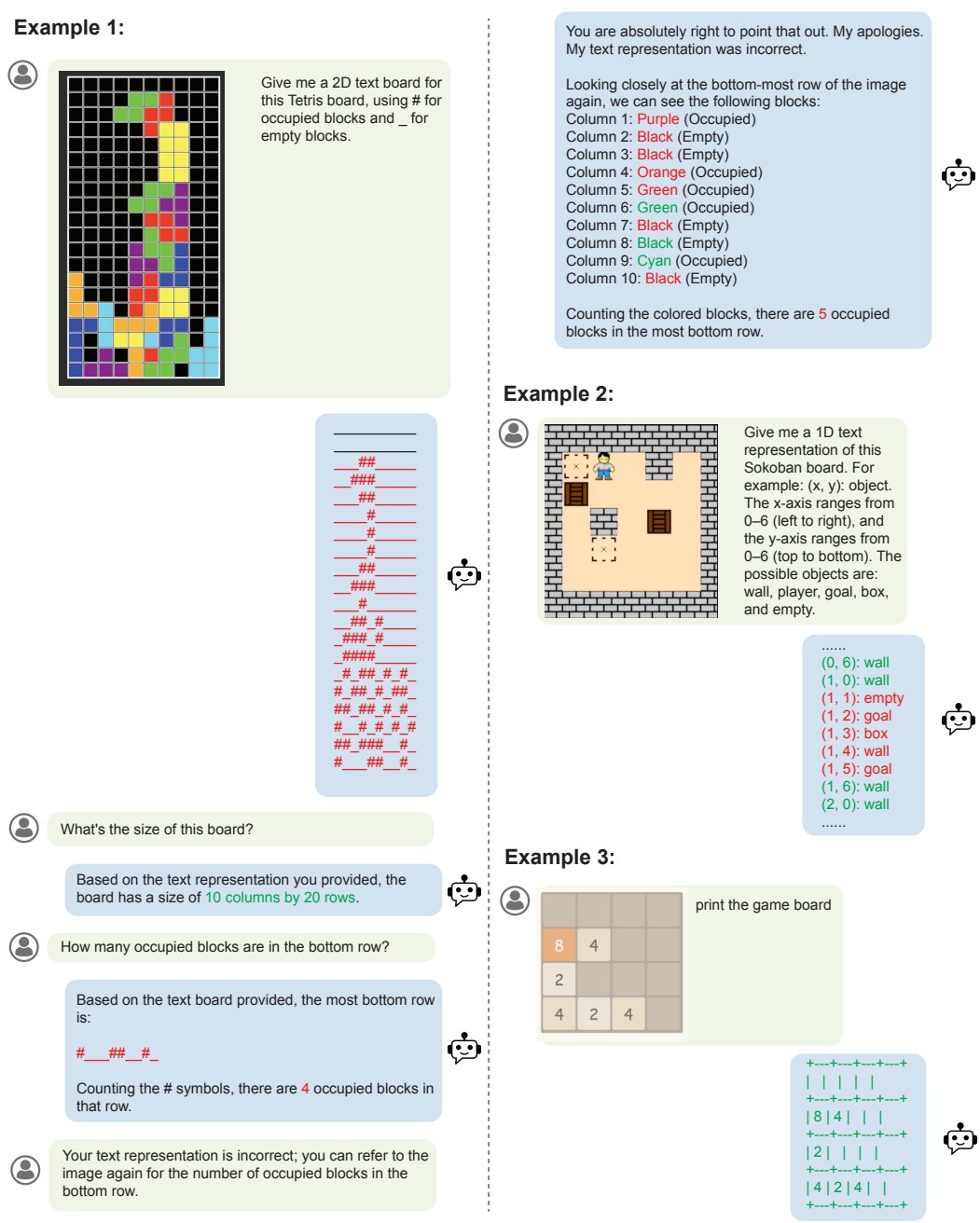

Figure 5: Examples of image perception tasks for Gemini-2.5-pro. Correct predictions are shown in green, and incorrect ones in red. In Tetris (example 1) and Sokoban (example 2), the model fails to reconstruct the game board from images. In contrast, for 2048 (example 3), the model succeeds, which is likely due to the board's small size (4×4) and the visually distinct numerical tiles.

distinct numerical tiles, which make the perception task much simpler than in larger or more complex games.

## C.1.2 EFFECTIVENESS OF PERCEPTION MODULE

Considering that perception is often the bottleneck of gameplay, we measure how the perception module boosts performance across different games and models. Using the data from Table 4, we first

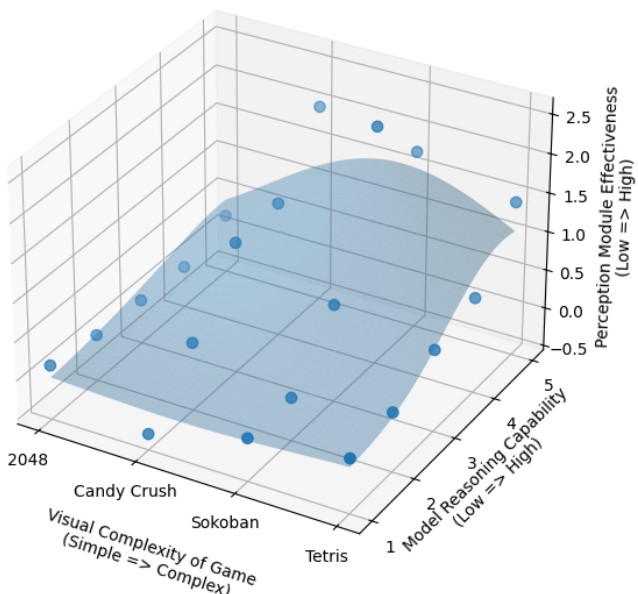

Figure 6: The effectiveness of perception module is correlated with the visual complexity of game and model reasoning capability.

assign each model a reasoning-capability rank and each game a visual-complexity rank. We then quantify the effectiveness of the perception module as the relative improvement over the zero-shot baseline. We then plot the perception-module effectiveness, the game's visual complexity, and the model's reasoning capability in a 3D space to analyze their correlations (Figure 6). Specially, we uses the following design choices:

- *Visual complexity of the game* is approximated by the average size of its board representation, giving the ordering: Tetris > Sokoban > Candy Crush > 2048.
- *Model reasoning capability* is measured by the model's full-harness performance, yielding the ranking: `o4-mini-2025-04-16` > `gemini-2.5-pro-03-25` > `claude-3-7-sonnet` > `gpt-4o` > `claude-3-5-sonnet`.
- *Perception module effectiveness* is measured using the relative improvement between the perception-enabled and perception-disabled settings. We then fit a Gaussian Process Regression surface over these rankings to capture the trend.

The resulting curve in Figure 6 shows that perception module is the most effective in visually demanding games (i.e., Sokoban, Tetris, and Candy Crush), where accurate state extraction is a prerequisite for effective reasoning. Improvements are also more pronounced for stronger reasoning models: once perception is the bottleneck, supplying an accurate scene description allows these models to fully leverage their reasoning ability. Overall, the perception module is more useful when the game is visually complex and the model's strong reasoning capability is limited by imperfect perception.

## C.2 REFLECTION ON FAILURES

A recurring weakness across multiple games is the tendency of models to fall into unproductive loops, repeatedly proposing ineffective or invalid actions without learning from past mistakes. This behavior is particularly pronounced in non-reasoning models that lack explicit memory or reflection mechanisms.

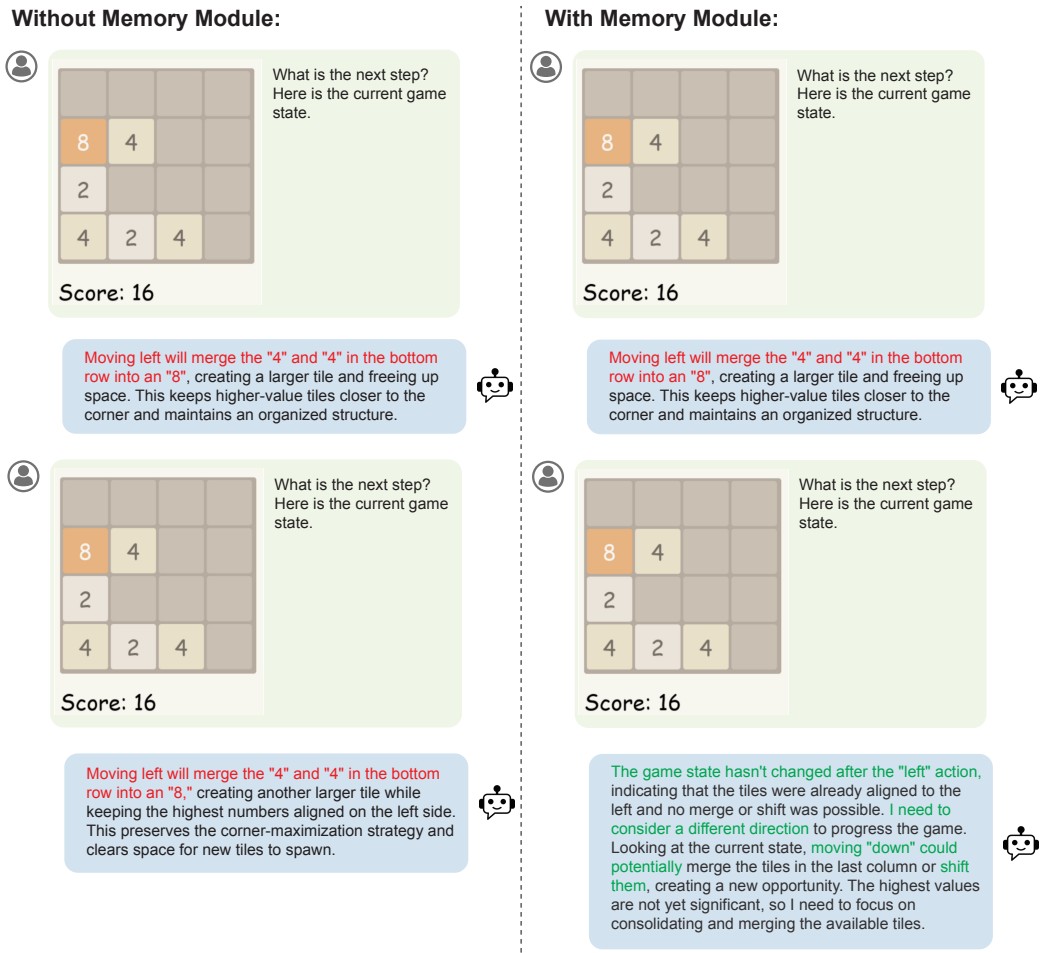

Figure 7: Examples of reflection from failures for Llama-4-Maverick. Correct predictions are shown in green, and incorrect ones in red. Without a memory module to store past history and enable self-reflection, the model cannot recognize its own failures; with memory support, it can. This effect is most significant for non-reasoning models.

Figure 7 shows in 2048, when asked for the next move, the model may continuously suggest the same direction (e.g., "left"), even after several failed attempts that yield no new merges. The model produces fluent but redundant justifications, such as "merging the 4 and 4 in the bottom row," even though no such merge is possible in the current board state. This repetition highlights an inability to track the consequences of previous actions and to update its internal state. Incorporating a memory module mitigates this problem by allowing models to record past actions and reflect on their outcomes, enabling them to detect and avoid repeated failures. In contrast, reasoning models can often recognize failures directly from a single game state without requiring additional memory.

## C.3  REASONING TRACE OF REFLECTION

Here we present case studies that demonstrate the reasoning traces of the latest models between GPT-4o and o3 during their 2048 gameplay. GPT-4o's reflections tend to focus on immediate effect of each move ("up increased empty spaces but produced no merges"), whereas o3's reflections unpack exactly which tiles merged, how merge ladders are forming, and even quantify changes in empty-cell counts("two 2's became a 4 and two 16's became 32's, opening an extra cell"). Importantly, an interactive game environment isn't a one-off Q&A but a multi-turn dialogue with a constantly shifting game board - each action affects next decision. o3 better links its fine-grained observations to concrete

next steps, integrating short-term move outcomes into its long-term corner-dominance strategy. The success of local decisions is crucial to achieving long-term goals in multi-turn interactive games.

**Highlight Legend:**

Red text: the model's *immediate* game-state reflection.

Yellow text: o3's *immediate* game-state reflection.

Green text: the model's *future move* suggestion.

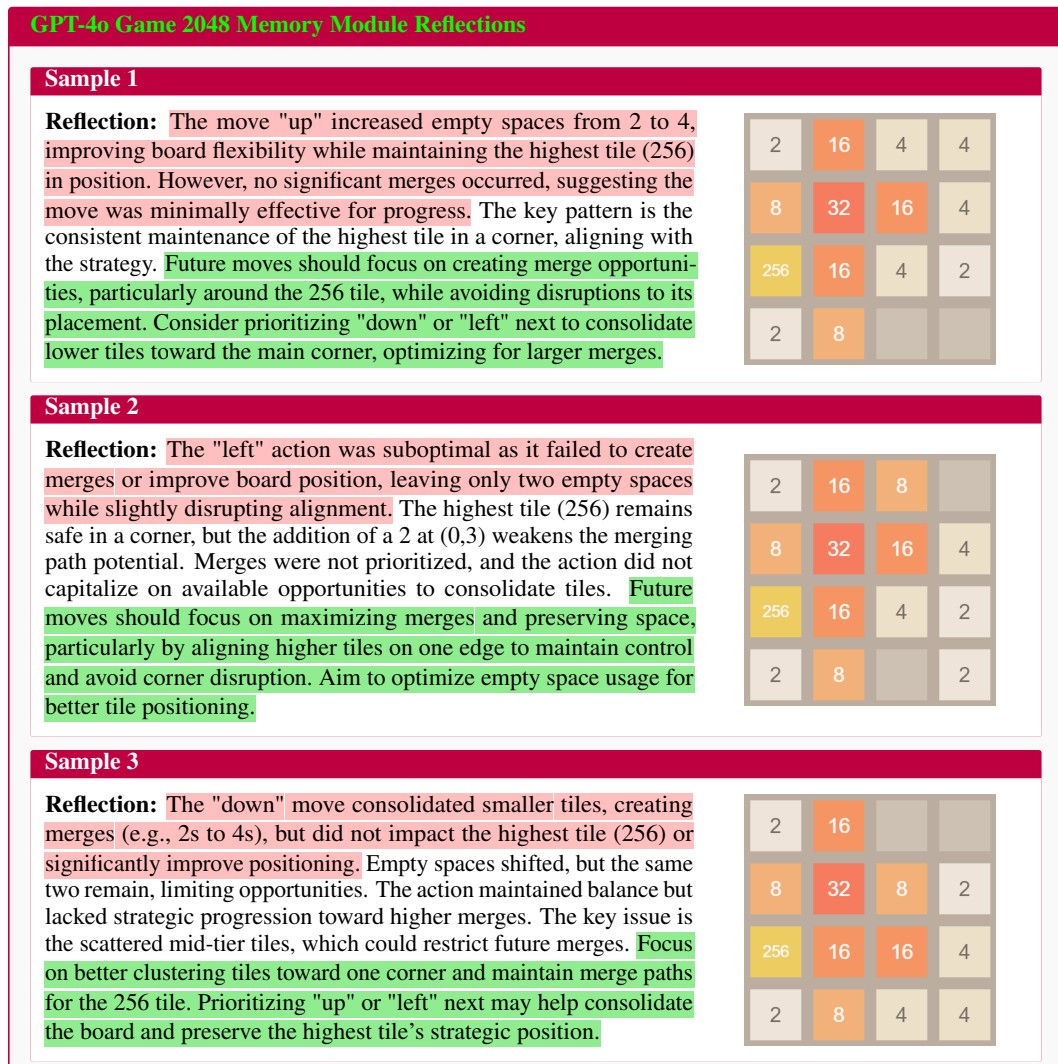

**GPT-4o Game 2048 Memory Module Reflections**

**Sample 1**

**Reflection:** The move "up" increased empty spaces from 2 to 4, improving board flexibility while maintaining the highest tile (256) in position. However, no significant merges occurred, suggesting the move was minimally effective for progress. The key pattern is the consistent maintenance of the highest tile in a corner, aligning with the strategy. Future moves should focus on creating merge opportunities, particularly around the 256 tile, while avoiding disruptions to its placement. Consider prioritizing "down" or "left" next to consolidate lower tiles toward the main corner, optimizing for larger merges.

**Sample 2**

**Reflection:** The "left" action was suboptimal as it failed to create merges or improve board position, leaving only two empty spaces while slightly disrupting alignment. The highest tile (256) remains safe in a corner, but the addition of a 2 at (0,3) weakens the merging path potential. Merges were not prioritized, and the action did not capitalize on available opportunities to consolidate tiles. Future moves should focus on maximizing merges and preserving space, particularly by aligning higher tiles on one edge to maintain control and avoid corner disruption. Aim to optimize empty space usage for better tile positioning.

**Sample 3**

**Reflection:** The "down" move consolidated smaller tiles, creating merges (e.g., 2s to 4s), but did not impact the highest tile (256) or significantly improve positioning. Empty spaces shifted, but the same two remain, limiting opportunities. The action maintained balance but lacked strategic progression toward higher merges. The key issue is the scattered mid-tier tiles, which could restrict future merges. Focus on better clustering tiles toward one corner and maintain merge paths for the 256 tile. Prioritizing "up" or "left" next may help consolidate the board and preserve the highest tile's strategic position.

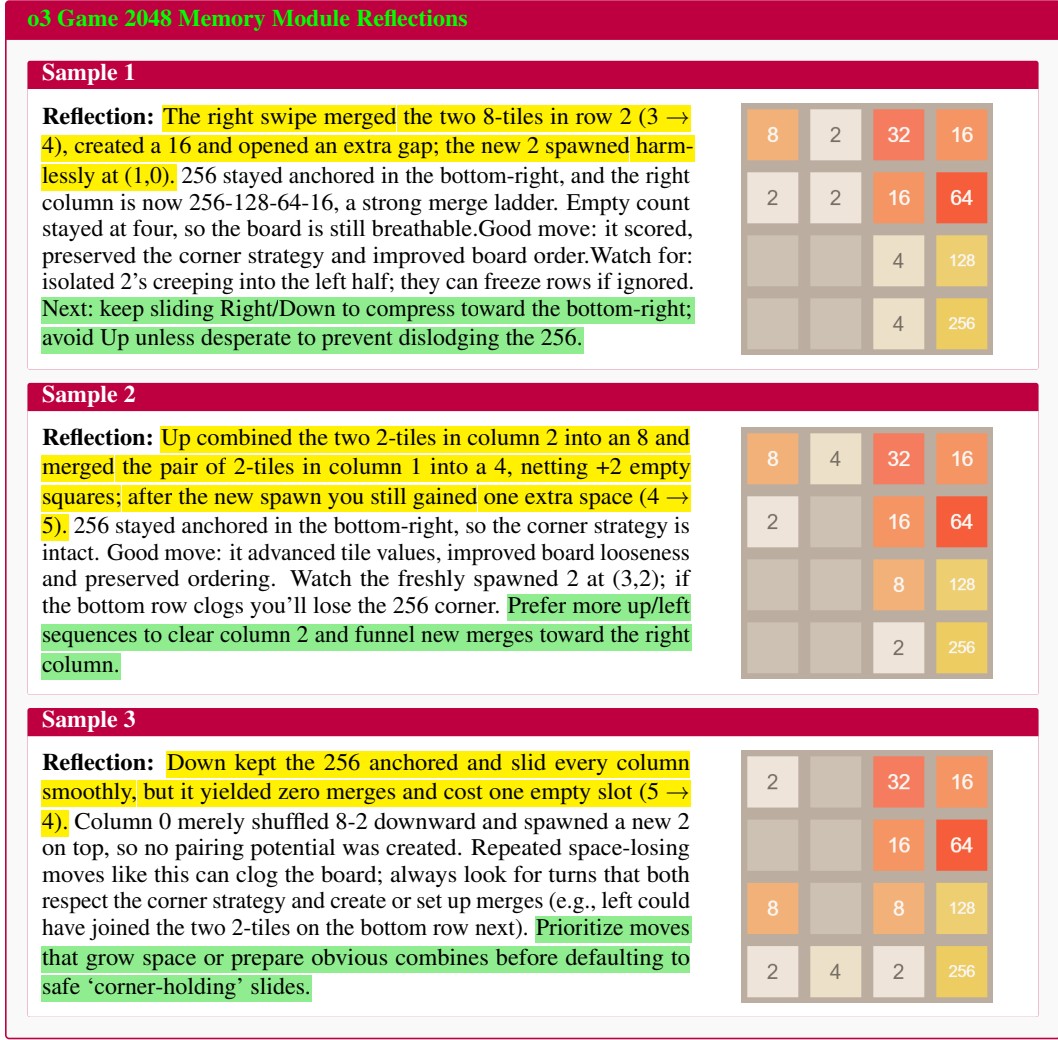

**o3 Game 2048 Memory Module Reflections**

**Sample 1**

**Reflection:** The right swipe merged the two 8-tiles in row 2 (3 → 4), created a 16 and opened an extra gap; the new 2 spawned harmlessly at (1,0). 256 stayed anchored in the bottom-right, and the right column is now 256-128-64-16, a strong merge ladder. Empty count stayed at four, so the board is still breathable.Good move: it scored, preserved the corner strategy and improved board order.Watch for: isolated 2's creeping into the left half; they can freeze rows if ignored. Next: keep sliding Right/Down to compress toward the bottom-right; avoid Up unless desperate to prevent dislodging the 256.

**Sample 2**

**Reflection:** Up combined the two 2-tiles in column 2 into an 8 and merged the pair of 2-tiles in column 1 into a 4, netting +2 empty squares; after the new spawn you still gained one extra space (4 → 5). 256 stayed anchored in the bottom-right, so the corner strategy is intact. Good move: it advanced tile values, improved board looseness and preserved ordering. Watch the freshly spawned 2 at (3,2); if the bottom row clogs you'll lose the 256 corner. Prefer more up/left sequences to clear column 2 and funnel new merges toward the right column.

**Sample 3**

**Reflection:** Down kept the 256 anchored and slid every column smoothly, but it yielded zero merges and cost one empty slot (5 → 4). Column 0 merely shuffled 8-2 downward and spawned a new 2 on top, so no pairing potential was created. Repeated space-losing moves like this can clog the board; always look for turns that both respect the corner strategy and create or set up merges (e.g., left could have joined the two 2-tiles on the bottom row next). Prioritize moves that grow space or prepare obvious combines before defaulting to safe 'corner-holding' slides.

## C.4 Spatiotemporal Reasoning

Another recurring weakness lies in spatiotemporal reasoning, which is the ability to coordinate actions with their duration in real-time gameplay. In Super Mario Bros, correct behavior often requires not only choosing the right action (e.g., jump) but also sustaining it for an appropriate number of frames. Models frequently misjudge either distance or timing, leading to repeated failures.

- **Example 1 (Tall Pipe).** When Mario stands one tile away from a tall pipe, the model correctly chooses to jump but commits to too few frames (e.g., 13 instead of the 30 required). As a result, Mario clips into the side of the pipe and fails to clear it.
- **Example 2 (Gap Crossing).** When approaching a gap, models often initiate their jump too early, causing Mario to lose forward momentum and fall short of the landing platform.

Figure 8 illustrates these two cases. Together, they show that current models lack robust mechanisms for integrating spatial planning with temporal duration. Unlike human players, who quickly adjust jump timing after failed attempts, LLMs often repeat the same miscalibrated actions without self-correction, underscoring a fundamental challenge in real-time decision-making.

## C.5 Long Context

Figure 9 shows a representative long-context failure in Ace Attorney. During cross-examination, the witness claims the time was "coming from the television" at around 1:00 PM. This directly contradicts E2, which states that electricity to Ms. Stone's building was out from noon to 6 PM,

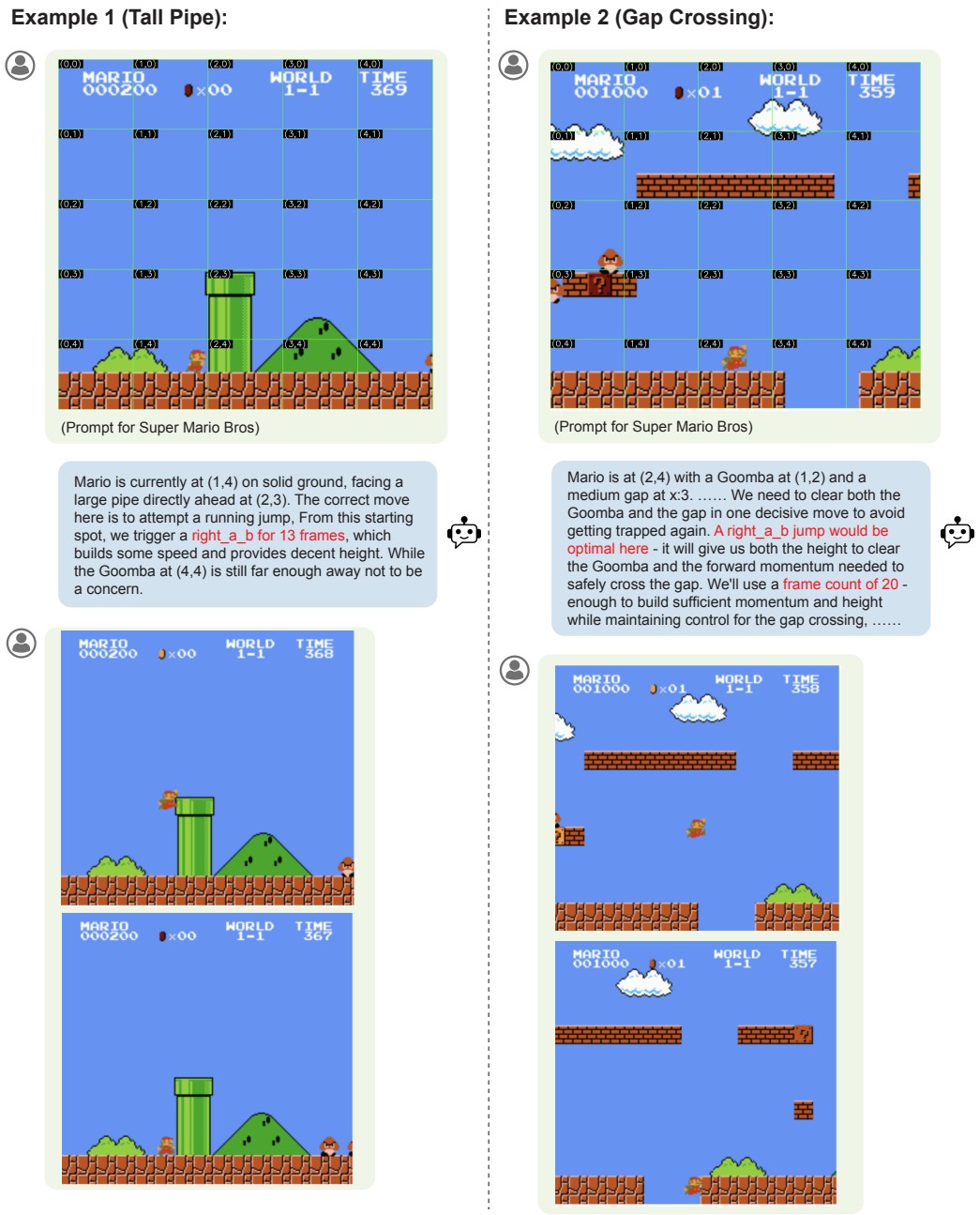

Figure 8: Failures in spatiotemporal reasoning for Claude-3-5-Sonnet playing Super Mario Bros. Example 1 shows insufficient jump duration when approaching a tall pipe, and Example 2 shows premature jump initiation when crossing a gap.

**Example 1:**

What is the next move?Here is the text history:
Evidence List:
E1: No one would believe I was a defense attorney if I didn't carry this.
E2: **Electricity to Ms. Stone's building was out from noon to 6PM on the day of the crime**.
E3: The victim apparently arrived home from Paris on 7/30, the day before the murder…….

Dialogue History (older -> newer):
witness 1: I was going door-to-door, selling subscriptions when I saw a man fleeing an apartment.
I thought he must be in a hurry because he left the door half-open behind him.
Thinking it strange, I looked inside the apartment.
Then I saw her lying there... A woman... not moving... dead!
I quailed in fright and found myself unable to go inside.
I thought to call the police immediately!

…….(~8k tokens with context from previous cross examination)

You see, when I found the body, I heard the time.
There was a voice saying the time... **It was probably coming from the television**.
Oh, but it was three hours off, wasn't it?
I guess the victim must have been watching a video of a taped program!
**That's why I thought it was 1:00 PM!**
Terribly sorry about the misunderstanding...

judge 1: Mr. Wright, you may cross-examine the witness.
mentor 1: Wright!
mentor 1: You know what to do!
lawyer 1: I've got this one."

Agent Action: "Continue Next Statement"
Thought: "Cause: We're in Cross-Examination and the current statement is 'coming from the television.' Dialogue Continuation is No, so I should act on this statement. After reviewing the current evidence list, there is no direct contradiction here—the statement does not conflict with the available evidence. Effect: Since no conflict is found, I will continue with the cross-examination without presenting evidence. Reflection: This aligns with the strategy of carefully checking each claim but only acting when a clear contradiction is established. Selected_Option: None
Selected_Evidence: None
Presented_Evidence: None
self_evaluation: Yes

**Example 2:**

[Delete the 8k-token context in Example 1, the other parts remain the same.]

……
E2 states that there was a blackout in Ms. Stone's building from noon to 6PM on the day of the crime. This is crucial because if there was no electricity, televisions wouldn't be working unless they were battery-powered.
……

Figure 9: Example of long-context reasoning from GPT-4o playing Ace Attorney. The model evaluates a witness statement, checks the evidence list, but incorrectly decides that no contradiction is present. After removing the long context, the model is able to identify the contradiction.

making a television broadcast impossible. However, instead of presenting E2, the model incorrectly concludes that no contradiction exists and continues to the next statement. This error illustrates a recurring weakness: although the necessary evidence is present in the context window, models often fail to retrieve and apply it to the current claim, leading to missed contradictions in long, text-heavy reasoning tasks.

To further examine long-context reasoning failures, we evaluated GPT-4o, GPT-4.1, and Gemini 2.5 Flash on three evidence checkpoints across two game levels. Each model was tested under two settings: *with long context* (which retains older, less relevant dialogue) and *without long context* (keeping only the most recent statements). All inputs were textual to avoid perception-related confounds.

| Model | Context Type | L1-C3 | L2-C2 | L2-C4 |
|---|---|---|---|---|
| GPT-4o | With Long Context | no | no | no |
| | Without Long Context | yes | no | no |
| GPT-4.1 | With Long Context | no | yes | no |
| | Without Long Context | no | yes | yes |
| Gemini 2.5 Flash | With Long Context | no | no | no |
| | Without Long Context | no | yes | no |

Table 10: Model accuracy on three evidence checkpoints under long-context and short-context conditions. All evaluations use textual descriptions only.

Across all three models, we observe a consistent degradation in performance when long context is included. GPT-4o fails all checkpoints under the long-context setting but recovers one (L1-C3) once irrelevant prior dialogue is removed. Gemini 2.5 Flash exhibits a similar pattern, correctly identifying only L2-C2 in the short-context setting. GPT-4.1 shows a different sensitivity profile: it consistently fails L1-C3 regardless of context length but performs substantially better in Level 2 under short context, correctly solving both L2-C2 and L2-C4.

These results indicate that even modest amounts of earlier, semantically weak dialogue can introduce noise that disrupts evidence retrieval and logical application. The accuracy drop under long-context settings suggests that current LLMs struggle to maintain precise reasoning chains when older conversational history remains in the prompt, making long-context interference a persistent failure mode in text-heavy reasoning tasks such as Ace Attorney.

## D  DATA CONTAMINATION STUDY

We evaluate two types of potential data contamination in LMGame-Bench : **vision-level** and **text-level**. Our goal is to determine whether pretrained LLMs rely on memorized assets instead of real-time reasoning.

### D.1  VISION-LEVEL CONTAMINATION: SUPER MARIO BROS

**Setup.** We extracted the first ten RGB frames from SMB Level 1-1 and randomly shuffled their order. Each model was then prompted to reconstruct the original temporal sequence 15 times. Reconstruction quality was measured by pairwise frame-order accuracy, Kendall's $\tau$ rank coefficient (Kendall, 1938), and Rank-Biased Overlap (RBO) (Webber et al., 2010). Finally, we computed the Pearson's and Spearman's correlation coefficients between these alignment metrics and the overall performance rank of each model.

**Models tested:** Claude-3.5-Sonnet, Claude-3.7-Sonnet-Thinking, Gemini-2.5-pro-Preview, o4-mini, o3, LLaMA-4-Maverick.

**Results.** The pairwise accuracy remains relatively low overall, with the highest accuracy reaching only around 30%, as shown in Table 11. Notably, both the beginning and final positions exhibit relatively high accuracy in terms of ordering, whereas the middle positions perform significantly worse, as illustrated in Fig. 10. Beyond pairwise accuracy, we compute Kendall's rank correlation coefficient and Rank-Biased Overlap (RBO) to evaluate how well each model's predicted frame

order aligns with the ground-truth temporal sequence. As shown in Fig. 11, all models exhibit positive correlation, although only Gemini-2.5-pro-preview, o3, and o4-mini achieve moderate agreement with ground truth, while the remaining models show weak alignment. To further quantify alignment strength, we normalize each metric by computing the percentage of the perfect score, using the formula $(value - random)/(perfect - random)$, which reveals similar ranking patterns for both Kendall's $\tau$ and RBO. To test whether alignment quality is predictive of general model performance, we compute both Pearson and Spearman correlations between the alignment metrics and the performance ranks of the models. Kendall's $\tau$ shows a moderate negative Pearson correlation with performance rank ($r = -0.7089$, $p = 0.1148$, testing the null hypothesis of no linear association) and a moderate negative Spearman correlation ($\rho = -0.5429$, $p = 0.2657$, testing the null hypothesis of no monotonic association). RBO shows a weaker Pearson correlation ($r = -0.3847$, $p = 0.4515$, testing the null hypothesis of no linear association) and a moderate negative Spearman correlation ($\rho = -0.6571$, $p = 0.1562$, testing the null hypothesis of no monotonic association). Kendall's $\tau$ and RBO remain highly correlated (Pearson $r = 0.8772$, $p = 0.0217$, testing the null hypothesis of no linear association). Despite this internal consistency, the lack of statistically significant correlation with model performance rank suggests that visual sequence alignment—used here as a proxy for vision-based data contamination—does not appear to be a major factor in determining current model performance rankings.

Table 11: Evaluation metrics (Average Pairwise Accuracy, Kendall's $\tau$, and RBO) on the frame ordering task for Super Mario Bros level 1-1. Higher values indicate better reconstruction of the correct temporal sequence from shuffled RGB inputs.

| Model | Accuracy | Kendall's $\tau$ | RBO |
|---|---|---|---|
| o3 | 0.307 | 0.449 | 0.498 |
| gemini-2.5-pro-preview (thinking) | 0.300 | 0.458 | 0.489 |
| o4-mini | 0.233 | 0.324 | 0.463 |
| claude-3-7-sonnet | 0.147 | 0.044 | 0.422 |
| claude-3-5-sonnet | 0.180 | 0.099 | 0.418 |
| llama-4-maverick | 0.100 | 0.019 | 0.324 |
| Perfect | 1.000 | 1.000 | 0.651 |
| Random | 0.090 | -0.037 | 0.299 |

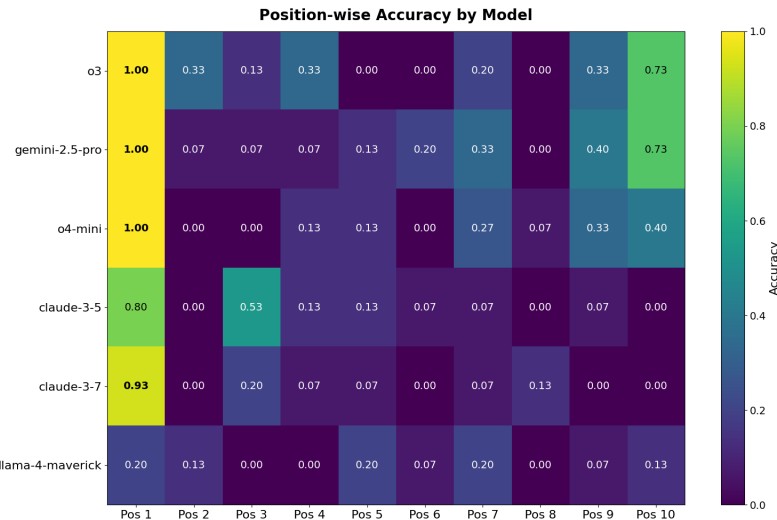

Figure 10: Position-wise reconstruction accuracy for the shuffled Super Mario Bros level 1-1 frames. Lighter cells denote higher accuracy; only the first and last positions remain relatively high accuracy

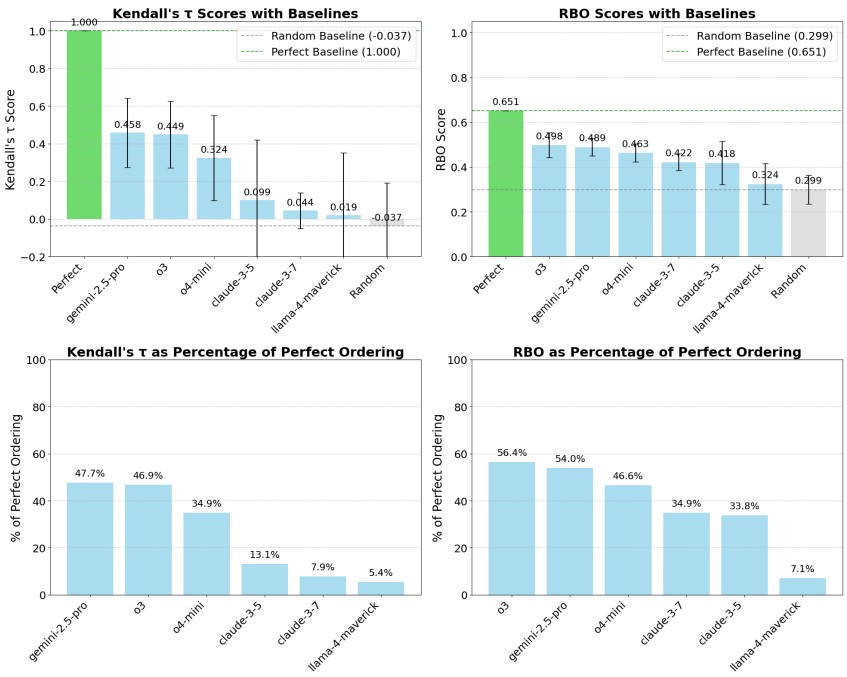

Figure 11: Kendall's $\tau$ and RBO scores for each model on the frame ordering task. Higher values indicate stronger alignment between predicted and ground-truth frame sequences. Only Gemini-2.5-pro-preview, o3, and o4-mini achieve moderate agreement.

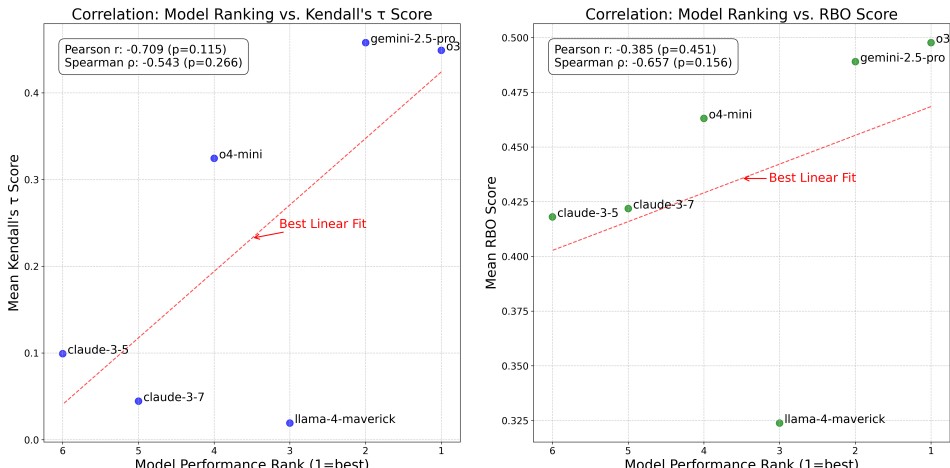

Figure 12: Pearson and spearman correlation between model performance ranks and their alignment scores (Kendall's $\tau$ and RBO). Although a negative trend is observed, the correlation is not statistically significant.

## D.2 TEXT-LEVEL CONTAMINATION: ACE ATTORNEY

**Setup.** We test whether models reproduce scripted lines from the first two publicly available cases of *Ace Attorney*. Each case is split into an evidence list and a cross-examination script. We then prompt models to generate these sections and compute cosine similarity to the ground truth using Sentence-BERT embeddings.

**Models tested:** Claude-3.5-Sonnet, Claude-3.7-Sonnet-Thinking, Gemini-2.5-pro-Preview, o4-mini, o3, LLaMA-4-Maverick.

Table 12: Text-level similarity metrics and performance on *Ace Attorney*. Similarity is measured via Sentence-BERT cosine scores. Cross-case comparisons verify metric reliability.

| Model | 1st Turnabout | Sister Turnabout | Total Sim. | Rank | Game Score | Gen[F] vs GT[S] | Gen[S] vs GT[F] | Gen[F] vs Gen[S] |
|---|---|---|---|---|---|---|---|---|
| o3-2025-04-16 | 0.863 | 0.904 | 1.767 | 2 | 23 | 0.607 | 0.593 | 0.657 |
| gemini-2.5-pro-preview-05-06 (thinking) | 0.867 | 0.867 | 1.734 | 3 | 20 | 0.561 | 0.544 | 0.532 |
| claude-3-5-sonnet-20241022 | 0.845 | 0.816 | 1.660 | 6 | 6 | 0.609 | 0.538 | 0.672 |
| o1-2024-12-17 | 0.809 | 0.842 | 1.651 | 1 | 26 | 0.634 | 0.589 | 0.675 |
| grok-3-mini-beta | 0.782 | 0.833 | 1.614 | 5 | 7 | 0.646 | 0.550 | 0.707 |
| gpt-4.1-2025-04-14 | 0.755 | 0.812 | 1.567 | 7 | 6 | 0.697 | 0.591 | 0.787 |
| claude-3-7-sonnet-20250219(thinking) | 0.708 | 0.830 | 1.538 | 4 | 8 | 0.684 | 0.535 | 0.588 |
| gemini-2.5-flash-preview-04-17 (thinking) | 0.731 | 0.728 | 1.460 | 8 | 4 | 0.617 | 0.538 | 0.729 |
| gemini-2.0-flash-thinking-exp-1219 | 0.659 | 0.780 | 1.438 | 9 | 4 | 0.719 | 0.562 | 0.810 |
| LLaMA-4 Maverick | 0.491 | 0.734 | 1.224 | 13 | 0 | 0.594 | 0.535 | 0.772 |
| O4 Mini | 0.586 | 0.625 | 1.212 | 11 | 1 | 0.574 | 0.531 | 0.764 |

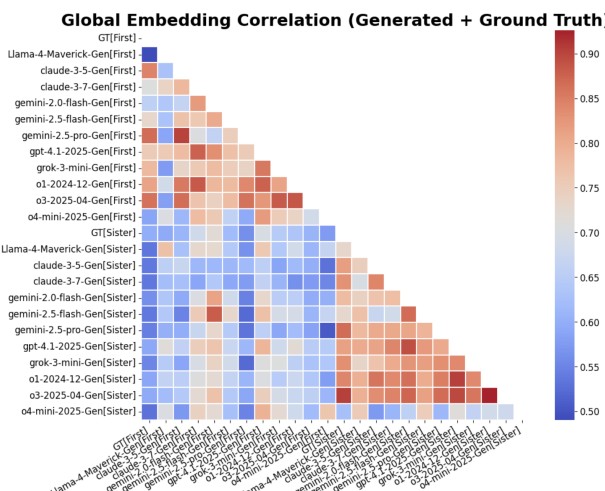

Figure 13: Global Embedding Correlation (Generated + Ground Truth) between model-generated texts across "First" and "Sister" turnabouts.

**Results.** Table 12 presents similarity scores for 11 models in both cases, alongside their in-game performance and cross-case comparisons. In particular:

- Models with higher similarity to the script tend to perform better in the game (e.g., o3, Gemini-2.5-Pro-Preview), suggesting possible memorization effects.

- Cross-case similarities (e.g., Gen[First] vs GT[Sister]) are consistently lower, demonstrating that the metric is sensitive to true alignment rather than generic language similarity.

- Self-similarity between generated case outputs (Gen[First] vs Gen[Sister]) is relatively high for some models, suggesting stylistic or template reuse.

- The Sentence-BERT cosine similarity between the ground-truth scripts for the two cases (1st Turnabout vs Sister Turnabout) is 0.599, which serves as a baseline for evaluating cross-case similarities.

To quantify the relationship between textual similarity and performance, we compute linear correlations between total similarity scores and both game score and leaderboard rank. As shown in Figure 14, the results hold consistently across both the full model set and the 6-model subset. In all cases, similarity is significantly correlated with better performance, confirming that models may benefit from memorized content.

**Mitigation.** To suppress memorized recall and enforce reasoning-based responses, we apply structured prompt interventions. These include: (1) explicitly instructing the model to forget prior knowledge of the game; (2) requiring detailed causal reasoning (cause, evidence, effect) for each action; (3) asking the model to self-evaluate whether its behavior was memory- or reasoning-driven; and (4) modifying the input by replacing all character and item names with neutral tokens (e.g., *"Lawyer 1"*, *"Evidence A"*), and paraphrasing both background context and key contradictions.

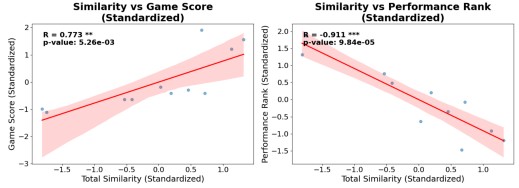 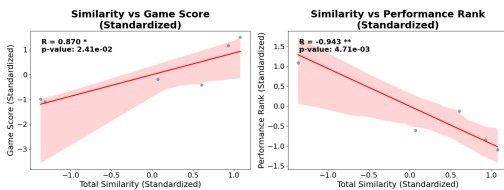

(a) Full model set. Left: similarity vs. game score ($r = 0.773$, $p = 0.005$). Right: similarity vs. leaderboard rank ($r = -0.911$, $p = 9.840 \times 10^{-5}$).

(b) 6-model subset. Left: similarity vs. game score ($r = 0.870$, $p = 0.024$). Right: similarity vs. leaderboard rank ($r = -0.943$, $p = 0.005$).

Figure 14: Linear correlations between script similarity and model performance across the full model set (left) and the 6-model subset used for cross-modality comparison (right). Shaded areas indicate 95% confidence intervals. In both settings, higher similarity is significantly associated with higher game scores and higher leaderboard position (i.e., lower rank number)

**Results.**

- Before mitigation (full model set), total similarity scores strongly correlate with leaderboard rank ($r = -0.773$, $p = 0.005$), indicating that models with higher overlap to the original script tend to perform better.

- After applying our prompt-based mitigation (name masking, paraphrasing, and reasoning enforcement), this correlation becomes statistically insignificant ($r = -0.700$, $p = 0.120$), suggesting a reduced reliance on memorization.

- Similarly, the correlation between similarity and game score drops to $r = 0.617$ ($p = 0.192$), further supporting the effectiveness of our intervention (see Figure 15a).

- Independent reasoning-based evaluations using o3 as an LLM judge remain predictive of post-mitigation performance, with a strong negative correlation to rank ($r = -0.850$, $p = 0.031$). (see Figure 15b).

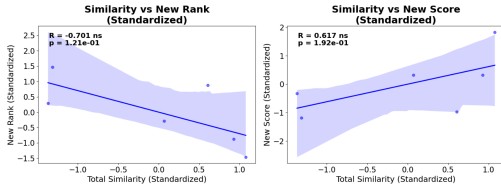 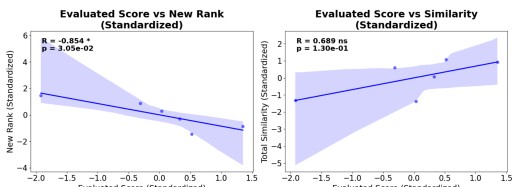

(a) Similarity vs. new rank (left) and new score (right). No statistically significant correlations remain after mitigation, suggesting similarity no longer explains model success.

(b) Evaluator score vs. new rank (left) and similarity (right). Evaluated reasoning correlates strongly with new rank ($r = -0.854$, $p = 0.031$) but not with similarity, suggesting post-mitigation rankings reflect reasoning.

Figure 15: Post-mitigation analysis. Left: model similarity no longer predicts rank or score. Right: LLM-as-Judge (o3) evaluations suggest model ranking is now aligned with reasoning quality, not memorized content.

**Conclusion.** Initial high performance in the *Ace Attorney* task is partially attributable to memorization of publicly available scripts. Prior to mitigation, script similarity was significantly predictive of performance. After structured prompt-based mitigation—including name masking, paraphrasing, and enforced reasoning—this correlation disappears. Post-intervention rankings instead align with reasoning quality, as verified by an independent evaluator (o3).

### D.3 EXTRA TEXT-LEVEL CONTAMINATION: SUPER MARIO BROS

**Setup.** To assess text-level contamination for *Super Mario Bros.*, we extract a detailed level layout description of World 1-1 from MarioWiki. This description is segmented into a sequence of temporally ordered segments, denoted by $[T1]$ through $[T18]$, representing key points in gameplay. We prompt models to generate layout descriptions aligned with these time points and compare them to the ground truth.

**Models tested:** o4-mini, o3, Claude-3.7-Sonnet, Claude-3.5-Sonnet, LLaMA-4-Maverick.

**Result.** We use Sentence-BERT embeddings to calculate pairwise cosine similarity between generated texts and the 1-1 ground truth. Notably, the similarity between the generated descriptions and the true 1-1 layout is consistently *lower* than the similarity between two distinct ground truth descriptions (World 1-1 vs. 1-2). This suggests that models do not rely on memorized textual layout from training corpora and instead generalize loosely or hallucinate. See Table 13 for quantitative comparisons. A representative side-by-side sample comparison between O3 and ground truth is shown in Figure 16.

Table 13: Text-level similarity metrics for *Super Mario Bros.* 1-1 layout descriptions. Similarity is measured using Sentence-BERT cosine scores. The last column shows the ground truth similarity between level 1-1 and 1-2 as a baseline.

| Model | Gen[1-1] vs GT[1-1] | Gen[1-2] vs GT[1-2] | Gen[1-1] vs Gen[1-2] | Gen[1-1] vs GT[1-2] | Gen[1-2] vs GT[1-1] | GT[1-1] vs GT[1-2] |
|---|---|---|---|---|---|---|
| o3 (2025-04-16) | 0.753076 | 0.813358 | 0.882990 | 0.749037 | 0.800998 | 0.844817 |
| o4-Mini (2025-04-16) | 0.708403 | 0.725378 | 0.852628 | 0.707597 | 0.799113 | 0.844817 |
| Claude-3.7 Sonnet (2025-02-19) | 0.679185 | 0.627961 | 0.821135 | 0.560105 | 0.750460 | 0.844817 |
| Claude-3.5 Sonnet (2024-10-22) | 0.686953 | 0.604426 | 0.795933 | 0.587022 | 0.701139 | 0.844817 |
| LLaMA-4 Maverick FP8 | 0.643627 | 0.590322 | 0.846092 | 0.542652 | 0.651531 | 0.844817 |

---

**Representative side-by-side sample comparison (T3–T8)**

**Ground Truth**

[T3] They then reach the first "?" Block of the game.
[T4] They encounter the six-block triangle formation of "?" Blocks and regular bricks, with a Magic Mushroom inside the left block.
[T5] A set of three Pipes can be found, with Little Goombas located in between them.
[T6] The final pipe among the set leads to a bonus area with 19 coins and can be entered to bypass much of the level, exiting at the fifth pipe near the end.
[T7] If the pipe is not taken, a hidden 1-Up Mushroom block appears just after the pipes.
[T8] A pit follows, then another "?" Block containing either a Magic Mushroom or Fire Flower, depending on the player's form.

**o3 Generated**

[T3] Striking the Question Block yields a Super Mushroom that slides right along the ground past the hill.
[T4] Mario next encounters a pair of spaced green pipes emerging from the ground, each occasionally spawning a snapping Piranha Plant.
[T5] Between those pipes, two more Little Goombas approach in tandem over flat terrain.
[T6] Three floating Question Blocks form a row above bricks, with the middle hiding a helpful Fire Flower.
[T7] A lone Koopa Troopa patrols beneath these blocks.
[T8] Shortly past this, a tall staircase of ground bricks leads up then down, concealing a hidden 1-Up Block atop the first step.

---

Figure 16: A representative side-by-side sample comparison between o3 and ground truth.

### D.4 SUMMARY

Our contamination analysis reveals:

- **Vision-level contamination** remains negligible: models reliably reconstruct only the first and last frames, and their alignment coefficients (Kendall's $\tau$, RBO) show at best moderate correlation that is not statistically linked to overall performance rank, indicating no reliance on memorized visual sequences."'

- **Text-level contamination** is initially significant in *Ace Attorney*, where models' performance strongly correlates with script similarity. After prompt-based mitigation, this correlation disappears, and performance aligns instead with independently judged reasoning quality.

- **LLM-as-Judge (o3)** evaluations confirm that post-mitigation success stems from causal reasoning rather than rote recall. This reinforces the importance of controlled prompting for disentangling memorization from genuine inference.

## E PROMPT OPTIMIZATION

In this section, we present a case study illustrating our two-stage prompt optimization approach. We design two empirically optimized baseline prompts developed by equally adequate computer science graduate students based on Figure 1. Subsequently, we employ DSPy to bootstrap an optimized prompt for each baseline, selecting from a diverse set of five optimizer models and retaining only the best-performing prompt. We show performance variance across prompts optimized through bootstrapping is lower than that of baseline prompts.

## E.1 EMPIRICALLY OPTIMIZED BASELINE PROMPTS

---

**Game 2048 — Empirically Optimized Prompt Template 1**

**system_prompt:**
You are an intelligent AI player playing the 2048 game. Your goal is to make strategic moves to combine tiles and reach the highest possible tile value.

IMPORTANT: You MUST format your response using EXACTLY these lines:
thought: [Your reasoning about the game state]
move: [move]
Where [move] must be one of: "up", "down", "left", or "right".
Do not include # or any other prefix. Start directly with "thought:" followed by your analysis.

**user_prompt:**
2048 Game Quick Guide:
Primary Goal: Combine like tiles to create tiles with higher values.
Ultimate Goal: Create a tile with the value 2048 or higher.
Game Mechanics:
- The game is played on a 4x4 grid.
- Each move (up, down, left, right) shifts all tiles in that direction.
- Tiles with the same value that collide during a move combine into a single tile with twice the value.
- After each move, a new tile (2 or 4) appears in a random empty cell.
- The game ends when there are no valid moves left.
Action Space:
You must select one of these 4 moves:
- up: Shift all tiles upward
- down: Shift all tiles downward
- left: Shift all tiles to the left
- right: Shift all tiles to the right
Key Strategies:
1. Build a stable structure - Keep your highest value tiles in a corner.
2. Maintain a clear path - Always have a direction where you can combine tiles.
3. Chain reactions - Set up sequences of merges that can happen in a single move.
4. Look ahead - Think about the consequences of your moves 2-3 steps ahead.
5. Building patterns - Common patterns include: (1) Snake/Zig-zag pattern: Arrange tiles in decreasing order in a zigzag; (2) Corner anchoring: Keep the highest tile in a corner and build around it.
Avoid:
- Getting high-value tiles stuck in the middle of the board
- Creating scattered small values that block potential merges
- Making moves that could lead to grid lock
Previous Game History:
{Previous Game History}
Please analyze the 2048 board and determine the best move.
{Symbolic Board Features}
Key considerations:
- Look for opportunities to merge similar tiles
- Maintain your highest tiles in a corner
- Keep space for new tiles to appear
- Avoid trapping high-value tiles in the middle
IMPORTANT - FORMAT YOUR RESPONSE EXACTLY LIKE THIS:
thought: [your analysis here]
move: [move]
Where [move] must be one of: "up", "down", "left", or "right".
Do NOT use # or any other prefix. Start directly with "thought:" followed by your analysis.

---

## Game 2048 — Empirically Optimized Prompt Template 2

**system_prompt:**
You are an AI agent specialized in 2048 gameplay, your purpose is to analyze board states and suggest optimal moves that maximize your scores.

## Your Available Actions
For each turn, you must select one command:
- up: Shifts the entire grid upward
- down: Shifts the entire grid downward
- left: Shifts the entire grid leftward
- right: Shifts the entire grid rightward

When you choose a direction (up, down, left, or right), all tiles shift accordingly. Matching tiles that collide during this shift combine into a single tile representing their sum. After every move, a new tile with a value of either 2 or 4 appears in a random empty cell. The game concludes when no legal moves remain.

**user_prompt:**
## 2048 Gameplay Strategies
### Principles The most successful 2048 strategies typically involve:
1. Establish your highest-value tile in one corner and build a descending value structure around it.
2. Maintain consistent movement patterns that preserve your high-value corner configuration while allowing for regular merges.
3. Anticipate how each potential move affects not just the immediate board state but your options 2-3 moves ahead.
4. Create opportunities for chain reactions where multiple merges can occur in a single directional move.
5. Implement proven arrangements such as:
- Decreasing value snakes that zigzag across the board.
- Corner-anchored structures with decreasing values along the edges.

### Pitfalls to Avoid
Certain decisions consistently lead to board deterioration:
- Allowing high-value tiles to become isolated in central positions.
- Creating scattered low-value tiles that impede potential combinations.
- Making moves that reduce overall board fluidity and movement options.

## Current Game Context
{Previous Game History}

## Board Analysis
{Symbolic Board Features}

## Response Protocol
**YOUR ANALYSIS MUST STRICTLY ADHERE TO THIS FORMAT:**
thought: [Provide your detailed reasoning about the current board state, potential moves, and strategic implications]
move: [move]

Your move selection must be one of these exact terms: "up", "down", "left", or "right".

Begin your response directly with "thought:" followed by your strategic analysis. Do not include any prefixes, headers, or additional formatting.

---

**Algorithm 1:** Standardizing Gaming Prompt Optimization with SIMBA from DSPY

---

**Input:** Training environments $\mathcal{E}_{\text{train}}$, development environments $\mathcal{E}_{\text{dev}}$, *target* LM set $\mathcal{M}_t$ for
performance evaluation,
*optimizer* LM set
$\mathcal{M}_o = \{\texttt{o3}, \texttt{gemini-2.5-pro}, \texttt{claude-3.7-think}, \texttt{deepseek-R1}, \texttt{grok3-mini}\}$,
maximum optimisation steps $k$
**Output:** Best prompt module $\mathcal{P}^\star$ (highest mean dev score over all $M_t$)

$\mathcal{P} \leftarrow \texttt{ChainOfThought}(\text{"state} \rightarrow \text{action"})$ $s_{\text{best}} \leftarrow -\infty, \mathcal{P}^\star \leftarrow \mathcal{P}$;
**foreach** $M_o \in \mathcal{M}_o$ **do**

    $\texttt{dspy.configure(lm=}M_o\texttt{)}$;
    **// joint optimisation across *all* target LMs**
    $\mathcal{O} \leftarrow \texttt{SIMBA}(\{M_t\}, k)$;
    $\widehat{\mathcal{P}} \leftarrow \mathcal{O}.\texttt{compile}(\mathcal{P}, \mathcal{E}_{\text{train}})$;

    **// evaluate average dev score over every $M_t$**
    $s_{\text{avg}} \leftarrow 0$;
    **foreach** $M_t \in \{M_t\}$ **do**
        $\texttt{dspy.configure(lm=}M_t\texttt{)}$;
        $s_{\text{avg}} \mathrel{+}= \texttt{Evaluate}(\widehat{\mathcal{P}}, \mathcal{E}_{\text{dev}})$;

    $s_{\text{avg}} \leftarrow s_{\text{avg}}/|\{M_t\}|$;
    **if** $s_{\text{avg}} > s_{\text{best}}$ **then**
        $s_{\text{best}} \leftarrow s_{\text{avg}}$;
        $\mathcal{P}^\star \leftarrow \widehat{\mathcal{P}}$;

**return** $\mathcal{P}^\star$

---

## E.2 DSPy Optimized Prompts and Comparison

---

### Game 2048 — DSPy Optimized Prompt Template 1

**system_prompt:**

You are an AI agent specifically designed to play the game 2048. Your primary objective is to make strategic moves that effectively merge tiles to achieve the highest possible tile value.

**user_prompt:**
## Game Overview
The game 2048 involves combining identical number tiles on a grid to create tiles with progressively higher values.

## Game Mechanics
- The game is played on a **4×4 grid**
- Each move (up, down, left, right) shifts all tiles in the chosen direction
- When two identical tiles collide during a move, they merge into a single tile with twice the value
- After each move, a new tile (either 2 or 4) appears randomly in an empty cell
- The game concludes when no legal moves remain available

## Action Space
- **up**: Shifts all tiles toward the top of the grid
- **down**: Shifts all tiles toward the bottom of the grid
- **left**: Shifts all tiles toward the left side of the grid
- **right**: Shifts all tiles toward the right side of the grid

## Strategic Principles
1. **Corner Anchoring**: Position your highest-value tile in a corner and build around it
2. **Structural Stability**: Arrange surrounding tiles in descending order to create a stable formation
3. **Maintaining Merge Paths**: Always keep at least one direction available for safe combinations
4. **Creating Chain Reactions**: Set up moves that trigger multiple merges in a single action
5. **Forward Planning**: Think 2-3 moves ahead to avoid grid-lock and maintain empty spaces

## Pitfalls to Avoid
- Allowing high-value tiles to drift into central positions
- Scattering small-value tiles that obstruct potential merges
- Making moves that leave the board with no follow-up merge opportunities

## Context Variables
### Previous Game History
{Previous Game History}

### Board Features
{Previous Game History}

## Response Format
Your response must follow this exact two-line format:

```
thought: [your brief analysis of the current board state]\\
move: [up|down|left|right]\\
```

**Important**: Include nothing else beyond these two lines. No additional text, prefixes, symbols, or explanations.

---

**Game 2048 — DSPy Optimized Prompt Template 2**

**system_prompt:**
You are an **AI agent** playing **2048**. Your objective is to select moves that merge tiles efficiently and achieve the highest possible tile value.

**user_prompt:**
# 2048 Gaming Guide
## Primary Goal
Combine like tiles to reach **2048** or higher.

—

## Game Mechanics
- Played on a **4×4 grid**; each move ("up", "down", "left", "right") shifts every tile.
- Identical tiles that collide merge into one tile with **double the value**.
- After each move, a new tile (**2** or **4**) appears randomly in an empty cell.
- The game ends when **no legal moves** remain.

—

## Action Space
- **up**: Shift all tiles upward.
- **down**: Shift all tiles downward.
- **left**: Shift all tiles to the left.
- **right**: Shift all tiles to the right.

—

## Key Strategies
1. **Corner anchor** - Park your highest tile in one corner and build around it.
2. **Stable structure** - Arrange surrounding tiles in descending order to protect the corner.
3. **Clear merging path** - Keep at least one direction available for safe combinations.
4. **Chain reactions** - Set up moves that trigger multiple merges in one swipe.
5. **Look ahead** - Plan 2–3 moves in advance to avoid grid-lock. Preserve empty spaces for new tiles.

—

## Avoid
- Letting high-value tiles drift into the center.
- Scattering small tiles that block merges.
- Moves that leave the board with no follow-up merges.

—

## Previous Game History
{Previous Game History}

## Board Features
{Symbolic Board Features}

**Response format (use exactly two lines):**
thought: your brief analysis of the current board
move: up | down | left | right

Include nothing else—no prefixes, symbols, or extra text.

### E.3 Performance Comparison

In LMGame-Bench, we follow the SIMBA optimizer implementation in DSPy with performance metrics defined in Section 3.1.2 to optimize the prompt with five optimizer models: Claude-3-7-sonnet, Gemini-2.5-Pro-Preview, o3, Deepseek-r1, Grok-3-Mini-Beta, to search the best performing prompt yielding highest average reward cross all target models in 20 optimization steps ($k = 20$).

Table 14: Model performance across various prompt types in Game 2048 with harness, where $\Delta_e$ and $\Delta_p$ stand for performance difference between empirically deisgn prompt pairs and DSPy optimzied prompt pairs. P1 and P2 denotes to different prompt templates.

| Model | Empirical P1 | Empirical P2 | $|\Delta_e|\,(\downarrow)$ | DSPy P1 | DSPy P2 | $|\Delta_p|\,(\downarrow)$ |
|---|---|---|---|---|---|---|
| gemini-2.5-flash-preview-04-17 | 1697.3±548 | 1478.7±440 | 218.6 | 1746.0±518 | 1601.3±174 | **144.7** |
| claude-3-5-sonnet-20241022 | 2624.0±466 | 2235.3±862 | 388.7 | 2786.0±290 | 2928.0±318 | **142.0** |
| o4-mini-2025-04-16 | 4432.0±1096 | 3680.0±963 | 752.0 | 3851.3±864.4 | 4320.0±700 | **468.7** |

Among three target models: Gemini-2.5-Flash-Preview, Claude-3-5-Sonnet, o4-mini, experiments results from LMGame-Bench show evidence that our prompt optimization pipeline can reduce performance discrepancy between two candidate prompts by 33.8% to 63.5% on the three models across 3 runs. Details are reported in Table 14.

## F Additional Correlation Study

### F.1 Benchmark List for Correlation Study

We use 20 publicly available benchmarks spanning seven capability categories, including factual knowledge (VALS AI, 2025; Hendrycks et al., 2021; Phan et al., 2024; Scale AI, b;c; Rein et al., 2023; VALS AI), physics (Qiu et al., 2025; Hao et al., 2025), mathematics (Hao et al., 2025; Vals AI, a; Patel et al., 2024; Vals AI; White et al., 2024; LiveBench Team), code generation (Zhuo et al., 2024; Aider Team; BigCodeBench Team), visual reasoning (Scale AI, e; LMSYS Org; Yue et al., 2023; Vals AI, b), language understanding (Sirdeshmukh et al., 2025; Scale AI, d), and puzzle solving (Wang et al., 2025; Scale AI, a; Mazur). These benchmarks are chosen to provide a comprehensive view of general-purpose model abilities and to support the correlation and decomposition analyses.

Table 15 summarizes the per-model rankings across all benchmarks, grouped by category. The rankings are used to compute Spearman correlations and to uncover latent capability axes through low-rank decomposition.

### F.2 Correlation and Latent Feature Analysis with Super Mario Bros.

To better understand the impact of high-variance games, we conducted a supplementary analysis that includes *Super Mario Bros. (SMB)* in both the Spearman correlation matrix and latent ability decomposition.

**Spearman Correlation with SMB.** Figure 17a shows the extended Spearman correlation matrix including SMB. Although most benchmarks retain positive mutual correlations, SMB displays a notably weaker with language-heavy and code-generation benchmarks, where correlations drop to near zero or slightly negative values. This suggests that SMB performance is less stable between models and less aligned with other benchmarked capabilities, likely due to high variance or vision-specific difficulties. These observations support our decision to exclude SMB from the main correlation and decomposition analyses in Section 4.3.

**Latent Ability Decomposition with SMB.** We also repeated the low-rank factorization analysis with SMB included. Figure 17b shows the contribution of each benchmark (including SMB) to the discovered latent features. SMB contributes moderately across most features, particularly in Feature 1 (long-context language reasoning) and Feature 3 (puzzle solving & coding capabilities). This aligns with SMB's demand for multimodal reasoning—visual perception, spatial planning, and action timing. However, its weights are less concentrated, likely due to variance in model rankings across runs.

Table 15: Model rankings (1 = best) across 20 benchmarks, grouped by capability category. Abbreviated model names: C3.5 = claude-3.5-Sonnet-20241022, C3.7 = claude-3.7-Sonnet-20250219-thinking, G4O = gpt-4o-2024-11-20, O1 = o1-2024-12-17, O3 = o3-2025-04-16, Gem = gemini-2.5-pro-preview-05-06(thinking), L4 = llama-4-maverick-17b-128e-instruct-fp8, O4m = o4-mini.

| Category | Benchmark | C3.5 | C3.7 | Gem | L4 | G4O | O1 | O3 | O4m |
|---|---|---|---|---|---|---|---|---|---|
| Factual | MMLU-Pro | 7 | 4 | 2 | 6 | 8 | 3 | 1 | 5 |
| | GPQA | 7 | 3 | 2 | 6 | 8 | 5 | 1 | 4 |
| | HLE | 7 | 4 | 3 | 6 | 8 | 5 | 1 | 2 |
| | HLE (Text) | 7 | 4 | 3 | 6 | 8 | 5 | 1 | 2 |
| Physics | EMMA-Physics | 7 | 4 | 1 | 8 | 6 | 5 | 3 | 2 |
| | PHYBench | 7 | 5 | 1 | 8 | 6 | 4 | 2 | 3 |
| Math | Math 500 | 8 | 4 | 1 | 6 | 7 | 5 | 2 | 3 |
| | AIME 2025 | 8 | 5 | 1 | 6 | 7 | 4 | 2 | 3 |
| | EMMA-Math | 6 | 4 | 1 | 8 | 6 | 5 | 3 | 2 |
| | LiveBench-Math | 7 | 5 | 1 | 6 | 8 | 4 | 2 | 3 |
| Code | BigCodeBench | 5 | 1 | 3 | 6 | 4 | 2 | 7 | 8 |
| | Aider Coding | 6 | 4 | 2 | 8 | 7 | 5 | 1 | 3 |
| | LiveBench-Code | 3 | 4 | 5 | 8 | 6 | 7 | 2 | 1 |
| Vision | VISTA | 6 | 4 | 1 | 7 | 8 | 5 | 3 | 2 |
| | MMMU | 7 | 5 | 1 | 6 | 8 | 4 | 2 | 3 |
| | Chatbot Arena (Vision) | 7 | 5 | 1 | 6 | 8 | 4 | 2 | 3 |
| Language | MultiChallenge | 5 | 2 | 3 | 7 | 8 | 4 | 1 | 6 |
| | LiveBench-Lang | 6 | 4 | 3 | 7 | 8 | 2 | 1 | 5 |
| Puzzle | EnigmaEval | 6 | 4 | 5 | 8 | 7 | 3 | 1 | 2 |
| | NYT Connections | 8 | 5 | 4 | 7 | 6 | 2 | 1 | 3 |

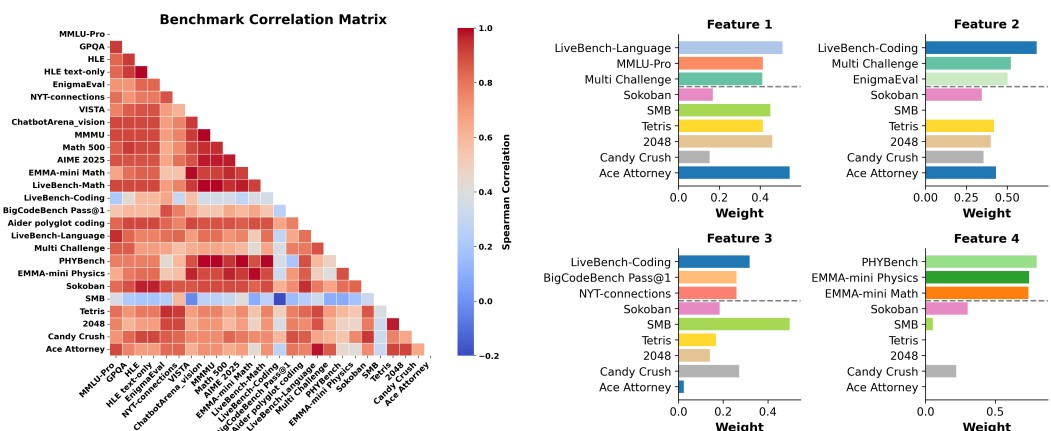

(a) Spearman correlation matrix including *Super Mario Bros. (SMB)*.

(b) Top-weight benchmarks for each latent feature when SMB is included.

Figure 17: Benchmark relationships overview.

## F.3 VISUALIZING BENCHMARK RELATIONSHIPS.

To better understand how LMGame-Bench compares with established benchmarks, we visualize benchmark similarity using t-SNE. We embed each benchmark and game as a high-dimensional vector based on either model performance scores (ranging from 0–100) or model rankings, and project them into 2D using t-SNE. Benchmarks with NaN values (e.g., missing model scores) were excluded to ensure reliable embeddings.

We show two versions:

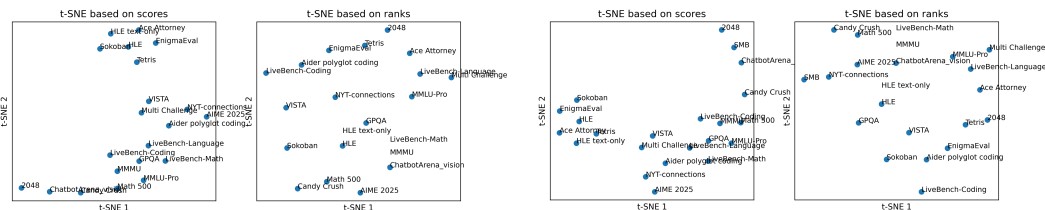

(a) t-SNE analysis. Without Super Mario Bros.    (b) t-SNE analysis. With Super Mario Bros.

Figure 18: Benchmark relationships overview.

- The first version (Figure 18b) excludes Super Mario Bros. (SMB), reflecting the setup used in the main paper.

- The second version (Figure 18a) includes SMB to explore its positioning relative to other benchmarks.

In the score-based plots, *Ace Attorney*, *Sokoban*, and *Tetris* cluster closely with reasoning-heavy benchmarks like EnigmaEval and HLE, reflecting shared demands in long-horizon planning and symbolic reasoning. With SMB included, it appears adjacent to 2048 and ChatbotArena-Vision, consistent with its reliance on visual perception and spatial coordination.

In the rank-based projections, *Candy Crush* and *2048* are near Math 500 and AIME 2025, while *Sokoban* remains isolated, likely due to its unforgiving action space. SMB's placement shifts closer to vision benchmarks, but does not cluster tightly, likely due to its high inter-model variance..

Putting together, the two tSNE graphs support the conclusion that different LMGame-Bench games probe distinct capabilities and align with well-established benchmark clusters in meaningful ways.

## F.4 LINEAR MODELING

We also employ linear modeling to predict game ranking based on model capabilities across different categories, serving as a complement to the other two methods in Sec. 4.3. Let $m$ be the number of models to be ranked, $n$ be the total number of existing benchmark categories, $R_{i,j} \in \mathcal{Z}_{\geq 1}$ be an ordinal rank of model $i$ on a benchmark from category $j$, $G_{i,g} \in \mathcal{Z}_{\geq 1}$ be the rank for model $i$ on game $g$. Correspondingly, we define $\mathbf{R}_j$ and $\mathbf{G}_g$ be the rank vector for category $j$ and a game $g$ across all models. Since we can collect ranking data from all models across combinations of benchmarks in different categories, for each benchmark combination, we can then define a polynomial feature map for polynomial expansion as shown in Eq. 4.

$$\phi : \mathbb{R}^n \to \mathbb{R}^{p(d)}, \text{ where } p(d) = \sum_{k=0}^{d} \binom{n+k-1}{k}, \text{ denote } \Phi = \begin{pmatrix} \phi(\mathbf{L}_1)^\top \\ \vdots \\ \phi(\mathbf{L}_m)^\top \end{pmatrix} \in \mathbb{R}^{m \times p} \quad (4)$$

As a result, let $\mathbf{w} \in \mathcal{R}^p$ be the parameters to be learned, we can predict the gaming ranking of a model $i$ given its ranking on a set of benchmarks from all categories of interest. $\hat{G}_{i,g} = \mathbf{w}\phi(\mathbf{L}_i)$, $\hat{\mathbf{G}}_g = \Phi\mathbf{w}$. With the non-negative least-square-fit objective, $\min_{\mathbf{w}} \left\| \Phi\mathbf{w} - \mathbf{g} \right\|_2^2$ s.t. $w_k \geq 0$ for all linear terms $k$, where the closed-form solution can be expressed as $\mathbf{w}^* = \left( \Phi^\top \Phi \right)^{-1} \Phi^\top \mathbf{g}$. Non-negativity ensures every feature contributes additively, otherwise being worse in category $j$ makes gaming performance better is counter-intuitive. Each term in $\mathbf{w}$ quantifies how much and in what direction the polynomial term of the $n$ categories drives the game ranking. We show linear and quadratic models trained on benchmarks presented in Appx. F.1.

When using the following four benchmark categories: language, physics understanding, mathematics, and coding as explanatory variables, Table 16 reveals how each game decomposes into familiar skill domains. Long-horizon games like Sokoban, Tetris and 2048's rankings are driven primarily by math and coding performance. Games requiring spatial reasoning, like Sokoban, Candy Crush and Super

Table 16: Learned weights for game ranking prediction using a linear model, where $r$ and RE denote for Pearson's r and mean-normalized residual errors respectively.

| Game | Language | Physics | Math | Coding | Offset | $r$ | RE |
|---|---|---|---|---|---|---|---|
| Sokoban | 0.823 | 1.658 | **2.043** | **1.954** | 0.372 | 0.928 | 0.475 |
| Tetris | 1.847 | 0.926 | **2.116** | **2.139** | 0.542 | 0.819 | 0.825 |
| Ace Attorney | **2.962** | 0.384 | 2.536 | 0.445 | 0.683 | 0.845 | 0.832 |
| Super Mario Bros | 0.480 | **1.318** | 0.902 | 0.227 | 2.852 | 0.264 | 1.392 |
| 2048 | 1.638 | 1.045 | **2.097** | 1.521 | 0.960 | 0.742 | 0.966 |
| Candy Crush | 1.074 | **1.909** | 1.586 | 1.409 | 0.440 | 0.854 | 0.783 |

Table 17: Additional ablation on polynomial (linear & quadratic) modeling with different combinations of core capablities.

| Linear Model ($n = 5$) | | | | | | | |
|---|---|---|---|---|---|---|---|
| Game | Knowledge | Puzzle | Visual | Math | Coding | Offset | $r$ |
| Sokoban | 1.299 | 2.426 | 0.009 | 1.731 | 1.482 | 0.106 | 0.9754 |
| Tetris | 0.000 | 6.559 | 0.964 | 0.005 | 0.455 | 0.009 | 0.9370 |
| Ace Attorney | 1.579 | 4.850 | 0.003 | 0.680 | 0.000 | 0.245 | 0.8519 |
| Super Mario Bros | 0.653 | 0.000 | 0.000 | 1.304 | 0.737 | 2.970 | 0.2215 |
| 2048 | 0.000 | 4.958 | 0.000 | 0.000 | 0.000 | 1.588 | 0.7338 |
| Candy Crush | 2.188 | 3.326 | 0.000 | 0.916 | 1.583 | 0.000 | 0.9086 |

| Linear Model ($n = 4$) | | | | | | |
|---|---|---|---|---|---|---|
| Game | Knowledge | Visual | Math | Coding | Offset | $r$ |
| Sokoban | 2.581 | 0.011 | 1.954 | 2.029 | 0.308 | 0.9471 |
| Tetris | 3.432 | 0.405 | 1.063 | 2.116 | 0.558 | 0.8128 |
| Ace Attorney | 4.493 | 0.014 | 1.582 | 0.000 | 0.820 | 0.7481 |
| Super Mario Bros | 0.942 | 0.715 | 1.102 | 0.974 | 2.289 | 0.2535 |
| 2048 | 0.004 | 0.000 | 2.249 | 0.000 | 3.111 | 0.3610 |
| Candy Crush | 3.646 | 0.000 | 1.600 | 2.393 | 0.198 | 0.8388 |

| Quadratic Model ($n = 3$) | | | | | | | | | | | |
|---|---|---|---|---|---|---|---|---|---|---|---|
| Game | Klg | Math | Coding | $Klg^2$ | $Math^2$ | $Coding^2$ | Klg $\times$ Math | Klg $\times$ Coding | Math $\times$ Coding | Offset | RE |
| Sokoban | 1.284 | 0.439 | 0.758 | 0.499 | 0.934 | 0.963 | 0.673 | 0.616 | 0.586 | 1.497 | 0.801 |
| Tetris | 1.266 | 0.423 | 0.745 | 0.481 | 0.921 | 0.946 | 0.657 | 0.623 | 0.600 | 1.489 | 0.797 |
| Ace Attorney | 1.546 | 0.629 | 0.0111 | 1.184 | 0.725 | 0.025 | 0.910 | 0.428 | 0.330 | 1.626 | 1.037 |
| Candy Crush | 1.456 | 0.428 | 1.410 | 0.343 | 1.627 | 1.021 | 0.951 | 0.000 | 0.000 | 1.098 | 0.7503 |

Mario Bros, align closely with the physics-understanding benchmark. Text-heavy narrative games like Ace Attorney are dominated by language-related benchmarks. Notably, prediction quality for Super Mario Bros is very low ($r = 0.26$), suggesting that the spatiotemporal reasoning it requires is unique when compared with the listed benchmarks. Comparisons of polynomial fitting using linear models and quadratic models of different categorical combinations are presented in Table 17.

# G MULTI-AGENT EVALUATION: TEXAS HOLD'EM

To expand the diversity of interactive reasoning settings in LMGame-Bench , we introduce a new **Texas Hold'em** game. Unlike our turn-based single-agent games, Texas Hold'em requires *multi-agent strategic reasoning under imperfect information*, making it a natural stress test for long-horizon planning, hidden-state inference, and risk-sensitive decision-making.

## G.1 GAME DESIGN

We adapt the PettingZoo Texas Hold'em environment for LMGame-Bench -Bench, enabling multi-agent interaction under standard no-limit rules. Each agent receives two private hole cards, and five community cards are revealed across the flop, turn, and river. At every decision point, an agent observes its hole cards, all public cards revealed so far, the current pot and blind level, the stack sizes of all players, and the full betting history of the hand. Agents choose among four canonical actions—`fold`, `check`, `call`, or `raise`—using a discrete set of legal bet sizes. Pots are awarded

either at showdown or when all other players fold, creating a strategic environment that mixes imperfect information, multi-round reasoning, and risk-sensitive decision-making.

For multi-agent evaluation, we run a round-robin format where every model plays against every other model. Each head-to-head match consists of 40 hands with equal stacks and fixed blinds, and we record all action traces, chip flows, and rendered videos for post-hoc analysis.

**Game Prompts.** To study how instruction framing affects playstyle, we design three prompt versions: (1) **V0: Rules-Only Baseline**, which supplies only the formal rules with no strategic hints; (2) **V1: Strategic Heuristics**, which adds common poker principles such as position, pot odds, and stack management; and (3) **V2: Passive Bias**, which encourages "seeing more cards" through checks, calls, and pot control, often producing more cautious play.

**DSPy Prompt Optimizer.** We automate prompt refinement using a lightweight DSPy-based optimizer that rewrites prompt sections, preserves placeholders, and evaluates each candidate through controlled self-play. Each iteration runs two mirrored tournaments—candidate vs. baseline and baseline vs. candidate—to avoid positional bias. The score is the average chip differential across hands, and the best-performing prompt is promoted to the next generation.

**Ranking with TrueSkill2.** Model performance is measured using TrueSkill2, a Bayesian rating system originally developed for competitive gaming. Each model begins with a prior $N(\mu = 25, \sigma \approx 8.33)$, and ratings are updated once per round-robin, rather than after each hand, to counteract the high variance of single-hand outcomes. Instead of binary win/loss signals, we rank players by final chip counts, convert them into weighted continuous ranks, and apply TrueSkill2's update rule. The mean $\mu$ reflects estimated skill, while the variance $\sigma$ quantifies confidence and decreases as more tournaments are completed. This setup yields stable, sample-efficient ratings that are sensitive to overall chip performance rather than noisy individual hands.

### G.2 BEHAVIORAL METRICS AND STYLE DEFINITIONS

**Aggression Factor (AF).**
$$AF = \frac{\text{Bets} + \text{Raises}}{\text{Calls}}$$
Measures how often a model chooses an aggressive action over a passive one.

**Fold Rate (FR).**
$$FR = \frac{\#\text{Folds}}{\#\text{Hands}}$$
Captures the model's overall level of caution across hands.

Balanced AF and FR values generally correspond to more stable long-horizon behavior, while extreme tendencies (very high FR or very high AF) typically lead to weaker performance.

We classify models using FR (tight vs. loose) and AF (aggressive vs. passive). Thresholds are tunable; by default we use: A model is classified as tight if $FR \geq 0.72$, otherwise loose. A model is classified as aggressive if $AF \geq 1$, otherwise passive. This yields four styles we report alongside TrueSkill2:

**Tight-Aggressive (TAG).** High FR and high AF; selective preflop entry, applies pressure when involved.

**Tight-Passive (TP).** High FR and low AF; selective entry with a conservative action profile.

**Loose-Aggressive (LAG).** Low FR and high AF; frequent entry and high-pressure multi-street play.

**Loose-Passive (LP).** Low FR and low AF; frequent entry but minimal application of pressure.

### G.3 V1 STRATEGIC HEURISTICS TOURNAMENT RESULTS

Using the V1 prompt, which adds foundational poker heuristics such as position, pot odds, and stack discipline, we ran a 110-round round-robin tournament across eight models. As shown in Figure 19,

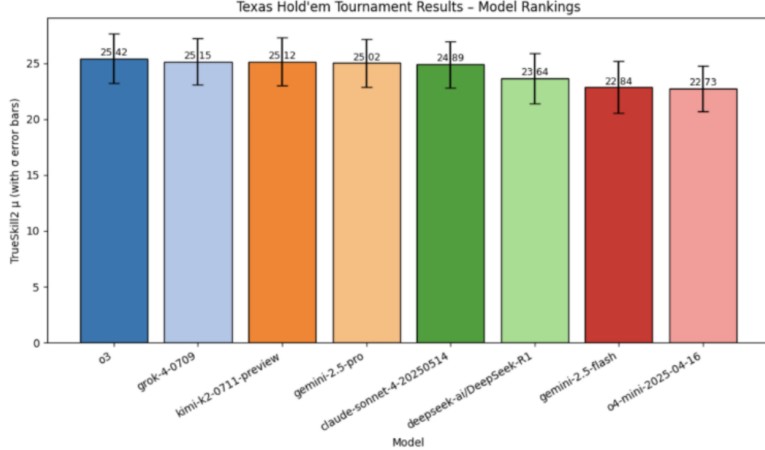

Figure 19: TrueSkill2 ranking of eight models under the **V1 Strategic Heuristics** prompt over 110 rounds.

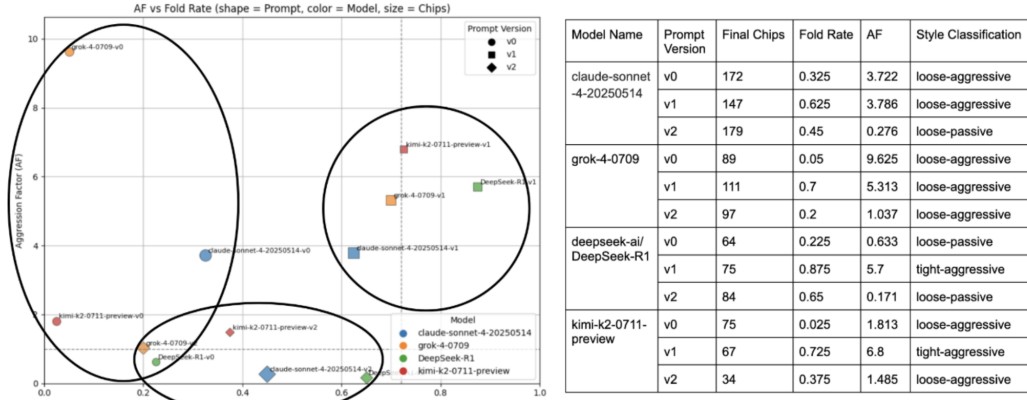

Figure 20: Prompt sensitivity across four models and three prompt versions. Left: AF–FR scatter plot (shape = prompt version, color = model). Right: Table summarizing AF, FR, and inferred play style.

all matches used mirrored seating, equal stacks, and fixed blinds to ensure reproducibility. The resulting TrueSkill2 posterior means show a clear ordering: **o3** performed strongest, followed by **grok-4**, **kimi-k2**, and **gemini-2.5-pro**. The middle tier included **claude-4-sonnet** and **deepseek-r1**, while **o4-mini** and **gemini-2.5-flash** formed the lower tier. These results indicate that V1's heuristic framing meaningfully influences betting stability and overall strategic consistency.

### G.4 PROMPT SENSITIVITY AND BEHAVIORAL TAKEAWAYS

Across four models and three prompt versions, we observe clear clustering in AF–FR space (Figure 20). Under the rules-only baseline (v0), models concentrate on the left side of the plot, exhibiting loose-aggressive tendencies with high variance across models. Adding the strategy-focused prompt (v1) shifts behavior toward a more disciplined region, producing tight-aggressive profiles with stronger ranges, more selective folding, and steadier chip accumulation. In contrast, the passive prompt (v2) pulls models into a passive cluster characterized by reduced betting, excessive checking, and conservative decision-making that limits upside potential.

These shifts reveal that models do not have fixed "personalities," but rather *stylistic priors* that respond strongly to prompt framing. The same system can move between loose-aggressive, tight-aggressive, and loose-passive regions depending on instruction clarity. Yet persistent biases remain:

Grok-4-0709 tends to remain loose-aggressive even under strategic prompts, DeepSeek-R1 defaults to loose-passive unless guided explicitly, and Claude and Kimi frequently revert to loose-aggressive patterns with varying degrees of modulation. Together, the three clusters illustrate how prompts function as a behavioral control knob—v0 expands variance, v1 encourages disciplined aggression, and v2 compresses decisions into overly passive play.

### G.5 CONCLUSION

Integrating multi-agent Texas Hold'em into LMGame-Bench introduces a new axis of evaluation for LLMs: strategic behavior under uncertainty. By combining round-robin play, TrueSkill2 ranking, and behavioral profiling through aggression and fold-rate metrics, our framework provides a richer view of how models compete, adapt, and reveal latent style biases in interactive settings. Prompt framing emerges as a powerful control mechanism, capable of shifting the same model across aggressive, disciplined, and passive behavioral regimes while also exposing persistent tendencies unique to each system. Future work may scale to larger tables, longer tournaments for improved statistical stability, and adaptive agents that adjust strategies dynamically in response to opponent behavior.

## H POKÉMON RED

Beyond the games presented in Section 3.1, we also experimented with Pokémon Red, tasking open-world navigation and competitive team building, as a candidate task for evaluating LLM and VLM agents. Despite the environment offers rich sets of tasking including navigation, combat control, and long-horizon planning, in the practice, we found Pokémon Red fails to serve a standardized and discriminative task with three main reasons.

Firstly, the navigation task involving visual perception and memorization of a partially observable map is extremely hard for current models: without any additional harness, all models we tested in Table 1 fail in navigation tasks. To enable navigation, we follow implementation by Anthropic and Gemini-2.5 (Anthropic, 2025; Comanici et al., 2025), which requires a sophisticated harness that reads internal emulator states like tile maps, collision flags, and warp locations and exposes a structured grid or mini-map to the model.

Secondly, once the navigation harness is in place, other tasks including battle control become very simple: with a reasonably well-trained team, gym battles are often decided by level advantage and type matchups, leaving limited headroom to distinguish model capabilities.

| Model | Steps to reach Oak's Lab | Cost | Time |
|---|---|---|---|
| OpenAI o3 | 1000 | $120 | 20h |
| Gemini 2.5 Flash | 1000 | $50 | 13h |

Table 18: Cost analysis on o3 and Gemini-2.5-Flash for reaching the first checkpoint, Professor Oak's Lab, in Pokémon Red.

Thirdly, the most important aspect of Pokémon Red as an evaluation - full-game, long-horizon team training - come with a unfavorable cost overhead when used as a standardized benchmark. As shown in Table 18, even in a very early-game setting like reaching Professor Oak's lab, we observe that a single 1,000-step run already incurs substantial cost and latency: for example, OpenAI o3 requires roughly 1,000 actions, about 120 dollars in API cost, and around 20 hours of wall-clock time, while Gemini 2.5 Flash needs a similar number of steps but still costs around 50 dollars and takes roughly 13 hours. Extrapolating to a full playthrough with tens of thousands of actions implies thousands of dollar costs per run and multi-day runtime, making it difficult to repeat experiments enough times for robust, statistically meaningful comparison across models and harness variants.

## I STANDARD ERROR

This appendix provides the standard–error version of the main results shown in Table 19. All reported scores follow the format *mean ± standard error*, offering a clearer view of run-to-run stability

Table 19: Model performance raw scores, evaluated in both with and without harness. For games marked with [†], evaluation for text-only models is not supported. The reported results represent averages over three runs, except for models marked with *. "N/A" = unsupported or unavailable.

| Model | Harness | Sokoban | SMB[†] | Tetris | 2048 | Candy Crush | Ace Attorney* |
|---|---|---|---|---|---|---|---|
| claude-3-5-sonnet-20241022 | No | 0.0±0.0 | 1540.0±12.5 | 12.3±1.5 | 57.8±9.5 | 17.0±10.4 | 1.0±0.0 |
| | Yes | 0.0±0.0 | 1267.7±279.5 | 14.7±0.7 | 108.2±3.3 | 106.0±30.8 | 2.0±0.0 |
| claude-3-7-sonnet-20250219 (thinking) | No | 0.0±0.0 | 1430.0±93.6 | 13.0±0.0 | 114.2±4.1 | 126.3±39.9 | 3.0±0.0 |
| | Yes | 2.3±0.9 | 1418.7±381.2 | 16.3±1.3 | 113.4±1.8 | 484.0±31.0 | 7.0±0.0 |
| deepseek-r1 | No | N/A | N/A | N/A | N/A | N/A | N/A |
| | Yes | 1.3±0.7 | N/A | 14.3±0.3 | 105.2±7.0 | 447.3±26.0 | 0.0±0.0 |
| gemini-2.5-flash-preview-04-17 | No | 0.0±0.0 | 1540.7±151.5 | 19.0±2.7 | 107.4±2.0 | 97.7±20.9 | 1.0±0.0 |
| | Yes | 1.7±0.9 | 1395.0±138.6 | 16.3±1.9 | 106.6±3.1 | 334.7±37.8 | 4.0±0.0 |
| gemini-2.5-pro-preview-05-06 | No | 1.0±0.0 | 1025.3±255.9 | 12.3±1.8 | 120.5±2.3 | 177.3±37.5 | 8.0±0.0 |
| | Yes | 4.3±0.3 | 1498.3±117.4 | 23.3±0.3 | 117.3±3.4 | 416.3±3.9 | 7.0±0.0 |
| llama-4-maverick-17b-128e-instruct-fp8 | No | 0.0±0.0 | 786.0±267.1 | 11.7±0.7 | 44.6±6.8 | 32.3±23.9 | 0.0±0.0 |
| | Yes | 0.0±0.0 | 1468.7±320.8 | 10.3±0.9 | 106.0±2.2 | 128.7±33.0 | 0.0±0.0 |
| gpt-4.1-2025-04-14 | No | 0.0±0.0 | 1991.3±588.0 | 13.0±1.0 | 94.5±9.8 | 101.0±69.4 | 0.0±0.0 |
| | Yes | 0.0±0.0 | 2126.3±1026.8 | 13.7±0.3 | 105.7±4.0 | 182.0±16.6 | 2.0±0.0 |
| gpt-4o-2024-11-20 | No | 0.0±0.0 | 1028.3±378.8 | 14.7±1.2 | 70.4±8.8 | 59.0±31.5 | 0.0±0.0 |
| | Yes | 0.0±0.0 | 2047.3±304.9 | 14.0±2.1 | 106.7±2.0 | 147.3±30.8 | 0.0±0.0 |
| o1-2024-12-17 * | No | 0.0±0.0 | 1434.0±0.0 | 13.0±0.0 | 128.1±0.0 | 90.0±0.0 | 3.0±0.0 |
| | Yes | 2.3±0.3 | 855.0±0.0 | 35.0±0.0 | 128.9±0.0 | 159.0±0.0 | 16.0±0.0 |
| o1-mini-2024-09-12 | No | N/A | N/A | N/A | N/A | N/A | N/A |
| | Yes | 1.3±0.3 | N/A | 11.7±0.7 | 114.0±2.2 | 48.0±19.6 | 0.0±0.0 |
| o3-2025-04-16 * | No | 2.0±0.0 | 1955.0±0.0 | 31.0±0.0 | 128.2±0.0 | 106.0±0.0 | 8.0±0.0 |
| | Yes | 8.0±2.0 | 3445.0±0.0 | 42.0±0.0 | 128.0±0.0 | 647.0±0.0 | 16.0±0.0 |
| o4-mini-2025-04-16 | No | 1.3±0.3 | 1348.3±102.8 | 15.0±2.1 | 97.6±16.8 | 110.7±28.7 | 2.0±0.0 |
| | Yes | 5.3±0.7 | 1448.0±92.9 | 25.3±4.9 | 120.6±2.8 | 487.3±114.3 | 4.0±0.0 |

compared to variance. For models evaluated with only a single run (marked with *), the standard error is effectively zero. "N/A" indicates evaluation settings that are not applicable, typically due to missing vision capabilities.

