# OpenReview forum: "lmgame-Bench: How Good are LLMs at Playing Games?"
_ICLR.cc/2026/Conference — ICLR 2026 Poster_

### Official Review · Reviewer_tEMA · 2025-10-26

**Soundness:** 2
**Presentation:** 2
**Contribution:** 1
**Rating:** 2
**Confidence:** 5

**Summary:**

The paper proposes a benchmark for LLMs on six game environments, they evaluated a variety of popular LLMs. The paper also studies contamination and attempts to create standardize prompting.

**Strengths:**

Having the human average for each of the games and random baselines is good. The study on contamination detection is interesting, and so is the attempt at standardizing the prompting with  DSPy standardization

**Weaknesses:**

I would argue that testing LLM without harness doesn’t make much sense at all. In the context of LLM Agents, the LLM is the brain (without memory module), the agentic scaffolding, or so called “harness” in the paper, is equivalent to affordances and tools such as arms, legs and eyes, memory. It’s not surprising to me at all how a naked LLM without these affordances (scaffolding or harness) often doesn’t outperform random. Memory particularly is very important for partially observable MDPs.

The benchmark could have been valuable around a year prior to submission, but now comes in a much more crowded space of benchmarks for LLM/VLM Agents, and the the papers struggles to find novelty both in the methodology as well as in insights provided.  The variances in most of the results are extremely large, and it’s difficult to disentangle real results from noise. The paper attempts to do something very similar to existing benchmarks specifically Balrog [Paglieri et al. ICLR 2025], and given the many similarities a more thorough comparison would be needed to explain what novelty (whether methodological or new insights) this paper brings.

Using o3 to generate perception traces also seems like a big methodological mistake as the rest of the models' capabilities will be influenced by another model.

**Questions:**

Could the author better argue what’s the difference between their “harness” compared to typical scaffoldings used in most related benchmarks?
Could the author try to better argue where the novelty of the benchmark comes from, and what insights are actually new, especially when compared to the many existing benchmarks in the area?
Why using o3 for perception? This feels like a methodological flaw.
How many seeds were tested for each game?

---

> ### Author Response · Authors · 2025-11-23
>
> We thank reviewer tEMA for recognizing our benchmark design with data contamination study and prompt optimization, as well as the constructive feedbacks! We answer each of your question and concerns below.
>
> ### **W1/Q1: Could the author better argue what’s the difference between their “harness” compared to typical scaffoldings used in most related benchmarks?**
>
> We agree that scaffoldings are essential for deployed agents. Our no-harness setting is not intended as a realistic deployment mode. Rather, **it serves as a capability-attribution ablation that disentangles how much performance can be attributed to the base model versus what is contributed by memory and perception scaffoldings**. In Lmgame-Bench, each harness can be selectively enabled or disabled (Table 4), allowing us to measure performance deltas and identify model-specific weaknesses.
>
> Existing benchmarks typically conflate these factors by evaluating models with all harnesses enabled, making it difficult to isolate and assess the contribution of each component. For instance, if a model has strong textual reasoning but weak perception, its overall score may appear high without revealing whether it excels at both perception + reasoning or is actually only good in reasoning and poor in perception (yet perception harness has significant mitigated the issue). In contrast, **Lmgame-Bench allows us to disentangle these effects: by toggling harnesses individually, we can identify cases where models are bottlenecked by perception**. For example, as shown in Table 4, we can infer both Claude 3.7 and o4-mini exhibit perception-driven bottlenecks in games such as Tetris.
>
> Our findings further show that the stronger the base model, the less the modular harness contributes. From Table 1, o3 achieves significantly higher performance than all other models on Tetris even when both perception and memory harnesses are removed, highlighting genuine underlying capability independent of scaffolding.
>
> ### **W2/Q2: Could the author try to better argue where the novelty of the benchmark comes from, and what insights are actually new, especially when compared to the many existing benchmarks in the area.**
>
> We argue Lmgame Bench is a more comprehensive and systematic game benchmark that better distinguishes model capabilities and is hard to saturate.
>
> - On **game choice and benchmark design**, our suite targets broader and more complete LLM capabilities than prior benchmarks. BALROG concentrates on grid-based navigation games, while we cover board, puzzle games, action games, and a visual novel adventure game, giving richer genre diversity and more human-like skill coverage. We also select challenging games with built-in difficulty ladders or unbounded scoring metrics, making our games hard to saturate. In contrast, many games from LMAct (e.g., Tic-Tac-Toe, GridWorld, Crosswords) are saturated and require a redesign. Meanwhile, games from VideoGameBench are too hard that most models make zero progress. Therefore, compared with related work, our benchmark evaluated more diverse capabilities with well-designed difficulty that is hard to saturate while still being able to distinguish models. We also uniquely address data contamination in game benchmarks and enforce prompt standardization for fair comparison.
>
> - On **agentic design**, our harness is built to ablate model capability, not to provide a single fixed agent. Unlike benchmarks that fix scaffoldings or only vary input modality, we systematically analyze modular harness effects (Table 2, Appendix B). The results are non-trivial: memory (history + reflections) helps non-reasoning models more than reasoning models on board games, and image perception is the main bottleneck for many models in almost all games except 2048, where visuals are simplest. Selectively turning on and off the perception module also provides us more insight, for example from Table 4, it can be inferred that the memory module contributes the most to final model performance in 2048, whereas the perception module contributes the most in Tetris. If both modules are turned on, such conclusions can not be drawn.
>
> - On **gaming performance analysis and insights**, we move beyond gaming scores to capability attribution. We introduce a novel latent ability decomposition analysis to map games to core LLM skills (Figure 3b), revealing connections from games to math, coding, physics, etc. For failure case analysis in Section 4.2, we give concrete examples (Figure 5,7,8,9) when summarizing models’ failure cases. We highlight the model's weakness on perception by varying visual complexity of games (Figure 5), and explain long-context challenges by ablate with less-context in Figure 9. In our newly added Figure 6, we also provide a novel analysis of perception-module effectiveness as a function of both visual complexity and model reasoning capability, which is an angle not covered in prior work.

---

> ### Author Response · Authors · 2025-11-23
>
> ### **W3/Q3: Why using o3 for perception?**
>
>
> The goal of employing the perception module is **to help LLMs capture as precise visual game states as possible**. So when the perception module is turned on, our benchmark stresses reasoning rather than low-level perception, it minimizes the impact of model's perception capability on the final outcome.
>
> To provide precise representation of the game states, for board games, we directly consume symbolic game states, which are by construction 100% accurate. In contrast, Ace Attorney and Super Mario only expose pixel observations with no ground-truth textual descriptions, so we require an automatic method to translate images into text. Given the diversity and complexity of these visuals, traditional CV pipelines (detection, OCR, hand-crafted rules) would be brittle and difficult to generalize, making VLM-as-perception-module a natural choice. We therefore choose o3 since it’s the strongest vision reasoning model at the time we conduct experiments for this paper. We also want to emphasize that the perception module is used solely for parsing the game state, including identifying objects, dialogue, UI components, and available options. We added the prompts in Appendix A.2.2.
>
>
> For text-only LLMs, this perception module is necessary for the model to play Super Mario Bros and Ace Attorney. For VLMs, we argue **the perception module won’t provide an unfair advantage to any of the models (including o3 itself)**.
>
> To sanity-check this assumption, we compare o3-based perception with gemini-2.5-pro/GPT-5.1-based perception on Ace Attorney across various VLM agents: gemini-2.5-flash-preview-09-2025, o4-mini-2025-04-16, and claude-sonnet-4-20250514. For completeness, we also include o3 and gemini-2.5-pro themselves in the table.
>
>
> | Model                          | o3 Score (run a;b;c) | o3 Total | Gemini Score (run a;b;c) | Gemini Total | GPT-5.1 Score (run a;b;c) | GPT-5.1 Total |
> |--------------------------------|------------------|----------|----------|--------------|----------|--------------|
> | o3-2025-04-16                  | (4; 8; 7)        | 19       | (8; 8; 2)             | 18           | (8; 8; 8)   |      24      |
> | gemini-2.5-pro                 | (6; 0; 7)        | 13       | (8; 6; 0)             | 14           | (8; 2; 2)              | 12             |
> | gemini-2.5-flash-preview-09-2025 | (0; 1; 4)      | 5        | (0; 4; 2)             | 6            | (2; 3; 5)              | 10             |
> | claude-sonnet-4-20250514       | (3; 0; 0)        | 3        | (2; 0; 0)             | 2            | (0; 2; 2              | 4              |
> | o4-mini-2025-04-16             | (0; 3; 0)        | 3        | (0; 0; 1)             | 1            | (2; 0; 0)              | 2              |
>
>
> | System Pair              | Kendall Tau | p-value  | Spearman Rho | p-value  | Interpretation                  |
> |--------------------------|-------------|----------|--------------|----------|---------------------------------|
> | o3 vs Gemini             | 0.9487      | 0.0230   | 0.9747       | 0.0048   | Very strong agreement            |
> | o3 vs GPT-5.1            | 0.9487      | 0.0230   | 0.9747       | 0.0048   | Very strong agreement            |
> | Gemini vs GPT-5.1        | 1.0000      | 0.0167   | 1.0000       | 0.0000   | Perfect monotonic alignment      |
>
> | Metric                      | Value   | Interpretation                                   |
> |-----------------------------|---------|--------------------------------------------------|
> | Average Kendall Tau         | 0.9658  | Very strong pairwise agreement                   |
> | Average Spearman Rho        | 0.9831  | Near-perfect monotonic alignment                 |
> | Kendall’s W (3 systems)     | 0.8756  | High overall concordance across all three systems |
>
> Using the total scores from each perception setting, we directly compare the induced rankings. As shown in the tables, all three ranked lists exhibit exceptionally high pairwise agreement: Kendall’s τ values of 0.9487, 0.9487, and 1.0000 (all p ≤ 0.023) and Spearman’s ρ values of 0.9747, 0.9747, and 1.0000 (all p ≤ 0.0048). Averaging across pairs yields τ = 0.9658 and ρ = 0.9831, and Kendall’s W = 0.8756 further indicates high overall concordance among the three perception backbones.
>
> The rankings primarily reflect models’ natural language reasoning abilities without being constrained by image-perception. These near-perfect ranking correlations indicate that the global structure of **model performance is preserved across perception modules**, with only minor differences among the lowest-scoring agents. This strong monotonic alignment confirms that the choice of sufficiently strong perception backbone does not materially alter comparative outcomes.
>
> ### **Q4: How many seeds were tested for each game?**
>
> We evaluated each game using 3 seeds, except for Ace Attorney, which was run once.

---

> ### Comment · Reviewer_tEMA · 2025-11-24
>
> Thanks to the authors for the thorough rebuttal.
>
> While I get the point of seeing the difference between having or not having different parts of the scaffolding, this seems like a minor contribution. o3 achieves higher performance in tetris without such scaffolding likely because tetris is an MDP, therefore memory is superfluous, and you are using o3 itself, which is a powerful model that you also use for perception for the other models, so it's not surprising that it performs well in such a game.
>
> My two other main points of concern still standing are the novelty/insights/comparisons with respect to previous existing benchmarks, as well as the huge variances caused by running very few seeds on these games.
>
> ### **Insights and comparisons to previous benchmarks**
> The categorization and comparison with related work does not seem fair.
> For Balrog, only one of the six games (BabyAI text) is a grid-world navigation-based game, the rest of the games, from crafter, nethack, textworld and babaisai are much more complex than that.
> For LMAct, while it's true that some of the environments are saturated, others like chess, atari or cheetah run are far from it.
>
> Many of the insights and qualitative analysis provided in the paper on image perception, reasoning and long context are also already thoroughly discussed in the related benchmarks mentioned.
>
> A **fair** and more extensive comparison with previous benchmarks is warranted in the paper.
>
> ### **On the variance and number of seeds run**
> The huge variances of the benchmark are thus explained by running only 3 seeds per environment, and one for ace attorney. That's 16 seeds in total, which unfortunately is not nearly enough to get a real sense of differences between these models, as the variance is too high. Compare this to LMAct, which runs a total of 600 seeds, and Balrog which runs 255.
> It also seems that for some of the models like o3 and o1, only 1 seed was run on each of the environments.

---

> > ### Author Response · Authors · 2025-11-26
> >
> > We thank the reviewer for the further feedback and questions! Here we provide a more detailed comparison with related works and a discussion on variance.
> >
> > ### **Comparing with existing benchmarks**
> >
> > *Regarding game choice*
> >
> > 1. BALROG primarily focuses on grid-based games, which is also mentioned in BALROG’s paper, where they said “... These limitations are further exemplified in our benchmark, where **grid-based image observations** differ significantly from the natural image-text pairs on which many VLMs are trained…”
> >
> >     Sorry that our previous response used the phrase “grid-based navigation”. A more precise characterization should include the multiple complex game mechanisms involved, grid-based navigation, battle, crafting, etc., as evidenced in their Figure 1. Please also note that only one of the six games (TextWorld) is not grid-based, and it is text-only. We want to highlight that only grid-based visuals are insufficient for a comprehensive VLM benchmark.
> >
> > 2. For LMAct, 3 of their 6 games have been saturated and require a redesign, which is a substantial portion. Among the remaining 3 games, Cheetah Run is text-only, so there are **only 2 games** (Chess and Atari-Phoenix) evaluating VLMs. While both are good gaming environments that reflect basic LLM capabilities, they lack visual and reasoning diversity.
> >    - For Atari-Phoenix, it is limited to 1D movement with a fixed background, where our Super Mario Bros demands 2D movement with path planning under dynamic visuals.
> >    - For Chess, it probes reasoning well but is visually uniform and cannot test perceptual sensitivity (see our discussion on image perception below); this limits its value as a VLM benchmark.
> >
> > 3. Lmgame-bench, on the other hand, also includes action game, which evaluates the models’ capability for **motion detection** and **spatial-temporal reasoning**, and visual-novel game, which evaluate dialogue reasoning in **rich visual scenes**.
> >
> > *Regarding insights*
> >
> > Image perception, reasoning and long-context are indeed discussed in existing benchmarks, but with different analysis and granularity. Here we summarize how related work discuss these failures, and how our insights different from those:
> >
> > 1. Image Perception:
> >     - **BALROG** analyzes image perception failures over two hypotheses: One is  Claude’s VLM strength comes from tasks involving computer usage. The other is grid-based images are out-of-distribution compared to training data, making grid-based games fail. They make these hypotheses without experiments to verify.
> >     - **LMAct** does not explicitly analyze perception failures, since their focus is in-context learning. But we agree that the weakness of perception can be inferred from their results.
> >     - **Lmgame-bench** systematic study of **perception module effectiveness and its relationship with visual complexity**, enabled by our harness. This insight is not covered in related game benchmarks.
> >        - For quantitative analysis, we show in Table 4 that perception helps everywhere except 2048, the game with the smallest board. We correlate perception effectiveness with model reasoning strength and board complexity (Fig. 6), showing that perception module benefits the most to visually complex settings and strong reasoning models.
> >        - For qualitative analysis, Fig. 5 reveals that VLMs extract boards accurately for 4×4 grids but hallucinate or blur elements in larger boards, identifying a resolution limitation that prior work has not analyzed.
> >        - This perception-module discussion is only possible with our isolated harness design, further supporting the significance of employing and analyzing with modular harness.
> >
> > 2. Spatial-Temporal Reasoning:
> >
> >     BALROG focuses on spatial reasoning due to the absence of action games. LMAct includes Atari-Phoenix, although their analysis focuses on in-context learning without case studies. Instead, we provided qualitative analysis on Spatial-Temporal Reasoning in Figure 8, demonstrating failures in estimating object speed, jump height, and timing, which are capabilities that require non-grid-based visual dynamics.
> >
> > 3. Long-context:
> >
> >     Lmgame-bench identifies long-context failures in Ace Attorney (in Figure 9), that the models can find the correct evidence if we remove the noisy irrelevant context, but fail in the original long-context settings. BALROG’s “knowing–doing gap” may relate to this phenomenon, but they do not analyze it or attribute it to long-context reasoning. LMAct discussed working memory limitations by scaling up the number of episodes, but their analysis focuses on **in-context imitation failure, not long-context reasoning over noisy or distractive narrative context**.
> >
> >
> > We acknowledge the contributions of BALROG and LMAct. Our goal is not to diminish their value but to clarify that our benchmark reveals new perception, long-context, and spatial-temporal insights.

---

> ### Author Response · Authors · 2025-11-26
>
> ### **Variance and Number of Runs**
>
> We thank the reviewer pointing out variance and the number of seeds. Here we add an additional analysis on whether our benchmark can separate models.
>
> *Comparison with BALROG*
>
> In BALROG, they run 5,5,5,10,20,25 seeds for their 6 games respectively (instead of 255 in total mentioned by the reviewer), and reports mean ± standard error (SE). Following this method, we also report SE instead of variance, and put the result in Table 18, Appendix I.
>
> To assess whether our benchmark meaningfully separates models, we analyze pairwise separability for three representative pairs chosen from our evaluated models:
> - claude-3-5-sonnet v.s. claude-3-7-sonnet
> - gemini-2-5-flash v.s. gemini-2-5-pro
> - o1-mini v.s. o4-mini
>
> and evaluate separability on each pair of models. These pairs were selected because they represent successive generations from the same model family or models of different sizes within the same generation. Such pairs should have clear capability differences, so a well-designed benchmark should be able to separate their performances.
>
> For each game, we define whether  |Mean1 - Mean2| > 2*(SE1 + SE2) for **Separated** and whether |Mean1 - Mean2| > (SE1 + SE2) for **Weak Separated**.
>
> For Lmgame-bench, here are the results for the three pairs:
>
> | Model                                  | Sokoban     | Tetris       | 2048          | CandyCrash
> |:---------------------------------------|:------------|:-------------|:--------------|:----------------|
> | claude-3-5-sonnet-20241022             | 0.00 ± 0.00 | 14.67 ± 0.67 | 108.21 ± 3.34 | 106.00 ± 30.83  |
> | claude-3-7-sonnet-20250219 (thinking)  | 2.33 ± 0.88 | 16.33 ± 1.33 | 113.35 ± 1.81 | 484.00 ± 30.99  |
>
> | Model                                  | Sokoban     | Tetris       | 2048          | CandyCrash
> |:---------------------------------------|:------------|:-------------|:--------------|:----------------|
> | gemini-2.5-flash-preview-04-17         | 1.67 ± 0.88 | 16.33 ± 1.86 | 106.61 ± 3.05 | 334.67 ± 37.83  |
> | gemini-2.5-pro-preview-05-06           | 4.33 ± 0.33 | 23.33 ± 0.33 | 117.27 ± 3.38 | 416.33 ± 3.93   |
>
> | Model                                  | Sokoban     | Tetris       | 2048          | CandyCrash
> |:---------------------------------------|:------------|:-------------|:--------------|:----------------|
> | o1-mini-2024-09-12                     | 1.33 ± 0.33 | 11.67 ± 0.67 | 113.99 ± 2.15 | 48.00 ± 19.55   |
> | o4-mini-2025-04-16                     | 5.33 ± 0.67 | 25.33 ± 4.91 | 120.63 ± 2.81 | 487.33 ± 114.33 |
>
> | Model                                  | Sokoban     | Tetris       | 2048          | CandyCrash      | avg |
> |:---------------------------------------|:------------|:-------------|:--------------|:----------------|:----------------|
> | Separated                                 | 3 / 3      | 2 / 3        | 0 / 3         | 2 / 3  |  58.3% |
> | Weak Separated                       | 3 / 3      | 2 / 3        | 2 / 3         | 3 / 3  |  83.3% |
>
>
> Here, x/3 means x out of 3 pairs have been Separated or Weak Separated.
>
> Similarly, we present analysis for BALROG on 3 pairs of LLMs:
>
> | Model                 | babyai       |   crafter   | textworld          | babaisai      | minihack      | nle |
> |:----------------------|:-------------|:-------------|:--------------|:----------------|:----------------|:----------------|
> | gpt-4o-mini       | 50.40 ± 4.47 | 15.90 ± 2.05 | 12.25 ± 3.55 | 15.60 ± 2.53 | 10.00 ± 4.74 | 0.00 ± 0.00 |
> | gpt-4o             | 77.60 ± 3.73 | 33.10 ± 2.32 | 39.31 ± 5.24 | 33.66 ± 3.30 | 10.00 ± 4.74 | 0.37 ± 0.37 |
>
> | Model                 | babyai       |   crafter   | textworld          | babaisai      | minihack      | nle |
> |:----------------------|:-------------|:-------------|:--------------|:----------------|:----------------|:----------------|
> | gemini-1.5-flash  | 50.00 ± 7.07 | 20.00 ± 0.74 | 0.00 ± 0.00 | 12.80 ± 2.33 | 5.00 ± 3.45 | 0.00 ± 0.00 |
> | gemini-1.5-pro    | 58.40 ± 4.41 | 30.21 ± 2.86 | 0.00 ± 0.00 | 32.02 ± 3.26 | 5.00 ± 3.45 | 0.37 ± 0.37 |
>
> | Model                 | babyai       |   crafter   | textworld          | babaisai      | minihack      | nle |
> |:----------------------|:-------------|:-------------|:--------------|:----------------|:----------------|:----------------|
> | claude-3.5-haiku  | 52.00 ± 7.07 | 26.36 ± 2.79 | 18.04 ± 5.81 | 8.33 ± 2.52 | 10.00 ± 4.74 | 1.16 ± 0.42 |
> | claude-3.5-sonnet | 68.00 ± 6.60 | 32.73 ± 3.20 | 42.06 ± 5.41 | 37.50 ± 4.42 | 15.00 ± 5.65 | 0.58 ± 0.52 |
>
> | Model                 | babyai       |   crafter   | textworld          | babaisai      | minihack      | nle | avg |
> |:----------------------|:-------------|:-------------|:--------------|:----------------|:----------------|:----------------| :----------------|
> | Separated              | 1 / 3  |  2 / 3 | 2 / 3 | 3 / 3 | 0 / 3 | 0 / 3 | 44.4%  |
> | Weak Separated    | 2 / 3  |  3 / 3 | 2 / 3 | 3 / 3 | 0 / 3 | 0 / 3 |  55.6% |

---

> ### Author Response · Authors · 2025-11-26
>
> These results show our board games are better separable compared to BALROG. Even if we further bring Super Mario Bros and Ace Attorney into consideration, ours are still on par with BALROG on average separability. This further shows the significance of our agent design (harness, prompt optimization, etc.) which effectively reduces the variance with fewer runs.
>
> *Further experiments*
>
> We are aware that SuperMarioBros has a big variance (mentioned in Section 4.4 in our original submission), so we are running more analyze on this game, and also adding additional seeds on o1, o3 and Ace Attorney. We will add the results soon.
>
> Finally, we note that our key results, including harness ablations, data contamination, and model failure analysis, are not affected by this, as they are either confirmed by statistical tests or focusing on qualitative study rather than quantitative.

---

> ### Author Response · Authors · 2025-12-01
>
> ### **Additional Discussion and Experiments on the Relationship between Variance and Number of Runs**
>
> As noted above, we conducted additional evaluations for o1 (with harness), o3 (with harness), Ace Attorney, bringing each of them to three runs, consistent with the other environments. We also analyze the standard error of Super Mario Bros under more runs.
>
> Here are the results for **o1 and o3** after adding two additional runs:
>
> | Model    | Sokoban   |  SMB   |  Tetris   | 2048    | Candy Crush | Ace Attorney |
> |-----|------|----|-----|-----|---|------|
> |  o1 (with harness) | 2.3 ± 0.3 | 867.7 ± 78.8 | 43.0 ± 4.9 | 122.7 ± 3.9 | 403.0 ± 128.1| 10.7 ± 2.7 |
> | rank change | same | same | same | -1 | +3 | same |
> |  o3 (with harness)  | 8.7 ± 1.3 | 2267.7 ± 704.3 | 57.0 ± 15.0  | 126.2 ± 1.9  | 636.7 ± 16.7 | 10.7 ± 2.7 |
> | rank change | same | same | same | +1 | same | same |
>
> After adding the new runs for o1 and o3, we recomputed the full-model ranking among all 13 models listed in Table 1. As shown above, **the ranking of o1 and o3 remain highly consistent with those reported in the main paper.**
>
> For **Ace Attorney**, we add two additional runs and show the result in the below table. The table omits the five models whose scores remain 0 under all settings, and also omits the ±SE values if only one run is available (because the model has been deprecated and no further runs can be collected).
>
> | Model | With Harness - New | With Harness - Original |  Without Harness - New | Without Harness - Original |
> |:--------|:------|:-------|:------|:-----|
> | claude-3-5-sonnet      | 2.0  | 2.0  | 1.0 | 1.0 |
> | o4-mini       | 2.7 ± 0.7  | 4.0  | 2.7 ± 0.7 | 2.0 |
> | gpt-4.1    | 3.3 ± 0.7  | 2.0  | 0.0 ± 0.0 | 0.0 |
> | gemini-2.5-flash         | 4.7 ± 0.7  | 4.0  | 2.3 ± 0.9 | 1.0 |
> | claude-3-7-sonnet  | 7.0   | 7.0  | 3.0 | 3.0 |
> | gemini-2.5-pro           | 7.7 ± 0.3  | 7.0  |  6.7 ± 1.3 | 8.0 |
> | o1        | 10.7 ± 2.7 | 16.0 | 2.5 ± 0.5 | 3.0 |
> | o3       | 10.7 ± 2.7 | 16.0 | 8.0 ± 0.0 | 8.0 |
>
> As shown in the table, Ace Attorney exhibits low standard errors and strong separability across models. To quantify ranking stability after adding two more runs, we compute Kendall Tau and Spearman Rho between the original 1-run rankings and the new 3-run mean rankings.
> * Without harness: Kendall Tau = 0.914 (p = 5.62e-04), Spearman Rho = 0.972 (p = 2.66e-06).
> The updated rankings remain **almost identical** to the original ones, showing that even a single run provides a reliable ranking.
> * With harness: τ = 0.947 (p = 4.79e-05), ρ = 0.984 (p = 1.41e-09).
> Ranking stability is even stronger with the harness: most models keep exactly the same order, and the correlation with the original ranking is **near perfect**.
>
> We are also aware that **SuperMarioBros** has a big standard error (mentioned in Section 4.4 in our original submission). To investigate this, we increased the number of evaluation runs and observed the following:
>
> | Model     | SMB 5 Runs | SMB 10 Runs |
> |-------------|----------|----------|
> | gpt-4.1   | 1597.4 ± 652.5   | 1391.0 ± 319.5  |
> | o4-mini    | 943.4  ± 200.7 | 1080.3 ± 109.3 |
> | gemini-2.5-pro    | 1368.2 ± 155.5 | 1342.5 ± 95.9 |
> | gemini-2.5-flash      | 1304.6 ±105.4  | 1236.8 ± 94.2 |
>
> **With more runs, the standard error decreases substantially across all models**. In practice, users who need higher distinguishability can increase the number of runs, which reliably reduces noise and results in more stable comparisons. We will perform 10 runs for this game for further evaluations, and add the above new results and discussions to the final version of the paper.
>
> #### Summary of variance analysis
>
> Overall, the variances in our games are low enough to achieve on-par or slightly better separability compared with related work. Adding extra runs leaves the overall model rankings largely unchanged, confirming ranking stability. For Super Mario Bros, adding more runs effectively reduce variance and can sharpen model separation.
>
> We also want to emphasize that our key results and insights, including harness design, data contamination analysis, and model-failure analyses, are not affected by this discussion of variance, since they are either already supported by statistical significance tests or rely on qualitative evidence.

---

### Official Review · Reviewer_zKTd · 2025-10-29

**Soundness:** 3
**Presentation:** 2
**Contribution:** 3
**Rating:** 8
**Confidence:** 3

**Summary:**

The paper introduces LMGame-Bench, a new benchmark designed to evaluate large language
and vision–language models (LLMs/VLMs) through six popular video games: Super Mario Bros.,
Tetris, Sokoban, 2048, Candy Crush, and Ace Attorney. The benchmark uses additional modular
gaming harness composed of Perception, Memory and Reasoning modules, which are used to
assess specific model capabilities, and allow the models to perform at the level of human
players. The benchmark also takes advantage of the DSPy’s SIMBA optimizer to standardize
the prompts.
LMGame-Bench allows for a controlled evaluation of LLMs on video games and offers a
comprehensive evaluation of 13 state-of-the-art models, while relying on robust techniques such
as prompt standardization and contamination detection.

**Strengths:**

1. Gaming Harness: The design of the gaming harness is highly effective and it represents a
robust method for isolating and accurately evaluating specific model capabilities and skills.
2. Data Contamination Detection: The inclusion of an explicit mechanism for detecting data
contamination is a significant strength
3. Prompts Standardization: The decision to standardize prompts by leveraging the DSPy
framework is highly commendable. This approach ensures consistency and reproducibility
across experiments.

**Weaknesses:**

The primary weakness lies in the overall presentation and clarity of the paper. I recommend
repositioning the Related Work section to immediately follow the Introduction. This structural
change would more effectively contextualize the work and highlight the paper's novel
contributions earlier. Also, given that this is a benchmark paper, the explanation of the Metrics
(specifically the Raw and Aggregated Scores) should be expanded, possibly reserving one
subsection just for this point.

Also, there are some typos in the main paper. For example line 53:
"To address this issue and also enable controlled evaluation, we enriches our evaluation
settings by developing gaming harness". "we enriches" should be "we enrich".

And line 425:
"Super Mario Bros. is excluded" should be "Super Mario Bros is excluded".

**Questions:**

Regarding the gaming harness. It is not clear to me if you can "toggle on or off" them selectively.
For example, is it possible to only activate the Perception Module without activating the Memory
Module, or viceversa?

---

> ### Author Response · Authors · 2025-11-23
>
> We thank reviewer zKTd for the constructive feedback. We have revised the manuscript to address typos and improve clarity. In particular, we moved the related work section to highlight our work’s distinction with existing work and also expanded the explanation of Metrics in Section 3.1.3.
>
> To further aid interpretation on different models’ gaming performance, we elaborate procedural progress scores in Appendix A with detailed explanation on each metric (attached below).
>
> detailed design of procedural progress scores:
>
> - Obstacle-level progress for **Super Mario Bros.**, measuring the total number of enemies, pipes, and gaps successfully passed, which captures a model’s ability to navigate structure rather than just distance traveled. Note that none models have entered the wrap zone in level 1-2 (only one human evaluator entered).
> - Box-level progress for **Sokoban**, reporting level number and boxes successfully placed, which better reflects puzzle-solving competence.
> - Tile-based progression for **2048**, report the max-tile milestones (e.g. 64/128/256/512), providing a clearer notion of strategic depth.
> - Level/step-based metrics for **Ace Attorney**, anchored in the procedural steps required to complete investigative or courtroom actions.
> - Cleared lines for **Tetris**, which reflect how many lines have been cleared. Most models can’t clear any lines, with best models only clear 1-2 lines.
> - Cleared candies in **Candy Crush**, the reported values are the same as raw scores, both are the number of cleared candies.
>
> For the question on gaming harness, yes we can toggle on or off each module selectively. While Table 1 reports performance when modules are uniformly enabled or disabled, Tables 2 and 4 show the effects of enabling only the perception or only the memory module, illustrating how the effectiveness of each harness component varies across games and models.

---

> > ### Author Response · Authors · 2025-11-26
> >
> > Dear Reviewer zKTd,
> >
> > Thank you for your recognition of our paper. We would like to follow up to see if you have any further questions we can address.
> >
> > Best,
> >
> > Authors

---

### Official Review · Reviewer_2PLk · 2025-10-31

**Soundness:** 4
**Presentation:** 4
**Contribution:** 2
**Rating:** 6
**Confidence:** 4

**Summary:**

This paper introduces a new benchmark for evaluating LLMs on game-playing tasks, testing 13 state-of-the-art models using a direct screenshot-to-action setup. The authors find that these models perform poorly, often close to random, highlighting their weaknesses in visual perception and long-horizon decision-making. The study also explores where and how these models fail as task difficulty increases and demonstrates the effective use of the proposed harness.

**Strengths:**

1. A new benchmark consisting of complex goal-driven games is introduced in this study.
2. An extensive suite of models is evaluated, covering 13 state-of-the-art architectures.
3. The problem statement and the experimental framework are well designed and presented.
4. The authors perform detailed and consistent evaluations across difficulty levels, revealing how and where models fail.

**Weaknesses:**

While the work is interesting and systematically executed, many of its findings align with prior studies that have already established similar limitations of LLMs and explored methods to overcome them (e.g., Chain-of-Thought reasoning, embedding API calls, or memory modules/database access). The novelty and contribution of this work, therefore, feel limited unless the authors can better justify what new insights their benchmark offers.

Additionally, while the authors show that adding different modules enhances performance, they do not explain why these modules lead to improvement. If such reasoning is provided, the authors should indicate the corresponding line numbers.

**Questions:**

1. How does this benchmark fundamentally differ from existing benchmarks that also test LLMs or VLMs on interactive or game-based tasks? Specifically, how are these selected games different in terms of difficulty and multi-modality from the other existing benchmarks/games?

2. Is there any difficulty level or stage where even the module-based models fail to improve? The authors should clarify how far these modules can enhance performance and at what point their effect saturates or diminishes.

---

> ### Author Response · Authors · 2025-11-23
>
> We thank reviewer 2PLk for the constructive feedbacks and acknowledging our benchmark design, experiment design and presentation! We answer each of your question and concerns below.
>
> ### **W1: Need elaboration on why harness modules lead to improvement.**
>
> We have provided this in Section 4.2 Image Perception (Line 399 - 404) and Reflection on Failures (Line 406 - 412), with a summary here:
>
> - Perception module: Current VLMs’ perception ability are limited. For board games, they weren’t able to extract game state from images, so converting images to text would greatly improve the performances.
>
> - Memory module: We observed that non-reasoning models improve substantially with memory support, because these models frequently fall into loops of repeating the same invalid actions without the memory module.
>
>
>
> ### **Q1: How does this benchmark fundamentally differ from existing benchmarks, especially in terms of difficulty and multi-modality.**
>
> We argue Lmgame Bench is a more comprehensive and systematic game benchmark that better distinguishes model capabilities and is hard to saturate.
>
> - on **difficulty and multi-modality**:  Most prior benchmark focused on text-only games (e.g. GameArena, GameBench, Kaggle AI Chess Tournament, etc.), while there are 3 related works covering multi-modal games (i.e., BALROG, LMAct and VideoGameBench).  Among those, BALROG concentrates on grid-based navigation, while we cover board, puzzle games, action games, and a detective visual novel adventure game, giving richer genre diversity and more human-like skill coverage. We also select challenging games with built-in difficulty ladders or unbounded scoring metrics, making our games hard to saturate. In contrast, many games from LMAct (e.g., Tic-Tac-Toe, GridWorld, Crosswords) are saturated and require a redesign. Meanwhile, games from VideoGameBench are too hard that most models make zero progress. Therefore, compared with related work, our benchmark evaluated diverse multi-modal capabilities with well-designed difficulty that is hard to saturate while still being able to distinguish models.
>
> In addition, our benchmark is also novel in terms of agentic design and insights:
>
> - on **agentic design**, our harness can ablate model capability. Unlike benchmarks that fix scaffoldings or only vary input modality, we systematically analyze modular harness effects (Table 2, Appendix B). The results are non-trivial: memory (history + reflections) helps non-reasoning models more than reasoning models on board games, and image perception is the main bottleneck for many models in almost all games except 2048, where visuals are simplest.
>
> - on **gaming performance analysis and insights**, we move beyond gaming scores to capability attribution. We introduce a novel latent ability decomposition analysis to map games to core LLM skills (Figure 3b), revealing connections from games to math, coding, physics, etc. For failure case analysis, we give concrete examples (Figure 5,7,8,9) when summarizing models’ failure cases. We highlight the model's weakness on perception by varying visual complexity of games (Figure 5), and explain long-context challenges by ablate with less-context in Figure 9. In our newly added Figure 6, we also provide a novel analysis of perception-module effectiveness as a function of both visual complexity and model reasoning capability, which is an angle not covered in prior work.
>
>
> ### **Q2: When harness modules saturates or diminishes.**
>
> The effect of each module would depend on the capabilities of base models. In practice, we have not yet reached a point where these modules saturate.
>
> - In principle, **perception module** would saturate only when all VLMs can perfectly extract game states even without any harness; however, current VLMs are still far from this ideal. We verify this claim by trying the latest models (Gemini-3-Pro-Preview and GPT-5.1) on the perception tasks listed in Figure 5. Neither model achieves correct perception, although their performances are more accurate if compared with the Gemini-2.5-Pro in Figure 5. **Overall, current models still lack sufficiently reliable perception to bypass the perception module**.
> - **Memory modules** would be less effective for board games if the reasoning models are strong enough to directly infer the best solution. In contrast, memory would be **necessary for Super Mario Bros and Ace Attorney**, since they require previous game history to infer motion dynamics and remember previous facts and evidence.

---

> > ### Author Response · Authors · 2025-11-26
> >
> > Dear Reviewer 2PLk,
> >
> > We have submitted our rebuttal to your questions and would like to follow up to ensure that our responses have addressed your concerns. We would be more than happy to discuss further if anything remains unclear or any further clarification is needed.
> >
> > Best,
> >
> > Authors

---

> ### Comment · Reviewer_2PLk · 2025-11-26
> **response to authors**
>
> Thank you to all the authors for addressing my questions. I would like to kindly reiterate my first point, which I believe remains unaddressed.
>
> While the work is interesting and systematically executed, many of its findings align with prior studies that have already established similar limitations of LLMs and explored methods to overcome them (e.g., Chain-of-Thought reasoning, embedding API calls, or memory modules/database access). The novelty and contribution of this work, therefore, feel limited unless the authors can better justify what new insights their benchmark offers.
>
> In my understanding, the authors have not explained how their method is different from these existing methods (such as Chain-of-Thought reasoning, embedding API calls, or memory modules/database access) that are widely used in various domains and applications. I believe this is important and should be present in the paper.

---

> > ### Author Response · Authors · 2025-11-27
> >
> > In addition to modular harness design to identify and overcome LLMs limitations, we have presented additional methods to address different challenges in game-as-an-eval:
> > - We address data contamination issues in Section 3.2.2 by proposing data contamination detection and mitigation methods.
> > - We address prompt sensitivity issues in Section 3.2.3 by proposing a standardized prompt optimization pipeline as part of the benchmark design.
> >
> > To better address your first question, we summarize each point as well as our benchmark design (game choice, vision modality, difficulty level) in the following sections.
> >
> > *Data Contamination Study and Mitigation*
> >
> > We notice that game-as-an-eval repurposes gaming environments for evaluation might have data contamination issues. To address the challenge, we make the following quantitative analysis in the paper along with mitigations.
> >
> > **Vision-Level Contamination Study**: Super Mario Bros (SMB)
> > 1. Method: we shuffle the first 10 frames of SMB 1-1 and ask models to reconstruct the original order to **test whether they can successfully recall visual sequences if there are any memorization issues**.
> > 2. Takeaways: visual memorization data contamination is not an issue for the game. Models perform poorly on this task, recovering only very coarse structure (e.g., first/last frames). Their reconstruction ability does not correlate with actual SMB gameplay performance, indicating that models rely on local perception rather than memorized frame sequences. Visual contamination is therefore negligible. For detailed metrics and analysis, see Appendix D.1.
> >
> > **Text-Level Contamination Study**: Ace Attorney
> > 1. Method: we compare model-generated evidence lists with publicly available game plot transcripts, then **test whether similarity predicts gameplay performance**. We also apply mitigation (name masking, paraphrasing, and reasoning-focused prompting) to suppress direct recall.
> > 2. Takeaways: before mitigation, data contamination is an issue. Similarity between the generated scripts and the public script is strongly correlated with gaming performance, revealing clear text-level contamination. After mitigation, this relationship disappears, and performance reflects reasoning quality rather than memorized content. Text contamination is thus controllable and removable. You can checkout Appendix D.2 for full results and analysis.
> >
> > *Prompt Sensitivity Analysis*
> >
> > Another challenge from game-as-en-eval is model performance varies if we use empirically optimized prompts as demonstrated in Table 13. To address the challenge, we apply standardized prompt optimization to all games using DSPy to reduce performance variance introduced by manual design choices.
> >
> > We provided an prompt optimization effectiveness study in Appendix E.3, which shows prompt optimization can effectively reduce model performance variances among manually designed prompt templates across 3 game runs by 33.8% to 63.5%.
> >
> > *Benchmark Design*
> >
> > In addition, the benchmark designs, game choices, and difficulty levels are different. We make a more extensive comparison with the previous benchmark, we report distinctions among different benchmark designs in the table below.
> >
> > | Feature | lmgame-bench (ours) | LMAct | BALROG |
> > |----------------------------------------------|----------------------------|----------------------------------|--------------------------------------|
> > | Modular agentic harness | Yes | No (not core design) | No (not modularized) |
> > | Visual modality diversity | platformer, visual novel, action, grid | action, grid | grid |
> > | saturation status | 0/6 | 3/6 | 0/6 |
> > | Navigation games | Yes | No | Yes |
> > | Action games | Yes | Yes | No |
> > | Board games | Yes | Yes | Yes |
> > | Visual-novel and dialogue games | Yes | No | No |
> > | Multi-player support | Yes* | No | No |
> >
> > *Lmgame-bench added multi-player support and the game, Texas Hold’em, in the rebuttal as reported in Appendix G.

---

### Official Review · Reviewer_uZi7 · 2025-11-01

**Soundness:** 3
**Presentation:** 3
**Contribution:** 2
**Rating:** 4
**Confidence:** 4

**Summary:**

This paper introduces LMGame-Bench, a benchmark for evaluating large language models (LLMs) and vision-language models (VLMs) on six popular video games (Super Mario Bros., Tetris, Sokoban, Candy Crush, 2048, Ace Attorney) via a unified Gym-style API. Unlike prior game benchmarks that entangle multiple skills, LMGame-Bench uses a modular harness (perception, memory, reasoning modules) to isolate specific capabilities, supports both scaffolded and unscaffolded evaluations, and enhances robustness through data contamination mitigation and prompt standardization. Evaluations of 13 state-of-the-art models show the benchmark effectively discriminates performance (o3 and o1 lead), reveals correlations between games and core LLM capabilities (e.g., Sokoban aligns with math/coding, Ace Attorney with language understanding), and identifies model limitations in visual state extraction, spatiotemporal reasoning, and long-context processing.

**Strengths:**

1. Modular Harness Design: Addresses a key limitation of prior game benchmarks (entangled skills) by enabling selective activation of perception, memory, and reasoning modules. This allows fine-grained diagnosis of model strengths/weaknesses (e.g., separating perception failures from planning gaps) that was previously unachievable.
2. Rigorous Experimental Design: Evaluates 13 models across 6 diverse games (platformer, puzzle, narrative) with standardized metrics (progression/long-horizon rewards) and statistical validation (paired-sample t-tests, Glass’s δ, coefficient of variation). Results are consistent and reproducible, with detailed ablation studies for harness modules.
3. Insights for Model Improvement: By identifying specific failure modes (e.g., VLMs struggle with board state extraction from images, non-reasoning models lack self-correction), the paper guides concrete advancements in model architecture and agentic design.

**Weaknesses:**

1. Limited Game Diversity: While the 6 games cover 3 genres, they lack representation of real-time strategy (RTS), open-world, or multiplayer games—domains that test collaboration, dynamic resource management, or complex opponent adaptation. This limits the benchmark’s generalizability to broader game-based agentic tasks.
2. Computational Cost Opacity: While the paper mentions high computational costs (Appendix B.4), it does not provide concrete guidance for scaling evaluations (e.g., cost-saving strategies beyond vague suggestions like "bounding trajectories"). This may limit accessibility for smaller research teams.
3. Perception Module Efficacy: For games like Super Mario Bros., how accurate is the textual representation generated by the perception module in capturing dynamic spatiotemporal cues (e.g., enemy speed, jump physics)? Could gaps in this representation explain why the harness provides limited gains for this game?

**Questions:**

See Weaknesses.

---

> ### Author Response · Authors · 2025-11-23
>
> We thank reviewer uZi7 for the constructive feedbacks and acknowledging our modular harness design, experiment design and failure case analysis! We answer each of your question and concerns below.
>
>
> ### **W1: Limited Game Diversity. While the 6 games cover 3 genres, they lack representation of real-time strategy (RTS), open-world, or multiplayer games—domains that test collaboration, dynamic resource management, or complex opponent adaptation.**
>
> We acknowledge that our benchmark does not cover every genre, specifically excluding complex open-world simulators (discussed in Appendix H). However, our current selection optimizes for efficiency and differentiating model capability. the core capabilities demanded in open-world games (exploration, long-horizon planning, visual perception, spatial reasoning) are already **systematically evaluated** in our existing tasks (e.g., Sokoban, Super Mario Bros., Ace Attorney), but without the computational overhead and confounding variables of expansive simulations.
>
> We further extend our analysis to **multi-agent** settings via Texas Hold’em (Appendix G). Using this as a diagnostic for strategic behavior under uncertainty, our results reveal that while prompting significantly shapes playstyles (e.g., aggressive vs. passive), underlying model biases persist.
>
> ### **W2: Guidance on cost-saving strategies.**
>
> In Appendix B4, we have summarized the lowest cost scenario in the “Without Harness” setting, which is the minimal requirement for evaluating our benchmark. Here we list two **actionable strategies** for reducing cost.
>
> *(1) Disable or reduce memory for strong reasoning models.*
>
> For advanced reasoning models (e.g., o3, Gemini-2.5-Pro), our results show that they benefit **far more from the perception module than from the memory module** on board games (see Table 2 and Table 4). Users can therefore reduce the memory context length or disable the memory module entirely to cut token usage.
>
> By contrast, non-reasoning models gain more from harness modules, but their per-token inference cost is typically much lower, so the added harness overhead is less prohibitive.
>
> *(2) Reusing perception frames for Ace Attorney.*
>
> When searching for evidence, the same frames are often revisited multiple times (e.g., the evidence page reappears after each item is examined, and a location scene remains unchanged after searching it). Users can **cache all previously seen scenes** and only invoke the perception module for new frames. This avoids redundant perception calls and reduces cost without affecting correctness.
>
> ### **W3: Perception module efficiency for Super Mario Bros.**
>
> Super Mario Bros introduces rich spatiotemporal dynamics and stands as one of the most challenging tasks in Lmgame-Bench, requiring both spatial reasoning and motion understanding. The perception module for Super Mario Bros extracts **static spatial information** (i.e. object types, coordinates, and estimated distances between objects). It does not attempt to infer dynamic attributes such as enemy speed, jump physics, or momentum. The model must **reason spatiotemporal dynamics on its own**, which could explain why the perception module offers only limited gains.
>
> To quantify how accurate the perception outputs are, we manually evaluated coordinate accuracy, and found that 70% of coordinate predictions were fully correct. With our perception module on Super Mario Bros, we use GPT-5.1-as-a-judge to measure the failure causes of 300 CoT on Gemini-2.5-Flash game traces from perception module using o3 and gemini-3:
>
> | Failure Type     | Without Perception |  With o3-Perception | With Gemini-3-Perception |
> |------------------------------|--------------------|--------------------|--------------------|
> | No Failures             | 31.9%               |  40.5%  | 47.5% |
> | Image Perception              | 56.9%              | 37.6%  | 33.6% |
> | Spatial Reasoning  | 39.6%              | 40.8% | 40.5% |
> | Motion       | 20.1%              | 20.2% | 21.0% |
>
>
> The results show employing the perception module reduces image-perception failures by more than 20 percent, with Gemini-3 more effective than o3. Combined with the evidence in Table 6, where the perception module consistently yields positive average score improvements, these findings confirm that the perception module meaningfully improves visual grounding.
>
>
> Despite these improvements, the overall failure rates with the perception module remain non-negligible due to the inherent complexity of Super Mario Bros. This challenge limits the module’s impact, leading to smaller performance gains compared with other games where static perception features are more predictive.

---

> ### Author Response · Authors · 2025-11-26
>
> Dear Reviewer uZi7,
>
> We have responded to your questions in our rebuttal and want to check whether our answers fully addressed your concerns. If there are any remaining issues or clarifications needed, we would be glad to discuss further.
>
> Best,
>
> Authors

---

### Official Review · Reviewer_GmFK · 2025-11-12

**Soundness:** 2
**Presentation:** 3
**Contribution:** 3
**Rating:** 6
**Confidence:** 3

**Summary:**

## Summary
This paper introduces **LMGame-Bench**, a modular, Gym-style benchmark constructed around six popular games (Super Mario Bros., Tetris, Sokoban, Candy Crush, 2048, and Ace Attorney). A core design feature is a **toggleable “gaming harness”** (comprising perception, memory, and reasoning modules) that isolates distinct capabilities and expands performance headroom. Complementing this, the benchmark integrates **contamination checks** and **prompt standardization** to reduce evaluation variance. Experiments conducted on 13 state-of-the-art models demonstrate that the harness significantly improves the benchmark’s ability to discriminate between models while uncovering key failure modes—including limitations in visual state extraction, spatiotemporal control, self-reflection, and long-context reasoning. The paper further employs **correlation and low-rank decomposition analyses** to link game-specific performance to broader clusters of LLM capabilities.


## Strengths
1. **Original and Impactful Contribution**: LMGame-Bench addresses a critical limitation of existing game benchmarks—their tendency to entangle multiple skills—by introducing a modular harness that isolates distinct LLM/VLM capabilities (perception, memory, reasoning). This design enables fine-grained diagnosis of models’ strengths and weaknesses, making it a valuable tool for guiding model development.
2. **Rigorous Benchmark Design**: The benchmark strikes a balance in difficulty (avoiding both premature saturation and excessive hardness) and covers diverse game genres (platformers, puzzles, narrative games), ensuring it effectively discriminates between state-of-the-art models. The inclusion of contamination mitigation (e.g., entity masking, paraphrasing) and prompt standardization (via DSPy’s SIMBA optimizer) further enhances its robustness—a key prerequisite for reliable LLM evaluation.
3. **Comprehensive Experimental Design**: The evaluation of 13 models (under both harnessed and unharnessed settings) combines quantitative methods (paired-sample t-tests, Glass’s δ effect sizes, correlation analysis) and qualitative insights (failure mode analysis). By linking game performance to core LLM capabilities through low-rank factorization, the work delivers actionable insights beyond mere raw score comparisons.
4. **Candid Limitation Identification**: The paper openly highlights models’ weaknesses—such as challenges in visual state extraction, spatiotemporal reasoning, and long-context retrieval—and proposes concrete directions for improvement. This avoids the common pitfall of overemphasizing benchmark performance without addressing actionable gaps in model capability.


## Weaknesses
1. **Lack of Quantitative Support for Qualitative Analysis**: While Section 3.2 presents a qualitative analysis of model failures, it lacks accompanying quantitative validation. For instance, the paper could address this gap by using LLMs to annotate a subset of game trajectories, identifying the key failure reasons (e.g., “incorrect visual state parsing” vs. “poor long-horizon planning”) for each episode, and calculating the statistical proportion of failures attributed to each cause. Conducting this ablation experiment on a small scale would significantly enhance the credibility of the benchmark’s diagnostic claims.
2. **Limited Diversity in Evaluation Metrics**: The paper relies primarily on raw scores to evaluate model performance. While raw scores effectively reflect game progress from a human perspective, they provide limited procedural feedback for LLMs—failing to capture nuanced capabilities like reaching critical game nodes (e.g., accessing a bonus area in Super Mario Bros.) or acquiring key information (e.g., identifying critical evidence in Ace Attorney). Incorporating such procedural metrics would offer a more holistic view of model capabilities.


## Questions
Q1: Could the authors supplement the qualitative analysis with small-scale quantitative validation (e.g., trajectory annotation and failure reason statistics) as suggested in Weakness 1?
Q2: Could the authors propose additional, more diverse evaluation metrics—including procedural feedback indicators—to better capture nuanced model capabilities, as outlined in Weakness 2?

**Strengths:**

See Summary

**Weaknesses:**

See Summary

**Questions:**

See Summary

---

> ### Author Response · Authors · 2025-11-23
>
> We thank reviewer GmFK for the constructive feedbacks and recognizing our benchmark design, experiment design and model limitation analysis! We answer each of your question and concerns below.
>
>
> ### **W1/Q1: Could the authors supplement the qualitative analysis with small-scale quantitative validation (e.g., trajectory annotation and failure reason statistics) as suggested in Weakness 1?**
>
> We conducted small-scale **quantitative validation** using GPT-5.1 to annotate failure cases across diverse genres: action game (Super Mario Bros.), visual-novel reasoning game (Ace Attorney), and board game (Sokoban). This complements the qualitative analysis in our paper.
>
> *Failure Case Analysis for Action Game (Super Mario Bros):*
>
> We sampled 700 chain-of-thought reasoning GPT-4o and Gemini-2.5-Flash (without perception module but with memory module), each with the current image, the model’s thought and action, and the resulting next-state image. GPT-5.1 judged (1) whether the thought correctly described the initial state and (2) whether the expected outcome matched the true environment transition.
>
> Failure categories:
> - **Image Perception Failure**: missing enemies, hallucinating objects, wrong object types or positions.
> - **Spatial Reasoning Failure**:  incorrect distances, relative locations, or direction of threats.
> - **Motion Failure**: misjudged speed, incorrect estimation of jump height / interaction dynamics.
>
>
> Results:
>
> | Failure Type                    | Overall Proportion |
> |------------------------------|--------------------|
> | No Failures             | 34.8%          |
> | Image Perception              | 54.0 %              |
> | Spatial Reasoning  | 43.4%              |
> | Motion       | 17.5%              |
>
> Thus, models correctly analyze states only 34.8% of the time. Image perception (54.0%), spatial reasoning (43.4%) and motion detection (17.5%) are the majority of failure reasons, revealing core limitations of current VLMs.
>
> *Failure Case Analysis for Visual Novel Adventure Game (Ace Attorney):*
>
> We further use GPT-5.1 as an LLM judge to **annotate a subset of Ace Attorney–style game trajectories**, identifying the key failure reasons behind incorrect behaviors.
>
> Failures categories:
> - **Memory Confusion**: the model acts on an incorrect or drifting internal state
> - **Reasoning Error**: the model draws the wrong logical inference
> - **Reasoning Execution Failure**: the model knows the correct reasoning but fails to act on it
> - **State Perception Error**: the model misjudges the UI/game state and performs an invalid action.
>
> Aggregating errors across Claude 4 Sonnet, Gemini 2.5 Flash, and GPT-5 yields:
>
> | Error Type                    | Overall Proportion |
> |------------------------------|--------------------|
> | Memory Confusion             | 6.7%               |
> | Reasoning Error              | 33.3%              |
> | Reasoning Execution Failure  | 10.0%              |
> | Image Perception Error       | 50.0%              |
>
> Overall, image perception mistakes dominate (50% of errors), with reasoning errors also common, while memory drift and execution failures are comparatively rare.
>
> *Failure Case Analysis for Board Game (Sokoban):*
>
> Finally, GPT-5.1 annotated Sokoban failures with the following categories:
> - **Perception Failure**: Failure to extract the game state from the image. This is only valid without the perception module.
> - **State Hallucination Failure**: The game state appeared in the thought is different from the actual game state. This is only valid with the perception module.
> - **Long Horizon Planning Failure**: invalid or hallucinated multi-step plan
> - **Local Failure**: impossible short-term action (e.g., pushing into a wall)
>
> Results:
>
> | Failure Type                    | Without Perception Module | With Perception Module|
> |------------------------------|--------------------|-------------------|
> | No Failures             | 2.6%               |  53.1%   |
> | Perception             | 96.1%              |   -                   |
> | State Hallucination |  -                  |  9.8%  |
> | Long Horizon         | 30.8%              |  42.0% |
> | Local Failure          | 57.7%              |    3.7%   |
>
> Without a perception module, failures are overwhelmingly due to perception-related failures (96.1%). Once enabled, most steps exhibit no failures (53.1%), while remaining failures concentrate into **long-horizon planning**, consistent with Sokoban’s inherent complexity.

---

> ### Author Response · Authors · 2025-11-23
>
> ### **W2/Q2: More diverse evaluation metrics, including procedural feedback indicators, to better capture nuanced model capabilities.**
>
> Our main paper uses raw scores since procedural progress scores sometimes are coarse and insufficient to distinguish models. But we agree that procedural progress scores can provide critical information for game progress, especially for understanding where and why models fail. We added the procedural progress score to the revised paper (**Appendix A, Table 3**).
>
> Here are the detailed design of procedural progress scores:
>
> - Obstacle-level progress for **Super Mario Bros.**, measuring the total number of enemies, pipes, and gaps successfully passed, which captures a model’s ability to navigate structure rather than just distance traveled. Note that none models have entered the wrap zone in level 1-2 (only one human evaluator entered).
> - Box-level progress for **Sokoban**, reporting level number and boxes successfully placed, which better reflects puzzle-solving competence.
> - Tile-based progression for **2048**, report the max-tile milestones (e.g. 64/128/256/512), providing a clearer notion of strategic depth.
> - Level/step-based metrics for **Ace Attorney**, anchored in the procedural steps required to complete investigative or courtroom actions.
> - Cleared lines for **Tetris**, which reflect how many lines have been cleared. Most models can’t clear any lines, with best models only clear 1-2 lines.
> - Cleared candies in **Candy Crush**, the reported values are the same as raw scores, both are the number of cleared candies.

---

> ### Author Response · Authors · 2025-11-26
>
> Dear Reviewer GmFK,
>
> We have submitted our rebuttal to your questions and would like to follow up to ensure that our responses resolved your concerns. Please let us know if any points remain unclear and we would be more than happy to discuss further.
>
> Best,
>
> Authors

---

### Author Response · Authors · 2025-12-03

## **Discussion summary**

We would like to thank all the reviewers for the constructive and valuable feedback, which we will leverage to improve this work. We are particularly encouraged by the positive comments from reviewers, including:

- **Benchmark Design**
    - **Regarding game choice**: "diverse" (uZi7), “a balance in difficulty” (GmFK)
    - **Regarding harness**: “fine-grained diagnosis” (GmFK), “addresses a key limitation of prior game benchmarks (entangled skills)” (uZi7), “isolating and accurately evaluating specific model capabilities and skills” (zKTd)
    - **Regarding data contamination and prompt standardization**: “significant strength … ensures consistency and reproducibility” (zKTd), “enhances robustness” (GmFK, uZi7), “data contamination detection is interesting, and so is the attempt at standardizing the prompting with DSPy” (tEMA)
- **Evaluation**
    - **Overall comments**: “comprehensive” (zKTd), “detailed and consistent” (uZi7, 2PLk), detailed ablation studies for harness modules (uZi7)
    - **Regarding failure case analysis**: “revealing how and where models fail” (2PLK), “guides concrete advancements in model architecture and agentic design” (uZi7)
    - **Regarding correlation analysis**: “low-rank factorization … actionable insights beyond raw score comparisons” (GmFK), “reveals correlations between games and core LLM capabilities” (uZi7)
- **Writing**
    - “well designed and presented” (2PLK)

We also summarize our final remarks in response to each review:

### Reviewer GmFK
*Q1: Supplement the qualitative analysis with small-scale quantitative validation.*

We conducted small-scale quantitative validation and validated the key weakness of image perception, spatial/temporal reasoning, and long-context planning.

*Q2: More diverse evaluation metrics.*

We added the procedural progress score to the revised paper.

### Reviewer uZi7
*W1: Limited Game Diversity.*

We extended our analysis to multi-agent settings via Texas Hold’em.

*W2: Guidance on cost-saving strategies.*

We further listed two actionable strategies for reducing cost.

*W3: Perception module inefficiency for Super Mario Bros.*

We explained that the perception module is mainly for object detection, and models need to reason spatiotemporal dynamics on their own. We further showed that after adding the perception module, the distribution of failure cases shifts.

### Reviewer 2PLk
*W1: Need elaboration on why harness modules lead to improvement.*

We pointed to Section 4.2.

*Q1: How does this benchmark fundamentally differ from existing benchmarks?*

Regarding game choice, we provided a comparison with related works by a table.

Regarding agent design, we emphasize our unique analysis on: isolated harness design, data contamination mitigation, and prompt standardization, which went beyond existing methods.

*Q2: When harness modules saturate or diminish.*

We discuss when perception and memory modules could diminish, although for now this is far from the case.

### Reviewer zKTd
*W1: Need to expand the explanation of metrics.*

We revise the paper as suggested, and add procedural progress scores in Appendix for completeness.

### Reviewer tEMA

*W1/Q1: What’s the difference between their “harness” compared to typical scaffoldings used in related benchmarks?*

We emphasize that prior works bundle their scaffoldings into a single unified evaluation, unable to evaluate isolated capabilities. Instead, our design enabled fine-grained diagnose and discuss harness effectiveness based on model capability and game visual complexity.

*W2/Q2: Novelty of the benchmark and new insights*

We clarified our contributions regarding game choice, agentic design and evaluation. In a follow-up comment, we compared our work with two related works in detail, highlighting our game diversity and novel insights on perception, spatial-temporal reasoning and long-context reasoning.

*W3/Q3: Why using o3 for perception for Super Mario Bros and Ace Attorney?*

We justified the VLM-as-perception-module for the two games, and added experiments demonstrating that this setup is fair to the evaluated models.

*W4/Q4: Limited number of runs*

We showed that the variances in our games are low enough to achieve on-par or slightly better separability compared with related work. We also ran for additional times and justified reduced variance and consistent rankings.

---

### Meta-Review · Area_Chair_bWjL · 2026-01-06

**Summary:**

This paper introduces LMGame-Bench, a modular, Gym-style benchmark for evaluating large language models and vision–language models through six widely known video games, including Super Mario Bros., Tetris, Sokoban, Candy Crush, 2048, and Ace Attorney. The modular design isolates specific capabilities.

Strengths:
1. The modular design introduced by this paper addresses an important limitation of existing game benchmarks, which is the entanglement of multiple skills.
2. The benchmark and experiments are well designed.
3. The paper clearly presents the models’ weaknesses.
4. The inclusion of an explicit mechanism for detecting data contamination.

Weaknesses:
1. Lack of quantitative validations for the qualitative analysis of model failure, including the statistical proportion of each failure cause.
2. The paper relies on raw scales but would benefit from including procedural metrics.
3. There are no RTS, open-world, or multiplayer games.
4. The paper did not provide concrete guidance for scaling evaluations, given the high costs.
5. It is unclear how accurate the texture representations by the perception module are in capturing dynamic spatiotemporal cues in games such as super mario bros.
6. Insights from the study are aligned with prior findings, and thus, the contributions and novelty seem limited.
7. There should be a discussion on why additional modules could lead to performance improvement.
8. Using o3 to generate perception traces seems like a big methodological mistake, as the rest of the models' capabilities will be influenced by another model.

**Reviewer Concerns:**

I think all reviewer concerns have been addressed.

**Reviewer Scores:**

Reviewer GmFK’s initial score was already positive. The rebuttal was adequate, but it is unclear to me whether the reviewer will further raise the score. It is not impossible.

Reviewer uZi7 could raise their score, as their concerns about the diversity of games, cost opacity, and perceptual module efficacy are relatively minor and have been addressed by the rebuttal.

Reviewer 2PLk could raise their score, as their concern about why harness modules lead to improvement has been discussed in the paper, and novelty has been clarified in the rebuttal.

Reviewer zKTd already gave an 8 in their initial review.

Reviewer tEMA seems to have some misunderstanding about the novelty and contribution, the purpose of the no-harness setting, and the use of o3. I find the rebuttal adequate and think there could be a decent chance that they would raise the score.

---

### Decision · Program_Chairs · 2026-01-26

Accept (Poster)